JCB Journal of Cell Biology

# MIM triggers formin to Arp2/3-based actin assembly in membrane remodeling in *Drosophila* embryos

Debasmita Mitra[1], Georgina K. Goddard[2]*, Sanjana S[1]*, Aparna K[1]*, Tom H. Millard[2], and Richa Rikhy[1]

**BAR domain–containing proteins are key regulators of endocytosis and actin remodeling. Their function in morphogenesis remains to be investigated. We report that the I-BAR domain–containing protein, missing-in-metastasis (MIM) (also called MTSS1), promotes branched actin network formation and endocytosis to drive rapid, cyclical plasma membrane remodeling during syncytial divisions in *Drosophila* embryos. Actin-rich villous protrusions in the apical caps in interphase are depleted in metaphase, concurrent with furrow extension between adjacent nuclei. MIM depletion results in a loss of furrow extension and in longer, more abundant apical protrusions containing the formin diaphanous. Branched actin networks promoted by MIM are in balance with bundled actin networks induced by RhoGEF2 and diaphanous. Cyclical recruitment of MIM to the cortex promotes localization of active Rac, the WAVE regulatory complex, and the Arp2/3 complex to drive endocytic membrane remodeling. These findings identify MIM as an integrator of actin and endocytic dynamics that enables rapid membrane remodeling during *Drosophila* syncytial division cycles.**

## Introduction

The cortical actin network in an epithelial cell is functionally versatile and must undergo highly coordinated changes during cell shape remodeling. It is composed of actin filaments, myosin motors, and actin-binding proteins located adjacent to the plasma membrane (Chugh and Paluch, 2018). The cortex contains both branched actin filaments, primarily nucleated by the Arp2/3 complex, and linear, unbranched filaments nucleated by formins (Bovellan et al., 2014; Rosa et al., 2015; Lu et al., 2017; Jiang and Harris, 2019; Rotty and Bear, 2014). Activation of the Arp2/3 complex requires it to associate with nucleation-promoting factors (NPFs), such as SCAR/WAVE, which are a part of the WAVE regulatory complex (WRC), Wiskott–Aldrich syndrome protein (WASP), WASP and Scar homolog, and cortactin, amongst several others. Arp2/3 complex–derived branched networks drive forces in the cell cortex, leading to plasma membrane remodeling events like extension of lamellipodia (Stramer et al., 2010; Moore et al., 2013; Cooper, 2013; Krause and Gautreau, 2014) and scission of vesicles during endocytosis (Krause and Gautreau, 2014; Mund et al., 2018; Jin et al., 2022; Qualmann et al., 2000). Endocytosis driven by NPFs like WASP and Arp2/3 aid the trafficking of cargoes such as Delta and DE-cadherin (DE-Cad) (Trylinski and Schweisguth, 2019) and is also crucial for regulating the epithelial cell shape changes (Shivas and Skop, 2012; Patel and Soto, 2013).

The early syncytial nuclear division cycles in *Drosophila* embryos are driven by actin polymerization and membrane

trafficking pathways and represent an excellent model for examining how actin and membrane dynamics are coordinated during cell shape changes (Warn, 1986; Foe et al., 2000; Riggs et al., 2003; Pelissier et al., 2003; Grosshans et al., 2005; Yan et al., 2013; Holly et al., 2015; Mavor et al., 2016; Xie and Blankenship, 2018; Zhang et al., 2018; Rikhy et al., 2015). Fertilized embryos undergo nine rounds of nuclear divisions in the center of the embryo, following which the nuclei migrate to the cortex of the syncytium and undergo four more rounds of nuclear division. An actin cap is formed at the cortex of a bulge on the top of each nucleus in the interphase of nuclear cycles 10–13 (Foe and Alberts, 1983; Miller et al., 1985; Turner and Mahowald, 1976). This actin cap consists of actin filaments in the form of both branched and bundled networks (Zhang et al., 2018). The periphery of the actin cap contains microprojections, which are F-actin–rich, needle-shaped, villous protrusions seen in grazing sections (Sherlekar et al., 2020; Mitra et al., 2022; Turner and Mahowald, 1976). These apical actin protrusions remodel and reduce in number as the cell cycle progresses to metaphase, and this process is coincident with the formation of pseudo-cleavage furrows and further increase in their length (Fig. 1 A). Interestingly, during this stage of syncytial nuclear cycles, Arp2/3 complex activity also displaces actin bundles enriched with the formin diaphanous (Dia) toward the cell circumference in the cap to expand the apical area (Jiang and Harris, 2019). The Arp2/3 complex meets the actomyosin borders to form pseudo-cleavage

[1]Indian Institute of Science Education and Research Pune, Pune, India; [2]The University of Manchester, Manchester, UK.

*G.K. Goddard, S. S, and A. K contributed equally to this paper. Correspondence to Richa Rikhy: richa@iiserpune.ac.in.

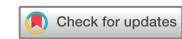

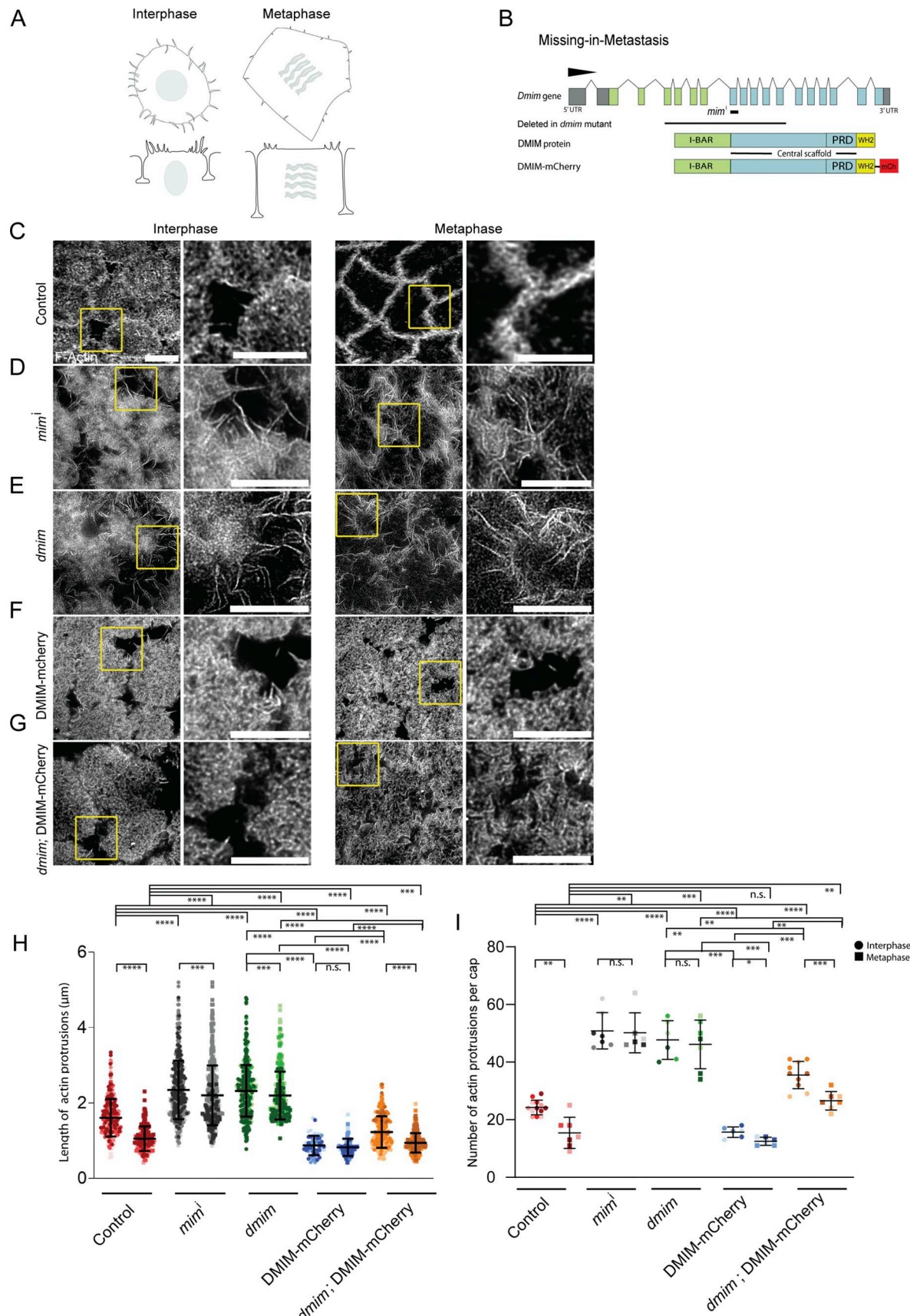

Figure 1. **DMIM depletion leads to increased actin protrusions in interphase and metaphase of the syncytial cycle 12. (A)** Schematic showing the grazing and sagittal sections of plasma membrane organization in interphase and metaphase of syncytial *Drosophila* blastoderm embryo. **(B)** Schematic showing

the gene organization, protein domains of DMIM, and the strategies used to knockdown and knockout the gene. **(C–G)** STED images showing phalloidin-labelled protrusions at interphase and metaphase of nuclear cycle 12 for control (C), *mim* RNAi (D), *dmim* (E), MIM overexpression (*nanos*-Gal4; UASp-DMIM-mCherry) (F), and *dmim*; DMIM-mCherry (G) embryos. The region in the yellow box is further zoomed (2.5×) in the right panel to show the actin protrusions. **(H and I)** Graphs showing the actin protrusion length (H) and numbers per apical cap (I) at interphase and metaphase; *n* = 2–3 cells from 3 embryos for each genotype, 50–200 protrusions from each embryo, and 3 embryos for each genotype. The plots are represented as super plots with different shades of the same color to distinguish different embryos of the same genotype. Data are represented as mean ± SD; *P < 0.05, **P < 0.01, ***P < 0.001, and ****P < 0.0001, n.s. not significant; Mann–Whitney test. Scale bar: 5 µm.

furrows (Zhang et al., 2018). Activation of the Arp2/3 complex in the syncytial embryo cortex is regulated by NPFs like SCAR, Dpod1, cortactin, coronin, and CARMIL. A combinatorial activity of these NPFs together with the Arp2/3 complex determines the apical cap expansion and subsequent pseudo-cleavage furrow ingression (Xie et al., 2021; Stevenson et al., 2002; Henry et al., 2022; Zallen et al., 2002). The trafficking machinery of the cell is also an important regulator of furrow ingression at this point of early *Drosophila* embryogenesis. A Rab39–lp98A–Rab35-mediated endocytic recycling pathway is essential for progression of furrow ingression (Miao et al., 2023). Rab8-mediated exocytic machinery enables the transfer of cargo through membrane vesicles to the furrow for appropriate extension (Mavor et al., 2016). The interaction of actin dynamics with endocytosis has not been well studied in the syncytial *Drosophila* embryos. Establishing how actin remodeling and endocytosis are integrated to achieve rapid cell shape changes in the syncytial stages of *Drosophila* embryogenesis will lead to a better understanding of their interaction during other morphogenetic processes involving rapid membrane remodeling.

The BAR domain family of proteins are strong candidates to integrate cytoskeletal dynamics and membrane trafficking to regulate cell shape remodeling (Zhao et al., 2011; Frost et al., 2009). Syndapin, an F-BAR domain–containing protein and a positive regulator of endocytosis, is crucial for regulating apical actin protrusion remodeling in syncytial *Drosophila* embryos (Sherlekar and Rikhy, 2016; Sherlekar et al., 2020). Here, we have characterized the role of the I-BAR domain–containing protein missing-in-metastasis (MIM) in driving plasma membrane remodeling by regulating F-actin networks and endocytosis through Arp2/3-mediated actin polymerization in syncytial *Drosophila* embryos. MIM, also known as metastasis suppressor protein 1 (MTSS1), is a multidomain protein that is known to regulate actin cytoskeleton remodeling. It contains an I-BAR domain that is known to lead to its dimerization and bind and bundle actin (Yamagishi et al., 2004; Cao et al., 2012) and also possesses a central scaffold region containing a proline-rich domain (PRD) and a WASP homology 2 (WH2) domain at the C terminus (Fig. 1 B). MIM/MTSS1 interacts with the PIP2 on the plasma membrane through its l-BAR domain, which also enables it to act as a scaffold for Rac1 and promote Arp2/3-mediated actin assembly (Saarikangas et al., 2011; Saarikangas et al., 2015; Bompard et al., 2005; Mattila et al., 2007; Machesky and Johnston, 2007). The PRD in the central scaffold region enables binding to the SH3 domain of other actin network regulator proteins like cortactin (Lin et al., 2005). The WH2 domain allows MIM to regulate the actin cytoskeleton via an interaction with actin monomers (Woodings et al., 2003; Mattila et al., 2003). MIM regulates the trafficking and internalization

of various cargoes (nanoparticles and receptors) through interacting with the actin cytoskeleton and Rab GTPases (Sathe et al., 2018; Yu et al., 2011; Zhao et al., 2016; Zhao et al., 2019; Li et al., 2017; Wang et al., 2023). The I-BAR domain family of proteins also induces protrusions or supports negative curvature of membranes (Zhao et al., 2011) and inhibits endocytosis (Quinones et al., 2010). Likewise, the function of MIM in membrane trafficking varies across tissues. It can also either promote or inhibit endocytosis based on the density of cells in a tissue (Dawson et al., 2012). Together, this shows that MIM is a scaffold protein that facilitates formation of branched actin networks and also regulates plasma membrane deformation through the integration of actin assembly and membrane trafficking.

In this study, we find that *Drosophila* MIM (DMIM) plays a crucial role in regulating plasma membrane remodeling in the syncytial cycles in *Drosophila* embryogenesis. DMIM localizes to the plasma membrane via its I-BAR domain and promotes recruitment of Rac-GTP and actin-regulatory proteins Arp3, cortactin, and SCAR to the cortex during the syncytial division cycle to regulate branched actin network and restrict the formation of villous-like actin protrusions via the formin Dia. DMIM also promotes branched actin-mediated endocytosis and drives remodeling of villous-like actin protrusions and furrow formation. We find that activated Rac is essential for localization of DMIM to the cortex for plasma membrane remodeling and endocytosis.

## Results

### Depletion of DMIM leads to sustained actin protrusions in the syncytial division cycle in *Drosophila* embryogenesis

Villous protrusions in apical caps undergo rapid remodeling during *Drosophila* syncytial division cycles and cellularization (Mitra et al., 2022; Sherlekar et al., 2020; Fabrowski et al., 2013); however, the molecular mechanisms that regulate this are poorly understood. To visualize apical protrusions in syncytial division cycle 12, we imaged in embryos expressing GFP-tagged PH domain of general receptor for phosphoinositides-1 that localizes to plasma membrane in the presence of phosphatidylinositol-3,4,5-trisphosphate (tGPH) (Britton et al., 2002) stained with fluorescently coupled phalloidin to label F-actin. We used superresolution STED microscopy in interphase and metaphase of syncytial cycle 12 to visualize the apical caps in grazing sections. tGPH labelled the entire apical cap and showed colocalization with phalloidin in villous-like actin protrusions at the cap periphery. We henceforth refer to these as actin protrusions or apical actin protrusions. They were more distinctly visible at the cap periphery between adjacent caps as compared with the surface of apical caps in these grazing sections (Fig. S1 A).

We assessed the role of DMIM in regulating the distribution of actin protrusions in interphase and metaphase of syncytial cycle 12 by staining with fluorescently coupled phalloidin in fixed embryos. We quantified the length and number of these protrusions at the cap periphery. In control embryos, the actin protrusions were enriched at the edges in interphase caps but were significantly reduced in number and length in metaphase (Fig. 1, A, C, H, and I) (Mitra et al., 2022; Sherlekar et al., 2020). We used two different strategies to obtain DMIM-depleted embryos: shRNA expression and a previously generated *dmim* mutant (Fig. 1 B). *mim* shRNA (*mim*[i]) was crossed to maternal Gal4 (see Materials and methods) to deplete DMIM during oogenesis and embryogenesis. Embryos from F1 females containing Gal4 and shRNA were collected and stained with phalloidin. In embryos expressing *mim*[i], the actin protrusions appeared to be significantly longer than the control in both interphase and metaphase. The *dmim* allele was previously generated from excision of a P transposon and has a deletion of most of the gene (Fig. 1 B). It lacks the conserved I-BAR domain and much of the scaffolding domain and is a loss-of-function mutant (Quinones et al., 2010). The *dmim* mutant embryos showed partial lethality, and 30% of the embryos did not hatch at 24 h after egg laying at 25°C (see Materials and methods). We quantified the protrusions from phalloidin-stained embryos to estimate their numbers and lengths of in-control and DMIM-depleted embryos. *dmim* embryos showed a significant increase in apical actin protrusion length and numbers per cap as compared with controls in interphase and metaphase and similar to *mim*[i] (Fig. 1, D, E, H, and I). To further test the defect in actin protrusions, we combined *dmim* embryos with a fly line containing a deletion of the *dmim* locus to obtain the *dmim*/Def combination adult flies. The embryos from these flies also showed increased actin protrusion numbers and length similar to *dmim*, and this defect was not seen in the embryos from heterozygous *dmim*/+ and Def/+ mothers (Fig. S1, B–F).

We cloned the cDNA of DMIM, comprising of the I-BAR and the central scaffold domain containing the PRD and WH2 domain, and generated a fusion with mCherry to study the distribution of DMIM in the syncytial division cycles (UASp-DMIM-mCherry) (Fig. 1 B). DMIM was overexpressed during oogenesis and embryogenesis by crossing UASp-DMIM-mCherry to maternal Gal4 in the background of the *dmim* mutant and control embryos (see Materials and methods). Embryos were collected from the F1 females containing the UASp-DMIM-mCherry and maternal Gal4 and were assessed for actin protrusion length distribution due to overexpression of DMIM. The actin protrusion length and number at interphase and metaphase in DMIM-mCherry–expressing embryos was significantly lower than the control, *dmim*, and embryos expressing *mim*[I] (Fig. 1, C, F, H, and I). The average apical actin protrusion length and number per cap in *dmim*; DMIM-mCherry embryos was found to be significantly lower than that of the *dmim* embryos and *mim*[i]-expressing embryos at both interphase and metaphase. Therefore, DMIM-mCherry expression alleviated the excess protrusion number and length defect seen on MIM depletion (Fig. 1, D, E, and G–I). Overall, in the absence of DMIM, the actin protrusions increased in number and length in interphase and failed to remodel and reduce in metaphase in the apical cap. The overexpression of DMIM reduced the length and number of the actin protrusions significantly in the interphase of nuclear cycle 12. We therefore conclude that DMIM is important for limiting the length and number of the actin protrusions as well as facilitating their remodeling to reduce the number per apical cap and length from interphase to metaphase of nuclear cycle 12.

## DMIM regulates cap and furrow dynamics in the syncytial division cycles

The plasma membrane in apical caps and furrows between adjacent nuclei shows extensive remodeling in the syncytial division cycles. We visualized apical cap area and furrow dynamics in control and *mim*[i] embryos in syncytial cycle 12 in embryos expressing tGPH. These apical caps, containing protrusions at the periphery (Fig. 2 A, yellow arrows, zoomed insets) in living embryos, expand until they collide with their neighbors, and lateral furrows form between adjacent nuclei (Fig. 2 A, zoomed insets, Video 1). The furrow length increases from interphase to metaphase (Fig. 2 A, sagittal view). In *mim*[i]-expressing embryos, the apical cell area was found to be significantly larger than controls in interphase and metaphase (Fig. 2 D). The plasma membrane boundaries possessed conspicuous protrusions along the periphery in *mim*[i]-expressing embryos (Fig 2 B, marked with yellow arrows, zoomed insets, Video 2). While in tGPH/+ embryos, these protrusions disappeared over time; the protrusions were visible in *mim*[i] from interphase to metaphase. The furrow length was also shorter in *mim*[i] in interphase and metaphase than controls (Fig. 2, A and B sagittal view, 2E). The embryos expressing DMIM-mCherry showed more rounded caps with decreased protrusions and apical cap areas in metaphase compared with controls and *mim*[i] (Fig. 2, C and D; and Video 3). The furrow length of DMIM-mCherry–expressing embryos was comparable with the control embryos at interphase, whereas the furrow remained shorter than the tGPH/+ embryos at metaphase (Fig. 2, A and C sagittal view, 2E). Increased apical actin protrusion numbers were coincident with increased cap area and decreased furrow length in DMIM-depleted embryos.

## DMIM localizes cortically at the cap edges and in furrows in interphase and becomes cytosolic at metaphase

We generated an antibody against DMIM (see Materials and methods) and immunostained embryos with DMIM and polarity protein Dlg. DMIM antibody staining was present throughout the apical cap (Fig. 3 A) and on the plasma membrane of the subapical cap periphery in interphase. DMIM was reduced in metaphase, while Dlg appeared uniform in both interphase and metaphase furrows in these embryos. DMIM antibody fluorescence was not observed in *dmim* mutant embryos confirming a loss of the DMIM protein (Fig. 3 B). MIM was found to be overexpressed in DMIM-mCherry embryos as compared with controls (Fig. S1, G and H). This was consistent with the decrease in actin protrusions seen due to the overexpression (Fig. 1).

We visualized DMIM-mCherry (Fig. 3 C) in *dmim* mutant embryos (*dmim*; DMIM-mCherry) to study the localization of DMIM in syncytial division cycle 12 (Fig. 3 E). DMIM-mCherry was recruited to the cortical region in the cap membrane at

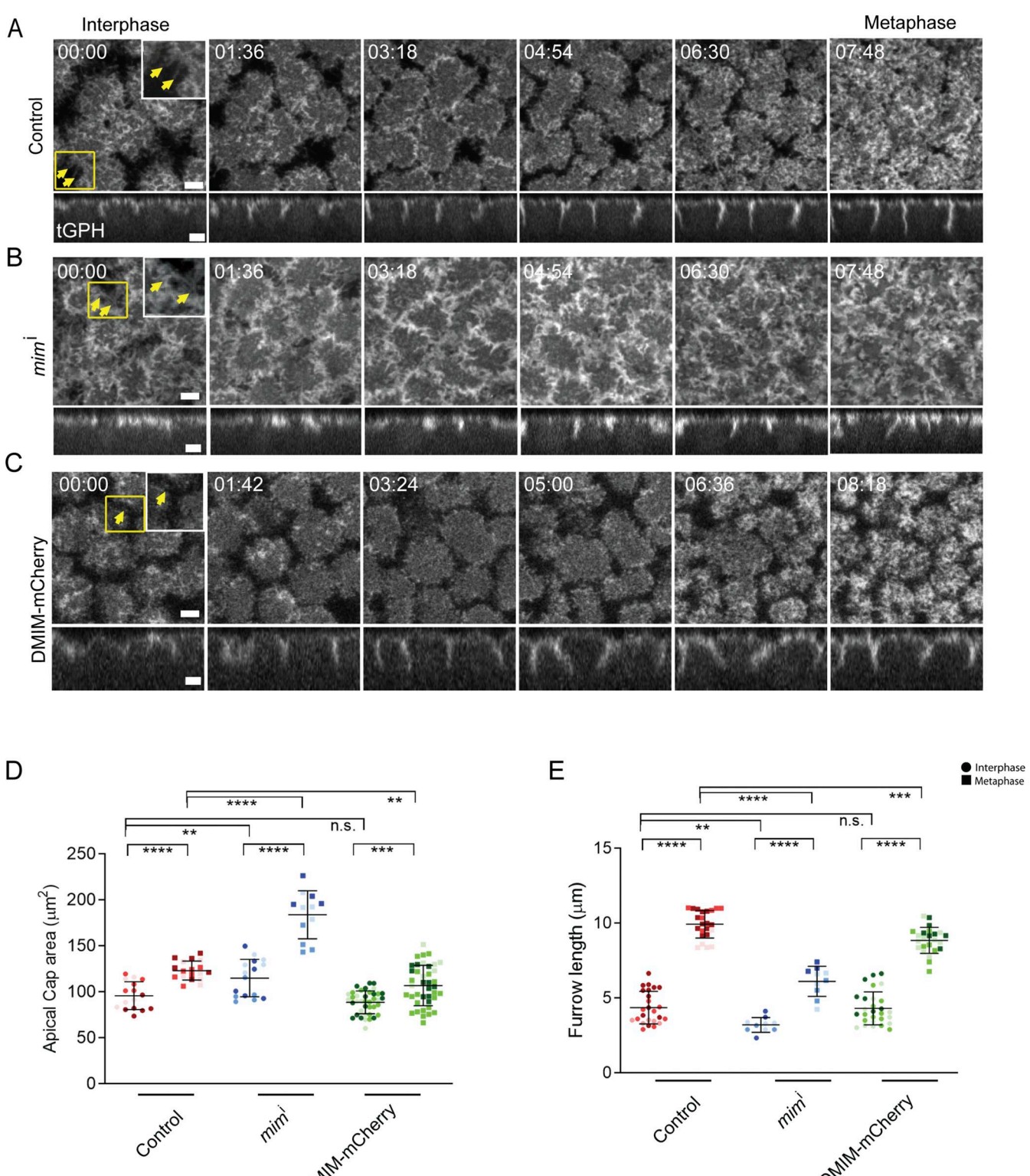

Figure 2. **DMIM depletion leads to altered cap area and furrow dynamics in syncytial cycle 12**. **(A–C)** Representative images of the grazing and sagittal sections of the syncytial embryos in cycle 12, from interphase till metaphase, expressing tGPH labelling the plasma membrane of control (A) *mim*[i] (yellow arrows point to increased actin protrusions) (B) and DMIM-mCherry (C) embryos. **(D and E)** The region in the yellow box is further zoomed (2.5×) in the right panel to show the actin protrusions (D and E). Yellow arrows mark the actin protrusions. Graphs are showing the quantification of apical cell area (D) and furrow length (E) at interphase and metaphase. *n* = 5 cells from 3 embryos for each genotype. The plots are represented as super plots with different shades of the same color to distinguish different embryos of the same genotype. Data are represented as mean ± SD; **P < 0.01, ***P < 0.001, and ****P < 0.0001, n.s. not significant; Mann–Whitney test. Time is in min:sec. Scale bar: 5 μm.

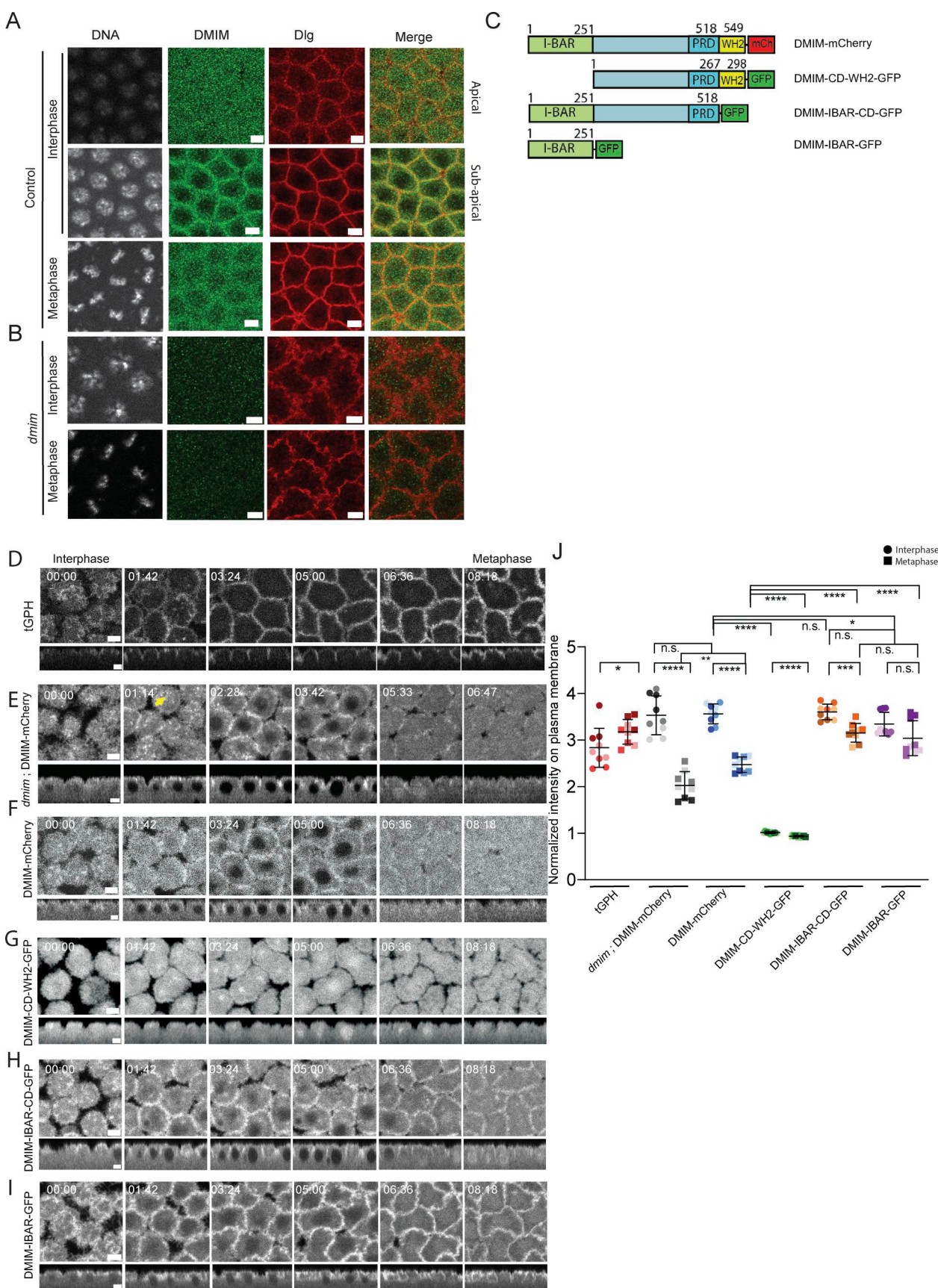

Figure 3.  **DMIM enriches cortically and at the furrow edge with the I-BAR domain in interphase and prophase and is cytosolic in metaphase of syncytial cycle 12. (A and B)** Representative grazing section images from control (A) and *dmim* (B) embryos in interphase (apical and lateral views for control)

and metaphase, showing nuclei labelled with Hoechst (gray), endogenous DMIM labelled with anti-DMIM antibody (green), and plasma membrane marked with polarity protein Dlg (red). **(C)** Schematic showing the domains of the full-length DMIM and the different domain deletion constructs. **(D and E)** Representative images of the grazing and sagittal sections of the syncytial embryos in nuclear cycle 12, from interphase till metaphase, expressing tGPH (D) labelling the plasma membrane and DMIM-mCherry in the *dmim* mutant embryos (E). **(F–I)** Representative images of the grazing and sagittal sections of the syncytial embryos in nuclear cycle 12, from interphase till metaphase, expressing DMIM-mCherry (F), DMIM-CD-WH2-GFP (G), DMIM-IBAR-CD-GFP (H), and DMIM-IBAR-GFP (I). **(J)** Graph showing an analysis of normalized fluorescence intensity of tGPH, dmim; DMIM-mCherry, DMIM-mCherry, DMIM-CD-WH2-GFP, DMIM-IBAR-CD-GFP, and DMIM-IBAR-GFP in interphase and metaphase of nuclear cycle 12. *n* = 3 cells from 3 embryos for each genotype. The plots are represented as super plots with different shades of the same color to distinguish different embryos of the same genotype. Data are represented as mean ± SD; *P < 0.05, **P < 0.01, ***P < 0.001, and ****P < 0.0001, n.s. not significant; Mann–Whitney test. Time is in min:sec. Scale bar: 5 µm.

furrows and was found in vesicles (Fig. 3 E yellow arrow) in interphase and prophase. DMIM-mCherry in *dmim*; DMIM-mCherry was depleted from the furrow in metaphase (Fig. 3, E and J; and Video 4). We overexpressed DMIM-mCherry with tGPH to visualize their relative spatiotemporal dynamics. While tGPH was present at the plasma membrane, DMIM-mCherry became predominantly cytosolic as the nuclear cycle progressed to metaphase similar to *dmim*; DMIM-mCherry (Fig. 3, D, F, and J; Fig. S1 I; and Video 3). A significant decrease in membrane intensity was observed in metaphase as compared with interphase for DMIM-mCherry, while tGPH increased on the membrane (Fig. 3, D, F, and J).

Further, to evaluate the role of the I-BAR and WH2 domains in the timely recruitment of DMIM to the plasma membrane, we generated three truncated DMIM transgenes: (1) containing the CD and WH2 domains and deletion of the I-BAR domain (UASp-DMIM-CD-WH2-GFP), (2) containing the I-BAR and CD domains and deletion of the WH2 domain (UASp-DMIM-IBAR-CD-GFP), and (3) containing the I-BAR domain and deletion of the CD and WH2 domain (UASp-DMIM-IBAR-GFP), tagged with GFP at the C terminus in the UASp vector (see Materials and methods for details) (Fig. 3 C). We expressed the GFP-tagged domain deletion lines with maternal Gal4 and visualized the fluorescence in living embryos in syncytial nuclear cycle 12. The DMIM-CD-WH2-GFP was found to be enriched in the centrosome region and was predominantly cytoplasmic and did not become enriched at the cortex or the plasma membrane from interphase to prophase of the syncytial cycle as observed in DMIM-mCherry–expressing embryos (Fig. 3, G and J; and Video 5). The DMIM-IBAR-CD-GFP localized to the plasma membrane in interphase but remained on the plasma membrane to a greater extent than DMIM-mCherry in metaphase. The levels of DMIM-IBAR-CD-GFP reduced slightly in metaphase as compared with interphase but were significantly higher than DMIM-mCherry controls (Fig. 3, H and J; and Video 6). The DMIM-IBAR-GFP localized to the plasma membrane in interphase and like DMIM-IBAR-CD-GFP retained this localization in metaphase. The DMIM-IBAR-GFP slightly reduced on the membrane in metaphase, but the reduction was not found to be statistically significant as compared with interphase (Fig. 3, I and J; and Video 7).

Together these localization data show that DMIM localizes to the plasma membrane in apical caps and furrows in interphase and becomes cytosolic in metaphase. The I-BAR domain of DMIM is essential for its recruitment to the plasma membrane, and in its absence, as seen in the DMIM-CD-WH2-GFP, it remained cytosolic. The DMIM-IBAR-CD-GFP and DMIM-IBAR-GFP showed a similar distribution of remaining on the furrow in

metaphase, implying a role for the WH2 domain in regulating the removal of MIM from the membrane in metaphase. The DMIM-CD-WH2-GFP remained cytosolic, showing that neither the central domain nor the WH2 domain plays a role in recruiting MIM to the plasma membrane or the cortex. These analyses, however, did not specifically test the role of the central domain in recruitment of MIM, and future studies will be required for addressing its function in MIM recruitment in the syncytial division cycles. These localization data are also consistent with previously described role of the I-BAR domain in the recruitment of MIM to the plasma membrane and vesicles in mammalian NIH3T3 cells and distribution of the WH2 domain in the cytoplasm (Woodings et al., 2003).

Taken together, DMIM is localized to the plasma membrane and vesicles when actin protrusions are also present and is cytoplasmic when these protrusions are depleted in metaphase of the syncytial division cycle. Loss of DMIM leads to retention of actin protrusions in both interphase and metaphase.

## DMIM regulates endocytosis in the syncytial division cycles

Membrane trafficking in the form of endocytosis is a key regulatory mechanism for furrow ingression in the syncytial blastoderm and cellularization stage of *Drosophila* embryogenesis. Dynamin distribution is reduced in metaphase of the syncytial cycle, and dynamin-mediated endocytosis is involved in furrow ingression (Rikhy et al., 2015). The actin protrusions also are cleared through dynamin- and Rab5-dependent endocytosis at a later stage in cellularization (Fabrowski et al., 2013). Perturbing early endosome and apical recycling by expressing dominant-negative Rab5 and Rab11 decreases furrow ingression in cellularization (Pelissier et al., 2003). BAR domain–containing proteins play an important role in regulating endocytosis at the plasma membrane. Activation of Cdc42, with IRSp53, an I-BAR domain–containing protein, and the Arp2/3 complex, leads to a burst of F-actin polymerization and facilitates scission of endocytic vesicles (Sathe et al., 2018). Syndapin, an F-BAR domain–containing protein involved in endocytosis, regulates remodeling of actin protrusions in the syncytial *Drosophila* embryos (Sherlekar and Rikhy, 2016; Sherlekar et al., 2020). MIM has been shown to be a positive regulator of endocytosis in S2 cells and a negative regulator of endocytosis in border cells (Sathe et al., 2018; Quinones et al., 2010). MIM/MTSS1 regulates the endocytic trafficking of chemokine receptor CXCR5 in mammalian HeLa cells (Li et al., 2017).

We assessed endocytosis in *dmim* mutant embryos in the syncytial cycle. We performed the fluorescent dye uptake assay in syncytial blastoderm embryos using the fixable analog of the

amphipathic dye FM1-43, FM1-43FX (see Materials and methods). Living embryos from control and *dmim* genotypes were permeabilized and incubated with the FM1-43FX dye, followed by washing to visualize the dye, which is trapped in vesicles after fixation. Numerous fluorescent vesicles (marked with yellow arrows) labelled with FM1-43FX were seen in control embryos in the interphase of the syncytial cycle. We observed a reduction in vesicles in embryos in metaphase (Fig. 4, A and B), demonstrating that endocytosis is reduced in metaphase as compared with interphase of the syncytial cycle. In *dmim* embryos, the FM1-43FX fluorescence was reduced in interphase and metaphase, indicating that the endocytic uptake of the dye is reduced in *dmim* embryos (Fig. 4, A and B).

We further assessed the distribution of the early endosome marker, Rab5, and recycling endosome marker, Rab11, using immunofluorescence in control and *dmim*. *dmim* embryos showed a reduction in Rab5 vesicles as compared with control embryos. (Fig. 4, C and D, marked with yellow arrows). Rab11 also marked vesicles in the perinuclear region in control embryos. *dmim* embryos showed a reduction of Rab11 vesicles (Fig. 4, E and F, marked with yellow arrows).

The levels of DE-Cad at the plasma membrane vary depending on the rates of endocytosis. Inhibition of dynamin-mediated endocytosis leads to an increased accumulation of DE-Cad at the membrane (Rikhy et al., 2015). Reduced endocytosis on the depletion of polarity protein Par3 homologs Bazooka and Septin Peanut also leads to an accumulation of DE-Cad at the furrow (Dey et al., 2023). We tested if DE-Cad levels changed in DMIM-depleted embryos. We imaged DE-Cad distribution in fixed *dmim* embryos using immunofluorescence with an anti–DE-Cad antibody and in living *mim*[i]-expressing embryos using DE-Cad-GFP. DE-Cad immunofluorescence significantly increased on the plasma membrane in nuclear cycle 12 in *dmim* embryos compared with controls (Fig. 4, G and H). DE-Cad-GFP was seen at the plasma membrane and in vesicles in control embryos. *mim*[i] embryos expressing DE-Cad-GFP showed increased accumulation of DE-Cad-GFP fluorescence at the plasma membrane compared with controls from interphase to metaphase of nuclear cycle 12. DE-Cad was also observed in protrusions at the cortex in movies. There was a reduction in the vesicular pool of DE-Cad in *mim*[i] embryos in the snapshots from live movies (Fig. 4 I and Video 8).

We tested the role of dynamin in regulating remodeling of actin protrusions in the syncytial division cycle by staining temperature-sensitive *shi*[ts2] mutant embryos (Rikhy et al., 2015) with fluorescently coupled phalloidin and performing STED microscopy (Fig. S2, A–D). We found that the length and number of actin protrusions increased in these embryos as compared with controls, further confirming that endocytosis plays a role in limiting them at the cortex in the syncytial division cycle (Fig. S2, A–D).

Taken together, we observed that endocytosis was reduced in embryos depleted of DMIM, and this led to higher levels of DE-Cad at the furrow membrane in the syncytial cycle. DMIM localized to the membrane in interphase and prophase when actin protrusions were abundant, and these stages also correlated with increased endocytosis.

## DMIM depletion leads to decrease in branched actin network regulators SCAR, Arp3, and cortactin and increase in bundled actin regulator Dia in the syncytial division cycles

MIM is a multidomain protein that regulates and interacts with the actin cytoskeleton network. The WH2 domain of MIM is known to bind actin monomers and promote actin assembly. MIM recruits cortactin through its PRD in the central scaffold region. Most importantly, the I-BAR domain in MIM acts as a possible scaffold to Rac1 and promotes its interaction with downstream effectors (Saarikangas et al., 2011; Saarikangas et al., 2015; Bompard et al., 2005; Mattila et al., 2007; Machesky and Johnston, 2007). MIM therefore interacts with the Arp2/3 complex and promotes the branched actin network. Branched actin network and its regulators are also known to play a role in endocytosis (Krause and Gautreau, 2014; Mund et al., 2018; Jin et al., 2022; Qualmann et al., 2000). Therefore, the reduction in endocytosis observed on depletion of DMIM could be due to the loss of the branched actin network.

We tested if the localization of the Arp2/3 complex changed in DMIM-depleted embryos. We assessed the distribution of Arp3 with respect to fluorescently coupled phalloidin in control embryos. We found that Arp3 was present in cortical regions and not present in the apical actin protrusions (Fig. 5 A). We further used immunostaining to visualize the distribution of Arp3 during syncytial cycle 12 in fixed *dmim* mutant and *mim*[i]-expressing embryos (Fig. 5 C). Arp3 was present at the furrow membrane in interphase and metaphase in control embryos; however, its localization was reduced significantly in *dmim* mutant and *mim*[i]-expressing embryos in both stages (Fig. 5 C). We quantified the membrane to cytosolic ratio of Arp3 fluorescence intensity and found that it was significantly decreased compared with control embryos (Fig. 5 D).

We also tested the levels of the Arp2/3 activator, SCAR/WAVE, and stabilizer of the Arp2/3 network, cortactin, in embryos depleted of DMIM. Cortactin was visualized in *mim*[i]-expressing embryos by immunostaining. Cortactin was present at the furrow membrane in interphase and metaphase in control embryos; however, cortactin fluorescence was reduced from the furrows in DMIM-depleted embryos compared with the control in both stages (Fig. 5, E and F). SCAR/WAVE levels were estimated in *dmim* embryos by immunostaining for SCAR. SCAR was present at the membrane in both interphase and metaphase control embryos. SCAR fluorescence was also reduced from the furrows in MIM depletion embryos as compared with the controls in both stages (Fig. 5, G and H). The reduction in localization in Arp3, cortactin, and SCAR was found in both the apical and subapical regions (Fig. 5, C–H and Fig. S2, E–G). These data suggest that recruitment and activation of the Arp2/3 complex is reduced in the absence of DMIM, likely resulting in loss of branched actin filaments.

A reduction in branched actin can also affect the composition of the cortical actin network. Arp2/3 network reduction leads to increased Dia-enriched F-actin bundles in the caps (Zhang et al., 2018). Since *dmim* embryos showed excess actin protrusions and loss of branched actin network–promoting proteins, we tested if this coincides with increased bundled actin to form excess actin protrusions. We, therefore, assessed the levels of Dia, a formin

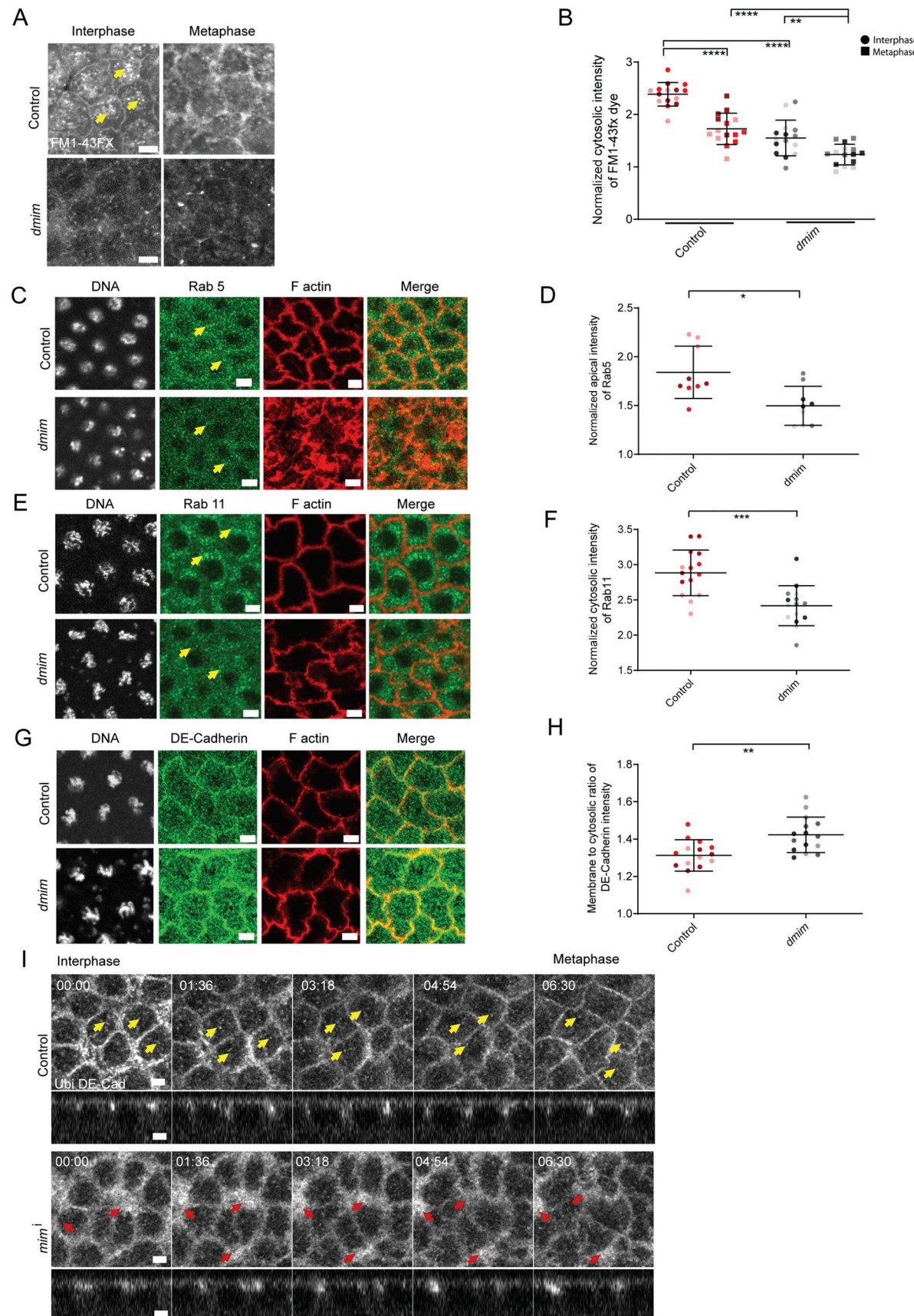

Figure 4. **DMIM depletion decreases endocytosis during the syncytial division cycles. (A)** Representative grazing section images of control and *dmim* embryos labelled with FM1-43FX dye at interphase and metaphase. **(B)** Graph showing the quantification of the normalized cytosolic intensity of the FM1-43FX

dye at interphase and metaphase in control and *dmim* embryos. *n* = 5 cells from 3 embryos for each genotype. **(C)** Representative grazing section images showing nuclei labelled with Hoechst (gray), early endosomes labelled with Rab5 (green), and cortical F-actin labelled with Alexa Fluor Phalloidin 568 (red) from control and *dmim* embryos. **(D)** Graph showing the quantification of the normalized apical intensity of Rab5 in control and *dmim* embryos. *n* = 3 cells from three embryos for each genotype. **(E)** Representative grazing section images showing the nuclei labelled with Hoechst (gray), recycling endosomes labelled with Rab11 (green), and cortical F-actin labelled with Alexa Fluor Phalloidin 647 (red) from control and *dmim* embryos. **(F)** Graph showing the quantification of the normalized intensity of Rab11 in control and *dmim* embryos. *n* = 5 cells from 3 embryos for each genotype. **(G)** Representative grazing section images containing nuclei labelled with Hoechst (gray), DE-Cad (green), and cortical F-actin labelled with Alexa Fluor Phalloidin 568 (red) in control and *dmim* embryos. **(H)** Graph showing quantification of the membrane to cytosolic ratio of DE-Cad. *n* = 5 cells from 3 embryos for each genotype. **(I)** Representative images of the grazing and sagittal sections of the syncytial epithelial cells in nuclear cycle 12, from interphase till metaphase, expressing Ubi DE-Cad-GFP in control (F) and *mim*[i] (G) embryos. Yellow arrows mark the cytosolic vesicles of DE-Cad in control embryos. Red arrows mark the excess DE-Cad accumulated on the plasma membrane in *mim*[i] embryos. The plots are represented as super plots with different shades of the same color to distinguish different embryos of the same genotype. Data are represented as mean ± SD; *P < 0.05, **P < 0.01, ***P < 0.001, and ****P < 0.0001; Mann–Whitney test. Time is in min:sec. Scale bar: 5 µm.

that regulates the formation of bundled actin structures in nuclear cycle 12 of *Drosophila* syncytial blastoderm (Jiang and Harris, 2019) depleted of DMIM. Dia was visualized in Dia-GFP–expressing embryos with respect to fluorescently coupled phalloidin in control embryos. Dia-GFP was present in the apical actin protrusions (Fig. 5 B). We further imaged the Dia distribution in fixed *dmim*- and *mim*[i]-expressing embryos by immunostaining in metaphase. Dia antibody fluorescence was increased in the apical cap in protrusions in both *dmim*- and *mim*[i]-expressing embryos, whereas Dia was present at the furrow membrane in controls. We quantified the intensity of Dia antibody fluorescence in the DMIM-depleted embryos and found that it was significantly increased at the cortex as compared with the controls (Fig. 5, I and J).

Based on these data, we conclude that DMIM is present at the cortex on the plasma membrane, when actin protrusions are present, and its function in regulating branched actin network and endocytosis restricts actin protrusion formation and promotes furrow formation. This allows the maintenance of apical actin protrusion length and number from interphase to metaphase, through branched actin-mediated endocytosis. When DMIM is absent, there is a loss of the balance between Dia- and Arp2/3-derived actin network favoring apical actin protrusion formation rather than endocytosis.

**DMIM overexpression leads to increased endocytosis and Arp3 and decreased Dia at the apical membrane**
DMIM-mCherry overexpression led to shorter actin protrusions (Fig. 1). DMIM-mCherry is present in the apical cap in these embryos (Fig. 6 A). Since overexpression of DMIM led to reduced actin protrusions, we tested endocytosis and recruitment of branched and bundled actin machinery in DMIM-overexpressing embryos. We performed the FM1-43FX dye uptake assay in embryos obtained from flies overexpressing DMIM-mCherry. Large fluorescent vesicles were visible in DMIM-mCherry–expressing embryos in interphase, and vesicles were also more abundant in metaphase as compared with controls (Fig. 6 B). We found that the apical region showed increased FM1-43FX fluorescence intensity compared with the control embryos in both interphase and metaphase (Fig. 6 C).

DMIM-mCherry–expressing embryos stained with phalloidin and Arp3 showed increased levels of apical Arp3 intensity (Fig. 6, D and E) as well as significantly higher cortical Arp3 spread over a wider region across the plasma membrane in

DMIM-mCherry embryos as compared with control images, while DMIM-mCherry is present on the membrane (Fig. 6, D, F, and G). Since changing DMIM levels affected the Arp3 distribution and its intensity, we assessed the colocalization of Arp3 and DMIM and found there is a partial colocalization between DMIM-mCherry (red) and Arp3 (green) with a Pearson's correlation coefficient = 0.527 ± 0.146, P < 0.0001 (*n* = df-2) (*n* = 12, 3 cells from 4 embryos) (Fig. 6 G). The fluorescence of DMIM-mCherry was at the furrow membrane, whereas Arp3 was present at the membrane as well in the cortical region. Interestingly, apical levels of Dia were significantly reduced in embryos overexpressing DMIM (Fig. 6, H and I). This confirmed that DMIM is a positive regulator of the Arp2/3 recruitment and endocytosis and a negative regulator of apical Dia levels. We find that DMIM was present at the membrane and endocytosis increased on its overexpression; however, future experiments will be needed to evaluate if DMIM is present at endocytic pits along with the branched actin network to directly mediate endocytic vesicle formation.

Together, these data suggest that MIM could regulate the extent of recruitment of proteins involved in the branched actin assembly while simultaneously limiting levels of bundled actin. It is likely that this increased recruitment of branched actin regulators by DMIM promotes endocytosis and apical actin protrusion remodeling.

**Rac-GTP accumulation is reduced and Rho-GTP accumulation is increased in DMIM-depleted embryos**
Rac promotes activation of the Arp2/3 complex when in its active, GTP-bound state (Eden et al., 2002), so we hypothesized that DMIM might increase Arp2/3-based actin assembly by regulating Rac-GTP accumulation in the cap. Consistent with this, the I-BAR domain of MIM binds to Rac-GTP, and overexpression of MIM has been shown to increase Rac activity in mammalian COS-7 cells (Bompard et al., 2005). We observed that total Rac1 levels did not change in DMIM-depleted embryos in interphase and metaphase (Fig. 7, A and B), suggesting that DMIM does not regulate Rac1 expression. To test for effects of DMIM on Rac-GTP distribution, we used a GFP-tagged form of the Rac-binding domain of PAK3 (PAK3-RBD-GFP), which has been shown to bind specifically to GTP-bound Rac1 (Abreu-Blanco et al., 2014). We expressed PAK3-RBD-GFP in *mim*[i]- and DMIM-mCherry–expressing embryos and visualized the distribution of its fluorescence in living embryos in syncytial division

Figure 5. **DMIM depletion leads to loss of Arp3, SCAR, and cortactin and increase of Dia in syncytial cycle 12. (A and B)** Representative images of apical caps in embryos immunostained with Arp3 (green) and phalloidin (red) (A) and Dia-GFP (green)-expressing embryos stained with phalloidin (red) (B). Yellow

arrows mark the actin protrusions. The region in the yellow box is further zoomed in the panel below to show the actin protrusions. **(C)** Representative grazing section images showing nuclei labelled with Hoechst (gray), Arp3 (green), and cortical F-actin labelled with Alexa Fluor 568 (red) in control, *mim*[i], and *dmim* embryos. **(D)** Graph showing the quantification of membrane to cytosolic ratio of Arp3 in control, *mim*[i], and *dmim* embryos; n = 5 cells from 3 embryos for each genotype. **(E)** Representative grazing section images showing nuclei labelled with Hoechst (gray), Arp2/3 network activator cortactin (green), and cortical F-actin labelled with Alexa Fluor 568 (red) in control and *mim*[i] embryos. **(F)** Graphs showing the quantification of the membrane to cytosolic intensity of cortactin in control and *mim*[i] embryos. n = 5 cells from 3 embryos for each genotype. **(G)** Representative grazing section images showing nuclei labelled with Hoechst (gray), Arp2/3 network activator SCAR/WAVE (green), and cortical F-actin labelled with Alexa Fluor 568 (red) in control and *dmim* embryos. **(H)** Graph showing the quantification of the membrane to cytosolic intensity of SCAR/WAVE in control and d*mim* embryos. n = 3 cells from 3 embryos for each genotype. **(I)** Representative grazing section images showing nuclei labelled with Hoechst (gray), Dia (green), and cortical F-actin labelled with Alexa Fluor 568 (red) in control, *mim*[i], and *dmim* embryos. **(J)** Graph showing the quantification of normalized apical intensity of Dia in control, *mim*[i], and *dmim* embryos. n = 5 cells from 3 embryos for each genotype. The plots are represented as super plots with different shades of the same color to distinguish different embryos of the same genotype. Data are represented as mean ± SD; **P < 0.01, and ****P < 0.0001, n.s. not significant; Kruskal–Wallis followed by Dunn's post hoc test was performed for Arp3 and Dia intensity analysis; for the analysis of cortactin and SCAR, Mann–Whitney test was performed. Scale bar: 5 µm.

cycle 12. We found that PAK3-RBD-GFP was localized to the cap and the furrow in control embryos. Its distribution at the cap decreased slightly but significantly from interphase to metaphase in control embryos. DMIM-depleted embryos showed a reduction in accumulation of fluorescence in apical caps in the syncytial cycle 12. DMIM-mCherry–overexpressing embryos showed an increase in accumulation of PAK3-RBD-GFP in apical caps, with patches of DMIM-mCherry distinctly colocalizing with PAK3-RBD-GFP. Quantification of the fluorescence of PAK3-RBD-GFP showed a significant decrease in DMIM-depleted embryos and a significant increase in DMIM-mCherry–expressing embryos, both in the apical (Fig. 7, C and E) and lateral membrane (Fig. S3 A). These data show that *Drosophila* DMIM functions in stabilizing Rac-GTP in actin caps.

In mammalian cells, the Rho and Rac pathways are spatially segregated and known to cooperate or mutually inhibit each other during cell migration (Hanna and El-Sibai, 2013; Parri and Chiarugi, 2010). Since DMIM overexpression increased Rac-GTP cap accumulation, we evaluated the Rho pathway by testing the levels of Rho-GTP and immunostaining for RhoGEF2, a RhoGEF previously found as a crucial exchange factor for Rho-GTP in the syncytial blastoderm embryos (Cao et al., 2010; Crest et al., 2012; Barmchi et al., 2005; Dey and Rikhy, 2020). We expressed the Rho-GTP–binding domain of Anillin tagged with GFP (Anillin-RBD-GFP) (Mason et al., 2016; Munjal et al., 2015; Sharma and Rikhy, 2021) in DMIM-depleted and -overexpressing embryos and visualized the fluorescence in living embryos in syncytial division cycle 12. Anillin-RBD-GFP was found to accumulate in sub-apical sections at the furrow in control embryos, but its levels in this location decreased in metaphase. We found that Anillin-RBD-GFP fluorescence significantly increased in *mim*[i]-expressing embryos and significantly decreased in DMIM-mCherry–expressing embryos, both in the lateral membrane (Fig. 7, D and F). The apical fluorescence levels of Anillin-RBD-GFP were very low as compared with the subapical levels (Fig. S3 B). We next tested if change in Rho-GTP as reported by Anillin-RBD-GFP was due to similar change in the RhoGEF2 levels in the syncytial division cycle. We found that RhoGEF2 accumulation in apical caps was significantly increased in *mim*[i]-expressing embryos and decreased in DMIM-mCherry–expressing embryos (Fig. 7, G and H). Increased RhoGEF2 and Rho-GTP accumulation in apical caps are possibly linked to stabilizing Dia in apical caps for supporting the formation of actin protrusions (Fig. 5, I and J).

Since RhoGEF2 and Rho-GTP affect myosin II activation at the contractile ring in *Drosophila* cellularization (Grosshans et al., 2005; Sharma and Rikhy, 2021), we also tested the change in myosin II levels by immunostaining for myosin II light chain Spaghetti squash (Sqh). There was a slight increase in Sqh fluorescence in interphase embryos, but this increase was not statistically significant. Sqh was cytoplasmic, similar to controls in metaphase (Fig. S3, C and D).

The analysis of Rac and Rho pathways together showed that DMIM regulates the accumulation of Rac-GTP at the cortex. DMIM has an opposing effect on the Rho pathway and leads to decrease in RhoGEF2 and Rho-GTP, thereby supporting branched actin morphology at the plasma membrane.

### Active Rac1 and Dia regulates apical actin protrusion remodeling and endocytosis

We further evaluated if Rac1 activation affected apical actin protrusion formation, endocytosis, and recruitment of DMIM in syncytial embryos. The Rac-GEF sponge regulates the activation of Rac in *Drosophila* embryos (Henry et al., 2022). We assessed *sponge* RNAi-expressing embryos (*spg*[i]) for actin protrusions in interphase and metaphase of the syncytial division cycle by staining them with fluorescently coupled phalloidin. We found that, like DMIM-depleted embryos, *spg*[i] embryos also showed longer and more abundant actin protrusions in interphase and metaphase as compared with controls (Fig. 8, A, B, D, and E).

We also expressed a dominant-negative version of Rac1 with a T17N mutation, *Rac1*[DN], in oogenesis and embryogenesis by crossing it to the maternal Gal4 line (Luo et al., 1994). Embryos from F1 females expressing *Rac1*[DN] were stained for Arp3 and Dia. Arp3 levels at the cortex were reduced in *Rac1*[DN]-expressing embryos (Fig. S4 A), while Dia levels were significantly increased (Fig. S4, B and C). We stained *Rac1*[DN] embryos with fluorescently labelled phalloidin and visualized the actin protrusions with STED microscopy. *Rac1*[DN] embryos showed a significant increase in the length and number of actin protrusions as compared with the control embryos in both interphase and metaphase of the syncytial cycle (Fig. 8, A and C–E). We examined endocytic uptake of FM1-43FX in *Rac1*[DN]-expressing embryos. In *Rac1*[DN]-expressing embryos, the fluorescence intensity of the dye in the vesicles was reduced as compared with the control embryos (Fig. 8, F and G). Rab5 accumulation at the cortex in the *Rac1*[DN] embryos was also reduced as compared with

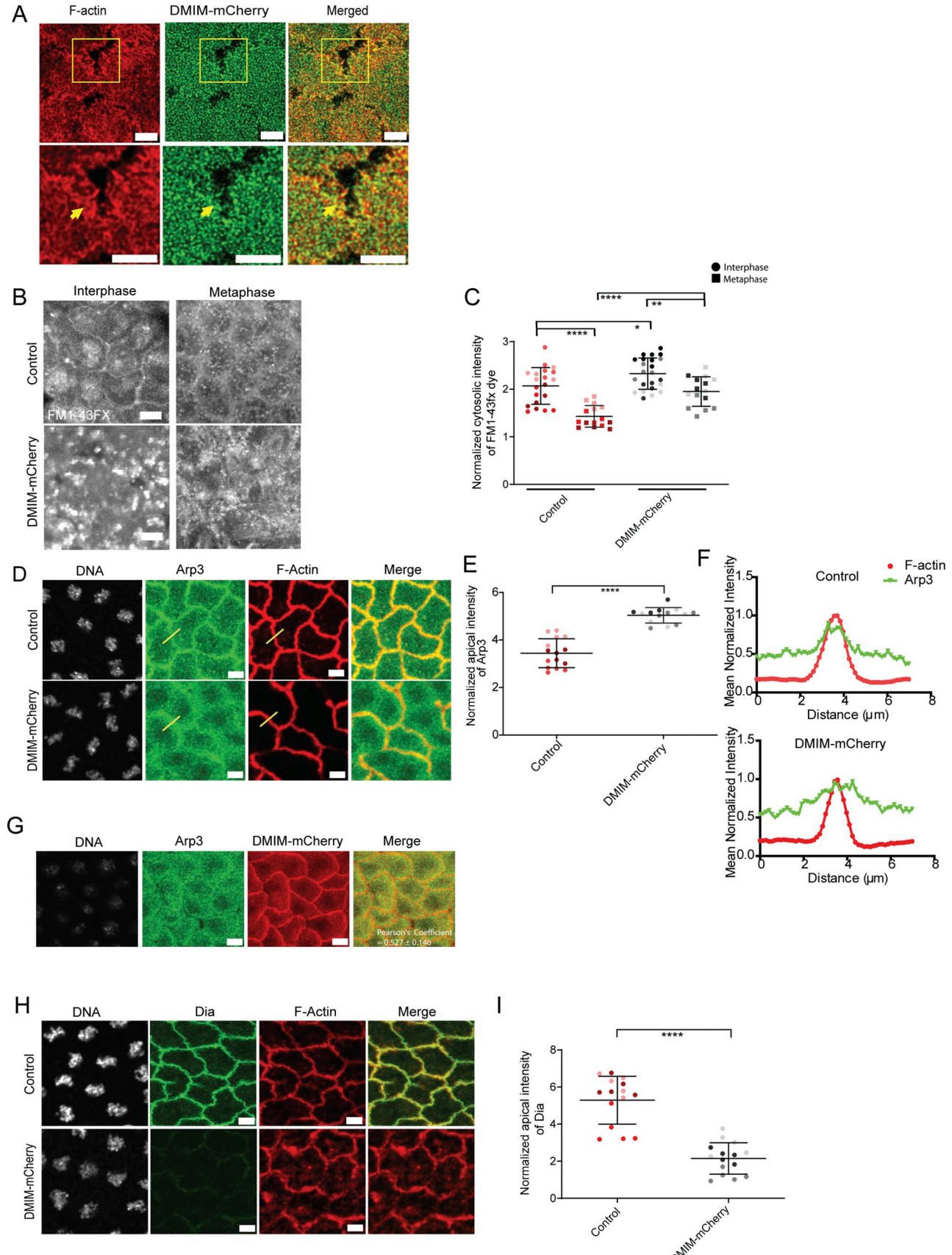

Figure 6. **Overexpressing DMIM increases endocytosis and Arp3 recruitment and reduces Dia in the syncytial division cycles. (A)** Representative images of apical caps in DMIM-mCherry–expressing embryos immunostained with phalloidin. Yellow arrows mark the actin protrusions. **(B)** Representative

grazing section images labelled with FM1-43FX dye at interphase and metaphase in control and DMIM-mCherry embryos. **(C)** Graphs showing the quantification of the normalized cytosolic intensity of the FM1-43FX dye at interphase and metaphase in control and DMIM-mCherry embryos. $n = 5$ cells from 3 embryos for each genotype. **(D)** Representative grazing section images showing nuclei labelled with Hoechst (gray), Arp3 (green), and cortical F-actin labelled with Alexa Fluor Phalloidin 568 (red) in control and DMIM-mCherry embryos. **(E)** Graph showing the quantification of normalized apical intensity of Arp3 in control and DMIM-mCherry embryos. $n = 5$ cells from 3 embryos for each genotype. **(F)** Graphs showing the relative distribution of Arp3 and F-actin along the line drawn across the cell boundary. Data are represented as the mean of 3 line profiles drawn across 3 edges, normalized to the maximum value, from a representative embryo. **(G)** Representative grazing section images showing nuclei labelled with Hoechst (gray), Arp3 (green), and DMIM-mCherry (red). Pearson's correlation coefficient for the colocalization of Arp3 and DMIM-mCherry was found to be $0.527 \pm 0.146$, $P < 0.0001$($n = $ df-2). **(H)** Representative grazing section images showing nuclei labelled with Hoechst (gray), Dia (green), and cortical F-actin labelled with Alexa Fluor 568 (red) in control and DMIM-mCherry embryos. **(I)** Graph showing the quantification of normalized apical intensity of Dia in control and DMIM-mCherry embryos. The plots are represented as super plots with different shades of the same color to distinguish different embryos of the same genotype. Data are represented as mean $\pm$ SD; *$P < 0.05$, **$P < 0.01$, and ****$P < 0.0001$; Mann–Whitney test. Scale bar: 5 μm.

the controls (Fig. S4, D and E). Immunostaining with a DE-Cad antibody also showed that DE-Cad levels increased on the furrow in nuclear cycle 12 in *Rac1*[DN]-expressing embryos as compared with controls (Fig. S4, F and G). Thus, *Rac1*[DN]-expressing embryos phenocopied the endocytic and apical actin protrusion distribution defect of *dmim*. Interestingly, *Rac1*[DN] expression led to a loss of DMIM antibody fluorescence at the cortex (Fig. 8, H and I).

We further tested apical actin protrusion distribution in embryos depleted of the bundled actin nucleator Dia. We reduced the expression of Dia using an RNAi (*dia*[i]) and increased its expression by overexpressing Dia-GFP to study its impact on protrusions. Dia RNAi-expressing embryos showed rounded caps with almost no protrusions, and this defect was more severe than DMIM-mCherry expression. Dia-EGFP overexpression increased protrusions, but its effect was not as strong as that seen with DMIM (Fig. S5 A). To test this further, we overexpressed a truncation of Dia, which is activated due to the removal of the C terminus auto inhibitory domain DiaΔDAD (Homem and Peifer, 2009). Maternally driven expression of DiaΔDAD did not yield embryos. We combined the maternal Gal4 with Gal80 to inhibit the Gal4 expression and obtained embryos which showed a profound increase in the length of actin protrusions on the apical surface (Fig. S5 B). We did not quantify these defects due to very aberrant apical membrane morphology and defects in the division cycles.

Arp3 levels in the *dia*[i] embryos were increased at the cap membrane (Fig. S5, C and D).

FM1-43FX uptake levels showed an increased variation with abundant fluorescent vesicles, but the fluorescence was not significantly increased in *dia*[i] embryos (Fig. S5, E and F). However, Rab5 levels were significantly higher, confirming an increased rate of endocytic trafficking on the depletion of Dia (Fig. S5, G and H). We also expressed DMIM-mCherry together with *dia*[i] to study the localization of DMIM on the depletion Dia and observed that the localization of both DMIM-mCherry was unaffected in *dia*[i] embryos (Fig. S5, I and J; and Video 9). DMIM was also localized to the cortex in DiaΔDAD expressing embryos even though polarity protein Dlg was lost from the plasma membrane and was aberrantly distributed in vesicles (Fig. S5 K). Together, these data showed that the branched actin network activity via Rac and DMIM and the bundled actin network activity via Dia are maintained in a balance to execute apical actin protrusion remodeling and endocytosis.

## DMIM-IBAR domain alleviates the apical actin protrusion remodeling defect in DMIM-depleted embryos, and *Rac1*[DN] suppresses the decrease in protrusions in DMIM-overexpressing embryos

We further tested the interaction between Rac1 and DMIM in the syncytial cycles. We assessed the role of the I-BAR domain of DMIM in apical actin protrusion remodeling in the syncytial cycles. The I-BAR domain of DMIM has been shown to be important for dimerization and interaction with Rac-GTP and F-actin (Cao et al., 2012; Bompard et al., 2005). We made combinations of DMIM-IBAR-GFP and *mim*[i] to test the impact of the I-BAR domain on apical actin protrusion remodeling defects of DMIM. Since the MIM shRNA construct was against the central domain and led to depletion of endogenous DMIM, it did not affect the DMIM-IBAR-GFP expression (Fig. 1 B and Fig. S6 A). We added mitochondrial matrix targeted-GFP (mito-GFP) to control for the effects of Gal4 dilution on apical actin protrusion length and numbers. In embryos expressing *mim*[i]; UAS-mito-GFP, apical actin protrusion length and number were increased as compared with controls (Fig. 9, A, B, E, and F), similar to that of *mim*[i] embryos (Fig. 1). However, this defect was alleviated in embryos expressing DMIM-IBAR-GFP and *mim*[i] together, and the apical actin protrusion length and numbers were reduced significantly in these embryos as compared with *mim*[i] alone (Fig. 9, A–F). This showed that the DMIM I-BAR domain was important for apical actin protrusion remodeling. The levels of Arp3 were also found to be significantly increased in DMIM-IBAR-GFP with *mim*[i] embryos compared with mito-GFP with *mim*[i] embryos and were comparable with controls (Fig. S6, A and B). This rescue is consistent with the known role of the MIM I-BAR domain in recruiting Rac-GTP, which in turn allowed the formation of the branched actin network.

We also expressed *Rac1*[DN] and DMIM-mCherry together during oogenesis and embryogenesis to test for rescue of the DMIM overexpression phenotype as compared with DMIM-mCherry alone by crossing them to maternal Gal4 lines. Since the *Rac1*[DN] and DMIM-mCherry combination resulted in two transgenes being driven by the same Gal4, we controlled for potential Gal4 dilution effects on the actin-protrusion remodeling in these crosses by adding GFP and comparing *Rac1*[DN]; DMIM-mCherry to *Rac1*[DN]; UAS-GFP and DMIM-mCherry; UAS-GFP. *Rac1*[DN]; UAS-GFP showed abundant and longer actin protrusions as compared with controls (Fig. 9, G, I, and J) and similar to *Rac1*[DN] alone (Fig. 8, A and C–E). DMIM-mCherry;

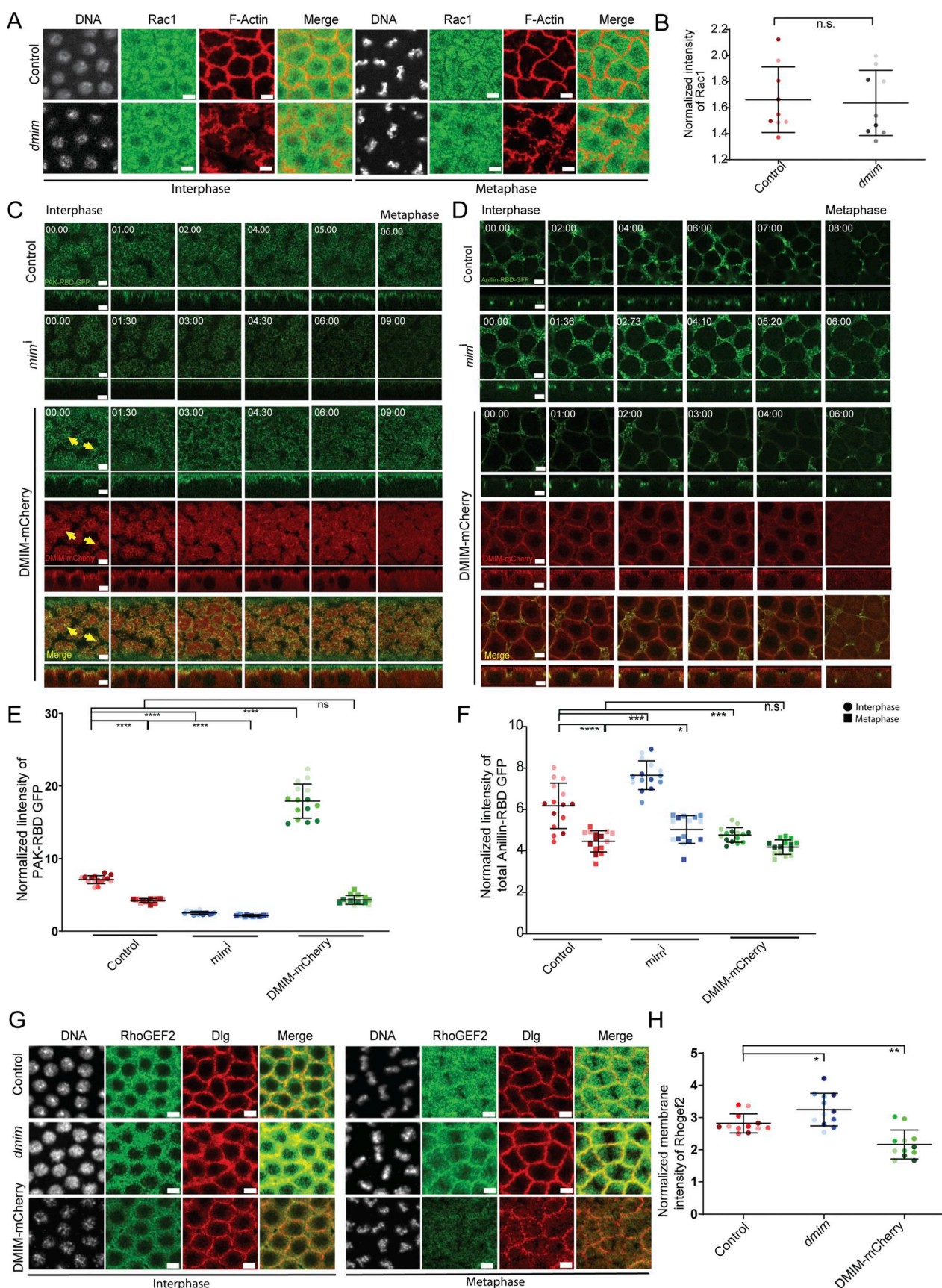

Figure 7. **Rac-GTP decreases, and Rho-GTP along with RhoGEF2 increases in DMIM-depleted embryos. (A)** Representative grazing section images showing nuclei labelled with Hoechst (gray), Rac1 (green), and cortical F-actin labelled with Alexa Fluor 568 (red) in control and *dmim* embryos. **(B)** Graph

showing the quantification of normalized apical intensity of Rac1 in control and *dmim* embryos. *n* = 3 cells from 3 embryos for each genotype. **(C)** Representative images of the grazing and sagittal sections of the syncytial embryos in cycle 12, from interphase till metaphase, expressing Rac-GTP sensor PAK3-RBD-GFP in control, *mim*[i], and DMIM-mCherry (yellow arrows point to DMIM-mCherry and PAK3-RBD-GFP enriched punctae). **(D)** Representative images of the grazing and sagittal sections of the syncytial embryos in cycle 12, from interphase till metaphase, expressing Rho-GTP sensor Anillin-RBD-GFP in control, *mim*[i], and DMIM-mCherry. **(E)** Graph showing the quantification of normalized intensity of PAK3-RBD-GFP in control, *mim*[i], and DMIM-mCherry embryos at interphase and metaphase. *n* = 5 cells from 3 embryos for each genotype. **(F)** Graph showing the quantification of normalized intensity of total Anillin-RBD-GFP in control, *mim*[i], and DMIM-mCherry embryos at interphase and metaphase. *n* = 5 cells from 3 embryos for each genotype. **(G)** Representative grazing section images showing nuclei labelled with Hoechst (gray), Rhogef2 (green), and Dlg (red) in control, *dmim*, and DMIM-mCherry embryos. **(H)** Graph showing the quantification of normalized intensity of PAK3-RBD-GFP in control, *dmim*, and DMIM-mCherry embryos. The plots are represented as super plots with different shades of the same color to distinguish different embryos of the same genotype. Data are represented as mean ± SD; *P < 0.05, **P < 0.01, ***P < 0.001, and ****P < 0.0001, n.s. not significant; Mann–Whitney test. Scale bar: 5 µm.

UAS-GFP showed shorter actin protrusions as compared with controls (Fig. 9, H–J) and similar to DMIM-mCherry alone (Fig. 1, F, H, and I), while the number of actin protrusions remained similar to controls (Fig. 9, H–J), unlike that of being reduced as observed in DMIM-mCherry alone (Fig. 1, F, H, and I). In embryos expressing DMIM-mCherry and *Rac1*[DN] together, the length of actin protrusions was similar to *Rac1*[DN]; UAS-GFP in interphase and showed a slight but significant decrease in metaphase (Fig. 9, G, I, and J). The average actin protrusion number per cap in *Rac1*[DN]; DMIM-mCherry–expressing embryos was similar to *Rac1*[DN]; UAS-GFP (Fig. 9, I and K). The phenotype of shorter protrusions as observed in embryos expressing DMIM-mCherry; UAS-GFP was not seen in embryos expressing both DMIM-mCherry and *Rac1*[DN] (Fig. 9, H and K). This shows that overexpressing MIM does not alleviate the defects of *Rac1*[DN] and that the presence of both activated Rac and MIM is crucial for regulating branched actin networks and actin protrusions remodeling in the syncytial cycle.

We assessed the localization of DMIM-mCherry in embryos expressing *Rac1*[DN]. *Rac1*[DN]; DMIM-mCherry–expressing embryos showed punctate distribution of DMIM-mCherry in the cytoplasm and at the cap periphery, whereas DMIM-mCherry alone was enriched at the membrane and cortex in interphase and prophase (Fig. 9, L and M; and Video 10). DMIM antibody fluorescence was absent at the cortex in *Rac1*[DN] alone (Fig. 8 H). DMIM-mCherry mislocalization on overexpression with *Rac1*[DN] further confirms the dependence of MIM localization on active Rac (Fig. 9, L and M; and Video 10). In summary, DMIM requires Rac1 activity for its recruitment to the syncytial plasma membrane cortex.

We find that DMIM regulates endocytosis and recruitment of Arp3, SCAR/WAVE, and cortactin. This results in assembly of branched actin, enabling apical actin protrusion remodeling by limiting assembly of bundled actin in syncytial *Drosophila* embryos.

## Discussion

We find a role for the I-BAR domain protein MIM in regulating branched actin network formation and endocytosis during apical actin protrusion remodeling in the rapid syncytial division cycles in *Drosophila* embryogenesis. DMIM is recruited in a precise spatiotemporal manner at the apical cap membrane during interphase to prophase of each syncytial nuclear cycle to enable recruitment of Rac-GTP and Arp2/3 complex, thereby promoting the assembly of branched actin network formation and endocytosis, while limiting RhoGEF2-, Rho-GTP, and Dia-mediated bundled actin assembly (Fig. 10). Here, we discuss our findings in the context of the following: (1) The role of MIM in regulating a transition between Arp2/3 complex–mediated branched actin network and Dia-mediated bundled actin network in apical actin protrusion remodeling, (2) the role of MIM, an I-BAR domain–containing protein in promoting endocytosis in the syncytial division cycles, and (3) the relationship between actin remodeling and endocytosis at the plasma membrane.

Bundled F-actin structures have been shown to be present in the cap supported by Dia in early syncytial cycles in *Drosophila* (Jiang and Harris, 2019). In our study, we also observed the apical actin protrusions formed in syncytial nuclear cycles contain bundled actin filaments and the formin Dia, while the Arp2/3 complex is present at the cortex. The actin protrusions reduce over time as Dia gets relocated to the furrows, while the Arp2/3 complex is retained on the plasma membrane and possibly limits the number and length of the protrusions. DMIM promotes the Arp2/3 network in syncytial *Drosophila* embryos through interacting with active Rac1 and promoting the WRC. WRC is recruited by negatively curved membranes, and it is possible that MIM recruits WRC by inducing membrane curvature (Wu et al., 2024, Preprint).

There are several potential mechanisms by which the balance between the branched and bundled actin might be perturbed in the absence of DMIM. DMIM-mediated opposing activation of Rac and Rho leads to the actin cytoskeleton remodeling during morphogenesis. Several studies show that the activation of one of the pathways downregulates the other (Worthylake et al., 2001; Vega and Ridley, 2008; El-Sibai et al., 2008). We observed that decreased Rac1 activity led to an increase in apical actin protrusions in interphase and metaphase, similar to the DMIM mutant embryos. DMIM was mislocalized in *Rac1*[DN]-expressing embryos. DMIM supports Rac-GTP enrichment in the apical cap, thereby facilitating Rac1 signalling to its effectors. Together, it is possible that DMIM promotes the formation of a scaffold with activated Rac1 to enable the formation of the branched actin network. DMIM depletion might lead to decreased functioning of Rac1 and promote an upregulation of RhoGEF2 recruitment at the furrow membrane, thereby increasing Rho1-dependent pathway and increasing the prevalence of actin protrusions through the activity of the downstream effector, formins. The mouse homolog of DMIM has been shown to directly inhibit the formin DAAM in dendritic

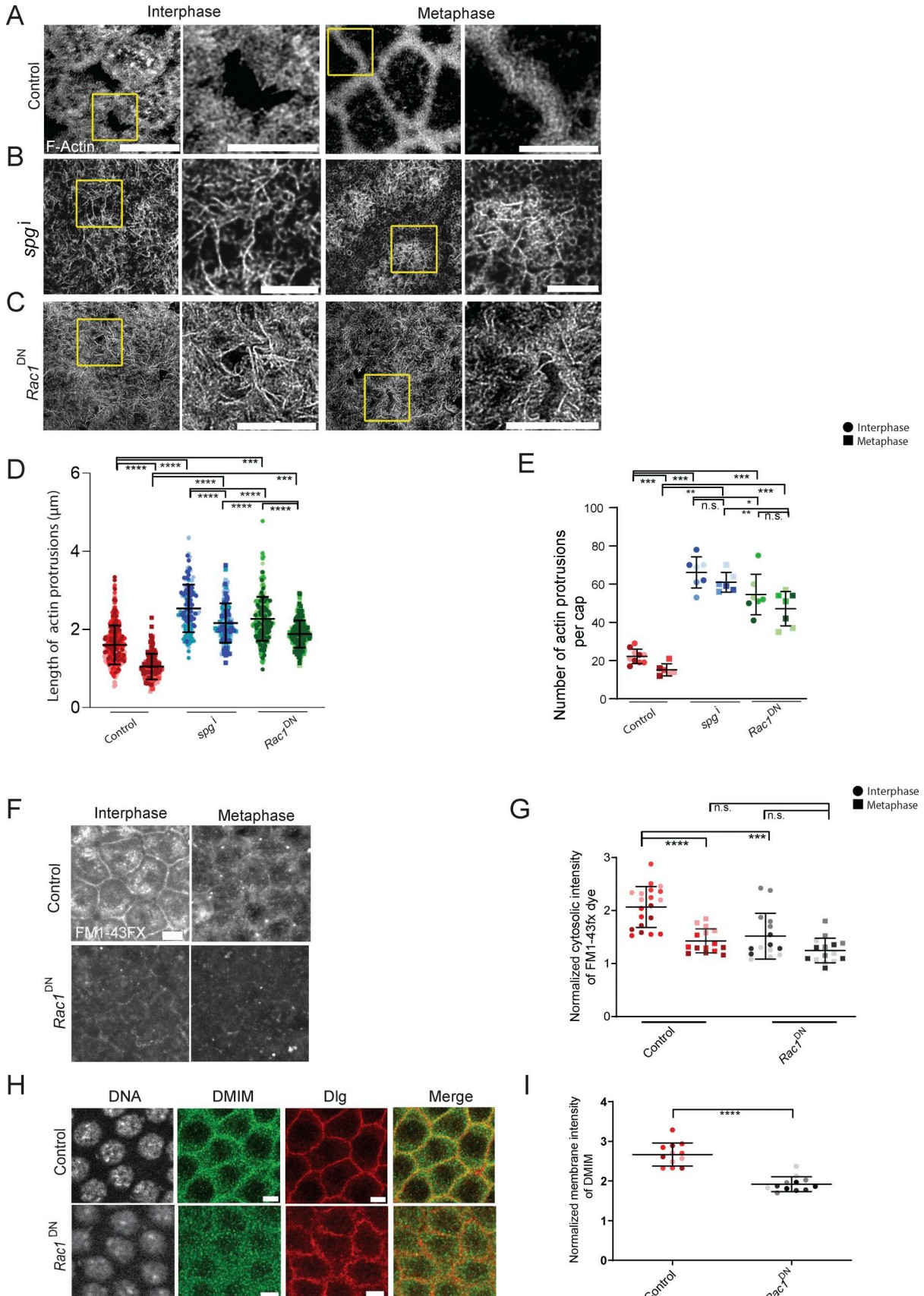

Figure 8. **Active Rac1 regulates actin protrusions remodeling and endocytosis. (A–C)** STED images showing the F-actin–containing protrusions at interphase and metaphase of nuclear cycle 12 for control (A) *spg*i (B) and *Rac1*DN (C) embryos. The region in the yellow box is further zoomed (2.5×) in to show the

actin protrusions. **(D and E)** Graphs showing the apical actin protrusion length (D) and numbers per apical cap (E) at interphase and metaphase; *n* = 2–3 cells from 3 embryos for each genotype, 50–200 protrusions from each embryo, and 3 embryos for each genotype. **(F)** Representative grazing section images labelled with FM1-43FX dye at interphase and metaphase in control and *Rac1*^DN^ embryos. **(G)** Graphs showing the quantification of the normalized cytosolic intensity of the FM1-43FX dye at interphase and metaphase in control and *Rac1*^DN^ embryos. *n* = 5 cells from 3 embryos for each genotype. **(H)** Representative grazing section images showing nuclei labelled with Hoechst (gray), DMIM (green), and cortical F-actin labelled with Alexa Fluor Phalloidin 568 (red) in control and *Rac1*^DN^ embryos. **(I)** Graphs showing the quantification of the normalized membrane intensity of DMIM in control and *Rac1*^DN^ embryos. *n* = 3 cells from 4 embryos for each genotype. The plots are represented as super plots with different shades of the same color to distinguish different embryos of the same genotype. Data are represented as mean ± SD; *P < 0.05, **P < 0.01, ***P < 0.001, and ****P < 0.0001, n.s. not significant; Mann–Whitney test. Time is in min: sec. Scale Bar: 5 µm.

protrusions (Kawabata Galbraith et al., 2018). In addition, our observations of DMIM loss giving rise to an increase in the formin Dia-supported protrusions in *Drosophila* syncytial cycles are consistent with increase in DAAM-supported dendritic protrusions. Further, depletion of *Drosophila* PAK1 protein kinase, known to be activated by Rac1, leads to oogenesis defects, which are mediated by increased Rho1 activation (Vlachos and Harden, 2011). Thus, a depletion of DMIM might lead to an upregulation of the Dia activity, leading to excess actin protrusion formation.

It has been previously observed that in the syncytial *Drosophila* embryos, Dia and Arp2/3 complex interact mechanistically. An Arp2/3 complex–mediated branched actin network pushes the bundled actin filaments promoted by Par-1 and Dia to the periphery of the apical caps. A loss of the branched actin network leads to an increase in the prevalence of the bundled actin filaments at the center of the cap (Jiang and Harris, 2019). Due to the loss of the Arp2/3-derived network in DMIM-depleted embryos, a lack of repositioning of the actin bundles to the periphery and the furrows could lead to an accumulation of the excess actin protrusions in the apical cap. The two nucleators, Arp2/3 complex and Dia, also compete with each other for the availability of the free pool of G-actin in the cells for filament assembly in this system (Xie et al., 2021). Studies in yeast have shown that competition for G-actin is crucial to regulating appropriate F-actin network size. Inhibition of the Arp2/3 complex depletes Arp2/3-mediated endocytic actin patches and induces an excess formation of formin-assembled F-actin and vice versa (Burke et al., 2014). Therefore, in the absence of the Arp2/3 network on DMIM depletion, Dia can nucleate more bundled actin filaments due to increased availability of G-actin in the system and lead to more actin bundles and protrusions.

MIM may function through an interaction with other BAR domain proteins, such as an F-BAR protein, Cip4. Cip4 positively regulates the Arp2/3 complex and inhibits Dia to promote receptor-mediated endocytosis (Fricke et al., 2009; Yan et al., 2013). DMIM-overexpressing embryos resulted in a decrease in Dia at the cortex and a decrease in actin protrusions similar to Dia depletion. A synergistic interaction of the two BAR domain–containing proteins could be responsible for regulating the apical actin protrusion remodeling. In summary, DMIM potentially acts as a scaffold for the small GTPase molecule Rac1, and together, they activate the WRC and Arp2/3 complex. This leads to a downregulation of the Rho1-dependent assembly of Dia-nucleated bundled actin in the apical region to facilitate reduction of the actin protrusions.

BAR domain proteins are positive regulators of endocytosis through interaction with the actin cytoskeleton during metazoan development (Sherlekar and Rikhy, 2016; Zhao et al., 2019; Leibfried et al., 2008; Sathe et al., 2018). MIM is an I-BAR domain–containing protein and is predicted to induce negative curvature. In doing so, it supports actin protrusion formation and inhibits endocytosis (Quinones et al., 2010). MIM has also been shown to promote endocytosis due to its role in binding to negative curvatures required during vesicle formation (Li et al., 2017; Li et al., 2019; Zhao et al., 2016; Zhao et al., 2019; Yu et al., 2011; Sathe et al., 2018). Although endocytosis requires a largely positive plasma membrane curvature like endocytic pits to be formed, an invagination contains negative curvature at the neck of a vesicle where the I-BAR domain–containing proteins can bind (Jones et al., 2020; Hurley and Wendland, 2002). MIM by binding to negative curvature also plays a role in fission in filopodia in the formation of extracellular vesicles in human cells (Nishimura et al., 2021; Fujioka et al., 2025). These contrasting results may also arise due to the context of MIM function. MIM function has been previously shown to vary across different tissues, sometimes also depending on the density of cells in a tissue (Dawson et al., 2012). We observed a depletion of endocytosis of the FM1-43FX dye and reduction in Rab5 and Rab11 vesicles in DMIM-depleted embryos. DMIM localizes on the plasma membrane only till prometaphase, possibly till the time point remodeling of actin protrusions occurs through tubular endocytosis, a mechanism also seen in cellularization (Fabrowski et al., 2013). We found that Rab5 levels decrease from interphase to metaphase, and a similar trend is observed with the localization of dynamin and clathrin on the plasma membrane during the syncytial cycles (Rikhy et al., 2015). We also observed an accumulation of DE-cad in DMIM-depleted embryos. These data support a dynamic regulation of endocytosis at the plasma membrane in the syncytial cycles. Future analysis on the recruitment of DMIM on endocytic pits will be needed to confirm its role in directly mediating endocytic vesicle formation.

We observed that in mutant embryos, where the Arp2/3 complex recruitment was reduced, i.e., DMIM-depleted embryos or embryos expressing *Rac1*^DN^, endocytosis was also reduced. This is concurrent with the increased number and density of actin protrusions and a lack of furrow formation. DMIM depletion led to a loss of Arp3 and its activators SCAR/WAVE and cortactin. Branched actin networks are linked with the regulation of endocytic machinery in a wide range of organisms. The Arp2/3 NPF WAVE, also known as SCAR, regulates vesicle movements in *Drosophila* S2R+ cells (Fricke et al., 2009) and recycling of E-cadherin at the plasma membrane in mammalian tissue culture cells (Silva et al., 2009). In B cells, Arp2/3

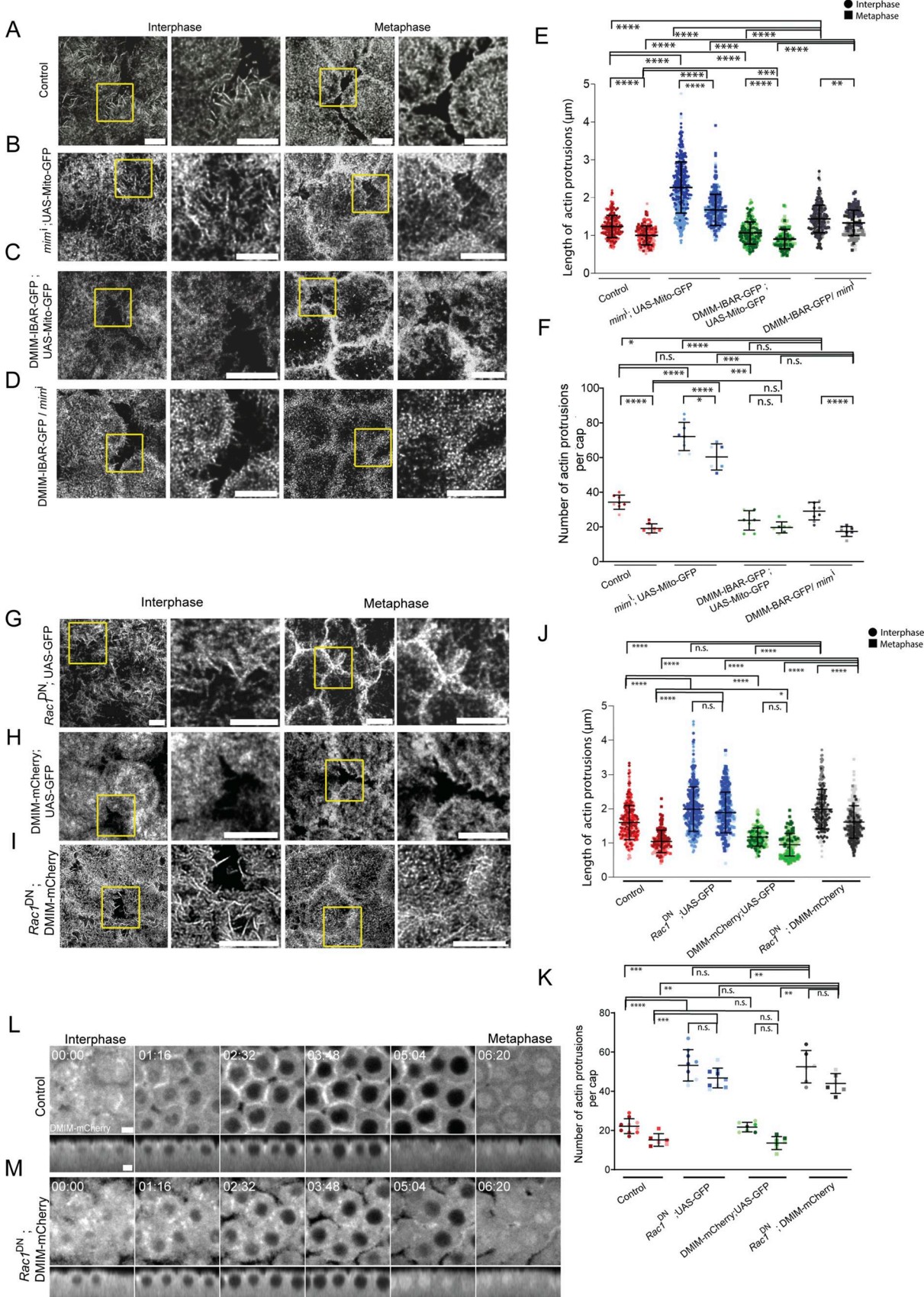

Figure 9. **DMIM I-BAR domain rescues apical actin protrusion remodeling defect in DMIM-depleted embryos, whereas Rac1[DN] partially suppresses the decrease in apical actin protrusion in MIM-overexpressing embryos. (A–D)** STED images showing the F-actin–containing protrusions at interphase and

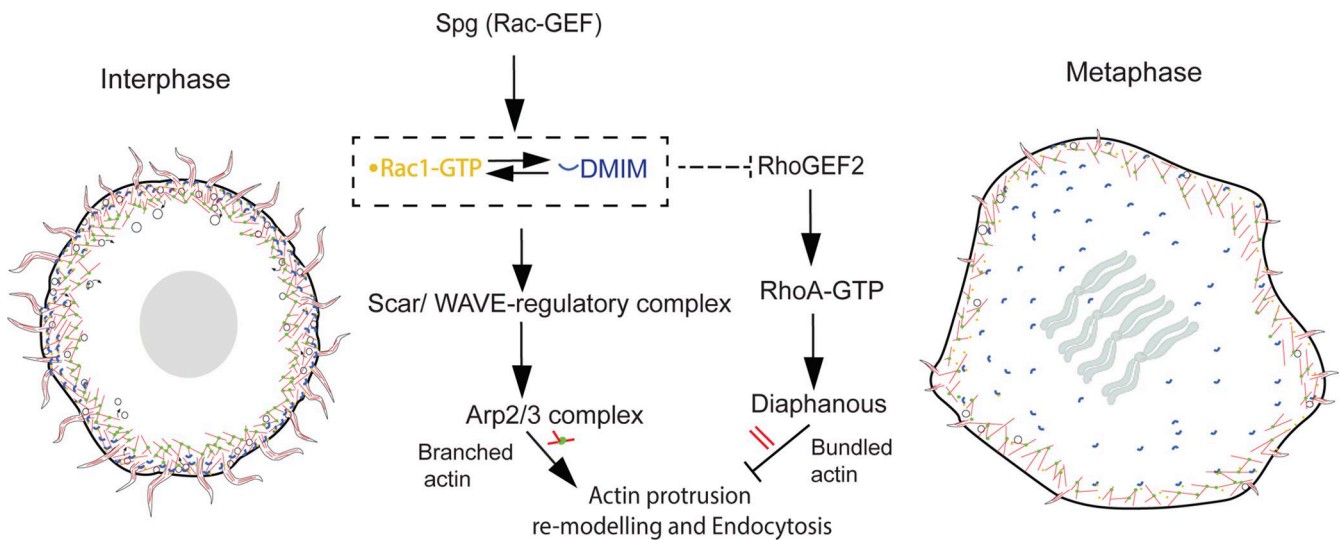

metaphase of nuclear cycle 12 for control (A), *mim*[i]; UAS-mito-GFP (B), DMIM-IBAR-GFP; UAS-mito-GFP (C), and *mim*[i]; DMIM-IBAR-GFP (D). The region in the yellow box is further zoomed (2.5×) in to show the actin protrusions. **(E and F)** Graphs showing the apical actin protrusion length (E) and numbers per apical cap (F) at interphase and metaphase; $n$ = 2–3 cells from 3 embryos for each genotype, 50–200 protrusions from each embryo, and 3 embryos for each genotype. **(G–I)** STED images showing the F-actin–containing protrusions at interphase and metaphase of nuclear cycle 12 for *Rac1*[DN]; UAS-GFP (G), DMIM-mCherry; UAS-GFP (H), and *Rac1*[DN]; DMIM-mCherry (I) embryos. The region in the yellow box is further zoomed (2.5×) in to show the actin protrusions. **(J and K)** Graphs showing the apical actin protrusion length (J) and numbers per apical cap (K) at interphase and metaphase; $n$ = 2–3 cells from 3 embryos for each genotype, 50–200 protrusions from each embryo, and 3 embryos for each genotype. **(L and M)** Representative images of the grazing and sagittal sections of the syncytial embryos in cycle 12 in DMIM-mCherry (L) and *Rac1*[DN]; DMIM-mCherry (M) embryos from interphase to metaphase. The plots are represented as super plots with different shades of the same color to distinguish different embryos of the same genotype. Data are represented as mean ± SD; *P < 0.05, **P < 0.01, ***P < 0.001, and ****P < 0.0001, n.s. not significant; Mann–Whitney test. Scale bar: 5 µm.

complex–dependent actin polymerization generates actin foci interspersed with linear bundles of actin and myosin II, which lead to actin branching and promote endocytosis (Roper et al., 2019). The branched actin network is required for the clathrin-dependent and clathrin-independent endocytosis. This actin remodeling enables the stepwise pulling, sculpting, and pushing of the plasma membrane, which leads to the invagination, maturation, and scission of the endocytic vesicles. Further, WRC and WASP preferentially bind to negative curvature at the neck of the endocytic pit (Pipathsouk et al., 2021; Brunetti et al., 2022). In mammalian immune cells, N-WASP and the Arp2/3 complex are recruited asymmetrically to one side of clathrin endocytic sites to stimulate actin assembly and force production for antigen internalization (Jin et al., 2022).

Trafficking at the plasma membrane can also be altered significantly due to the change in cortical tension. Clathrin-independent or CLIC/GEEC (CG) pathway endocytosis is upregulated upon tension reduction. Inhibition or activation of the CG pathway results in lower or higher membrane tension, respectively, via mechanochemical feedback inhibition in adherent cells (Thottacherry et al., 2018). During syncytial cycles of *Drosophila* embryogenesis, there is an increase in the junctional tension as the cells transition from interphase to metaphase with the polygonal organization of the plasma membrane (Dey and Rikhy, 2020). DMIM and Arp2/3 complex together regulate branched actin network formation and endocytosis in interphase. With endocytosis and depletion of actin protrusions, the cell boundary is likely to become taut, and tension increases, leading to inhibition of membrane trafficking. MIM could potentially facilitate endocytic trafficking by harnessing the branched actin network at low tension. Our work motivates future studies that aim to understand the relationship between cortical tension, BAR domain protein recruitment, actin network remodeling, and membrane trafficking.

## Materials and methods

### Fly husbandry and stocks

All *Drosophila* stocks are maintained at 25°C in the standard corn meal agar medium. The crosses are maintained at 28°C and/or 25°C. The fly stocks tGPH (RRID:BDSC_8163), *nanos*-Gal4 (lab stock), *mat-α-tub*-Gal4 67, Rac1.N17 (RRID:BDSC_6292), *dia*[i] (RRID:BDSC_33424), Dia-EGFP (RRID:BDSC_56751), DiaΔDAD: w*; P{UASp-dia.ΔDad.EGFP}3 (RRID:BDSC_56752);

**Figure 10.** **Schematic model showing the role of DMIM in apical actin protrusion remodeling in the syncytial division cycles of *Drosophila* embryos.** MIM regulates endocytic trafficking by promoting a branched cortical actin network with Arp2/3 complex, SCAR/WAVE, and cortactin to facilitate apical actin protrusion remodeling in interphase of the syncytial division cycle. Rac1 and MIM stabilize each other and, in turn, recruit SCAR/WAVE, cortactin, and Arp2/3 complex for the formation of the branched actin network. The activation of the branched actin network by Rac, MIM, and Arp2/3 complex restricts the formation of actin protrusions by activation of RhoGEF2 and bundled actin network supported by Dia.

*spg*[i] (RRID:BDSC_34367), w[1118]; Df(2R)BSC261/CyO (RRID: BDSC_23161), *mim*[i] (RRID:BDSC_43223), UASp-mito-GFP (Rachel Cox, USUHS, Bethesda, MD, USA), and UAS-Valium-GFP (RRID:BDSC_35786) were obtained from Bloomington *Drosophila* Stock Center, Indiana, USA (BDSC). *mat-α-tub*-Gal4 67; *mat-α-tub*-Gal4 15 was obtained from the Girish Ratnaparkhi lab, IISER, Pune, India. tGPH::*mat-α-tub*-Gal4 67; *mat-α-tub*-Gal4 15 was made with standard genetic crosses. The deletion mutant *dmim* (Quinones et al., 2010) was used for the study. The fly stocks sqh-Sqh-mCherry, *mat-α-tub*-Gal4 67; *ubi*-DE-Cad-GFP, and *mat-α-tub*-Gal4 15/TM3Sb were obtained from Adam Martin lab, MIT, MA, USA, and the wild-type Canton S strain was obtained from the *Drosophila* facility at IISER, Pune, India. The Anillin-RBD-GFP stock was obtained from Thomas Lecuit, IBDM, France. The PAK3-RBD-GFP was obtained from Tony Harris, University of Toronto, Canada. *nanos*-Gal4 and *mat-α-tub*-Gal4 67, when crossed to DiaΔDAD, did not give embryos. A combination of *nanos*-Gal4 with Gal80 and DiaΔDAD at 25°C was used to obtain embryos.

### Testing for the impact of Gal4 dilution
For testing, the effect of dilution of *nanos*-Gal4 in the DMIM-mCherry; Rac1[DN] combination as compared with DMIM-mCherry and Rac1[DN], DMIM-mCherry and Rac1[DN] transgenes were crossed to UAS-GFP. For testing the effect of dilution of *nanos*-Gal4 in the DMIM-IBAR-GFP; *mim*[i] compared with DMIM-IBAR-GFP and *mim*[i], DMIM-IBAR-GFP and *mim*[i] were individually crossed to mito-GFP.

### Embryo lethality
Control and *dmim* flies were kept in cages at 25°C with sucrose agar plates with yeast paste. They were acclimatized for at least 2 days. Embryos were collected for 2 h and washed with water and arranged in a grid of 100 embryos. The *dmim* mutant embryos showed partial lethality, and 30% of the embryos did not hatch at 24 h after egg laying at 25°C.

### Generation of DMIM-mCherry and domain deletion transgenes
The total RNA was purified from 10 Oregon R (wild-type flies), and the first cDNA strand was synthesized from the RNA. From the cDNA, dMIM splice variants were amplified via PCR. The forward and reverse primers were designed to amplify each end of the dMIM coding sequence. The primers 5′-CTG-CGG-CCG-CGA-TGG-ATC-TAA-GTC-TGG-AAC-G-3′ and 5′- CGT-CTA-GAC-TAA-TGT-ATC-TTG-GGA-GCG-CTG-3′ bound to the first and last coding exons, respectively. The purified PCR products were TA cloned into the pGEM-T vector.

The truncated forms of DMIM were generated following the same procedures using the full-length DMIM splice variant as template. The forward and reverse primer sequences used for generating the truncated form of DMIM lacking the WH2 domain only (exon 17) (DMIM-IBAR-CD) were 5′-GCG-TCT-AGA-CTA-GCG-AGG-GTC-CGG-CCC-3′ and 5′-CGT-CTA-GAC-TAA-TGT-ATC-TTG-GGA-GCG-CTG-3′, respectively, whereas the forward and reverse primer sequences used for generating the truncated forms of dMIM lacking the central scaffold and C-terminal WH2 domain, comprising only the N-terminal

I-BAR domain (exons 2–6)(DMIM-IBAR), were 5′-GCG-TCT-AGA-CTA-GCT-GGC-CTT-AGC-GTC-ATG-3′ and 5′-CGT-CTA-GAC-TAA-TGT-ATC-TTG-GGA-GCG-CTG-3′, respectively. The truncated form of DMIM lacking the N-terminal I-BAR domain (exons 2–6) (DMIM-CD-WH2) was generated using the forward and reverse primers 5′-CGT-CTA-GAC-TAA-TGT-ATC-TTG-GGA-GCG-CTG-3′ and 5′- CTG-CGG-CCG-CGG-CCA-GCA-TTA-ATC-TGT-AC-3′, respectively. Each of the splice variants was subcloned into a variant of the pUASP expression vector containing the coding sequence of eGFP (for the truncated lines) or mCherry (for the full-length DMIM splice variant). The transgenes are inserted into Not1 and Xba1 restriction sites, in frame with the eGFP or mCherry, to generate fusion proteins. The successfully subcloned variants were sent to BestGene Inc., USA, for injection into *Drosophila* embryos and injected into early embryos prior to blastoderm cellularization, for it to enter the pole cell nuclei and becomes incorporated into chromosomes of germline cells by the P element transposase, which is encoded by a plasmid injected alongside the transgenic DNA. In the subsequent offspring, the inheritance of the transgene was recognized via red eye color, and successful transgenic lines were amplified and, following genetic mapping of the chromosome in which they are located, maintained over balancer stocks (Ashburner and Bergman, 2005).

### DMIM antibody generation
Antibody generation for DMIM was carried out by Boster Bio, USA. The cDNA encoding a peptide of 165 amino acids in the I-BAR domain was cloned in pColdI and expressed in *Escherichia coli*. The sequence of the protein domain is as follows: 5′-RHKAVETRLKTFTSTIMDCLVQPLQERIEDWKRAT VTIDKDHAKEYKRCRSELKKRSSDTLRLQKKARKGQTDGLQSL MDSHMQDVTLRRAELEEVEKKSLRAAMVEERLRYCSFVHML QPVVHEECEVMSELGHLQEAMQSIALVTKEPSVLPQASEELI HDAK-3′. After expression, the protein was purified and injected in rabbits four times to obtain the antibody. The titer was determined by ELISA. The antibody was used 1:500 dilution against controls and mutants.

### Embryo immunostaining
The flies are put in a cage for embryo collection. 2.5–3-h-old embryos are collected in plates containing 3% sucrose, 2.5% agar, and a blob of yeast paste. Embryos are dechorionated by incubating in 100% bleach for 1 min and then fixed using 1:1 heptane and 4% paraformaldehyde (PFA) in 1× PBS (1 mM EGTA was added to this for DE-Cad staining) for 25 min. This is followed by hand de-vitellinization of the fixed embryos in PBS or devitellinization by vigorous shaking in a 1:1 mixture of heptane and chilled methanol (for Sqh staining). De-vitellinized embryos are washed three times in 1× PBS and 0.3% Triton X-100 (1× PBST). As a blocking agent, 2% bovine serum albumin (BSA) is added for 1 h. Embryos are then incubated overnight in the primary antibody of required dilution at 4°C and washed again with 1× PBST. We used the following primary antibodies: anti-Arp3 (1: 500, raised in rabbit, William Theurkauf, University of Massachusetts Medical School, Worcester, MA, USA), anti-Dia (1:1,000, raised in rabbit, Wasserman lab, University of

California San Diego, La Jolla, CA, USA), anti-DCAD2 (1:1,000, RRID:AB_528120, DSHB), anti-Cortactin (1:100, raised in mouse, #610049; BD Biosciences), anti-SCAR (1:100, raised in mouse, DSHB), anti-Sqh (1:1,000, raised in rabbit, Adam C. Martin, Massachusetts Institute of Technology, Cambridge, MA, USA), anti-Rab5 and anti-Rab11 (1:500, raised in guinea pig and rabbit, respectively, Akira Nakamura, RIKEN Center for Developmental Biology, Kobe, Japan), anti-Dlg (1:50, 4F3, RRID:AB_528203, DSHB), anti-Rac1 (1:100, #AB_1553767, DSHB), and anti-RhoGEF2 (1:500, Jorg Grosshans, Phillips University of Marburg, Germany). We also used the antibody against *Drosophila* MIM (1: 500, raised in rabbit) generated by Boster Bio for this study. Fluorescently coupled secondary antibodies of the respective animals (1:1,000, anti-mouse, A-1100, RRID:AB_2534069, anti-rabbit, A-11008, RRID:AB_143165, anti-mouse, A-11004, RRID:AB_141371, anti-rabbit, A-1101, RRID:AB_143157) were added and incubated for 45 min–1 h at room temperature, followed by washing with 1× PBST, addition of Hoechst 33258 (1:1,000, #H-3569; Molecular Probes) for DNA staining, and mounting on slides with Slow fade Gold antifade (Molecular Probes). F-actin staining was carried out by fluorescently coupled phalloidin (1:100, Phalloidin Alexa 488, A-12379, RRID:AB_2315147, Phalloidin Alexa 568, A-12380, RRID:AB_2759224).

### Endocytic dye uptake assay
Embryos were collected after 1.5 h in the sucrose-agar plate. The embryos were washed with deionized water and passed through the sieve. The embryos were dechorionated with 100 % bleach for 4 min and then permeabilized for 5 min using 1:10 Limonene detergent (CitraSolv) and 1× PBS. This is followed by a wash with 1× PBS. The embryos are then incubated with the amphipathic and fixable FM1-43FX dye (fixable analog of FM 1–43 membrane stain, catalog number: F35355; Thermo Fischer Scientific) diluted in 1× PBS for 15 min. The embryos are then washed in 1× PBS to wash off nonspecific staining on the plasma membrane. The embryos are then fixed using 1:1 heptane and 4% paraformaldehyde (PFA) in 1× PBS for 25 min. The embryos were then hand de-vitellinized, followed by addition of Hoechst 33258 (1:1,000, Molecular Probes) for DNA staining, and mounted on slides with Slow fade Gold antifade (Molecular Probes) and imaged using a confocal microscope. FM1-43FX dye is excited at 490–510 nm and initially emits at around 626–636 nm and in the 565–590 nm range upon membrane integration.

### Sample preparation for live imaging
Embryos are collected after 1 h in the sucrose-agar plate and dechorionated with 100% bleach. Embryos are then washed with water, dried on a tissue, and carefully mounted on coverslip 2 chamber 1 number coverslip bottom Lab-Tek chambers and immersed in 1× PBS.

### Microscopy
Leica laser scanning microscope SP8 and Zeiss laser scanning confocal microscope LSM710, LSM780 were used to image immunostained fixed or live embryos. The laser lines 488, 561, and 633 nm and a 40× oil objective having NA 1.4 of these microscopes were used. For both live and fixed imaging, a line

averaging of 2 was used. Images were captured on an 8-bit scale with 512 × 512 format.

### Super resolution imaging on STED (Mitra et al., 2022)
Embryos were collected for 1.5–2 h at 25°C and were labelled with phospho-histone H3 (1:600, catalog no.: MA5-15220; Invitrogen), which is used to label the nucleus and identify different syncytial nuclear cycles. This was coupled with anti-mouse Alexa Fluor 488 (1:1,000, Life technologies). F-actin was labelled with Alexa Fluor 488 Phalloidin (1:100, A-12379; Life technologies, RRID:AB_2315147). The samples were embedded in Mowiol supplemented with 2.5% DABCO to reduce the fading of the fluorescence. The embryos were then imaged within 1–2 days with an inverted TCS SP8 3× microscope (Leica 326 Microsystems) with a 100×/1.4 NA oil immersion (refractive index 327 1.518) objective (Leica HC PL APO CS2-STED WHITE) operated by the Leica LAS X 328 software (version 3.5.3). Fluorophores were excited with 488 nm (for Phalloidin) and 633 nm (for phospho-histone H3) derived from 80 MHz pulsed White Light Laser (Leica 330 Microsystems) at 70% intensity, and depletion was performed with a 775 nm pulsed laser (Leica Microsystems) with a maximum power of 820–860 mW. All images were acquired in 2D STED mode. The Fluorophore emission was collected with Hybrid Detectors (Leica Microsystems) and a typical spectral window of 507–542 nm as used for detecting Alexa Fluor Phalloidin 488 and 653–733 nm for phospho-histone H3. Images were captured on an 8-bit scale with 1024 × 1024 TIFF format with a pixel size of about 30–40 nm.

### Image processing, quantification, and analysis
#### Deconvolution of STED images using Huygens professional
The STED images were deconvolved using the STED module on Huygens Professional version 17.04 (Scientific Volume Imaging). The average background was estimated using the in/near object module by controlling the radius parameter. The classical maximum likelihood estimation algorithm was used and the maximum iterations at 27. The signal-to-noise ratio was set at 7.5 and quality threshold at 0.05. The deconvolved images were saved as high-quality TIFF images.

#### Quantification of apical actin protrusion length and number from immunostaining
A region of interest (ROI) was drawn to form an open-ended line, along the apical actin protrusion using the "Segmented line" tool in Image J. The actin protrusions that could be observed in sharp focus emerging from one apical cap were measured. At least 3–5 caps were measured for each embryo and at least 3 embryos for each genotype. The lengths of each ROI were the length measurements for each microvillus, and the number of ROIs per cap was the number of actin protrusions for each cap. The mean ± SD for the apical actin protrusion length and number were computed and plotted with time using GraphPad Prism 8.0.

#### Quantification of apical cell area and furrow length from live imaging
A grazing section of the z-stack that showed the apical-most section of the cell in interphase and metaphase in tGPH-expressing

embryos was chosen. A ROI was drawn to form a closed area, along the boundaries of the apical caps using the "Polygon Selection" tool in ImageJ. 5 caps were measured for each embryo and 3 embryos for each genotype. The mean ± SD for the area were computed and plotted with time using GraphPad Prism 5.0.

An orthogonal plane of the z-stack that showed the lateral furrows at the interphase and metaphase was chosen. A ROI was drawn to form an open-ended line, along the furrows using the segmented line tool in ImageJ. At least 3–5 furrows were measured for each embryo and 3 embryos for each genotype. The mean ± SD for the furrow lengths were computed and plotted with time using GraphPad Prism 8.0.

### Quantification of membrane to cytosolic ratio from immunostaining

A grazing section of the z-stack that showed the subapical section of the cell was chosen. A ROI was drawn on the cell boundary to measure the mean intensity of the fluorophore on the cell membrane. Another ROI was drawn adjacent to the nuclear region of the cell to measure the intensity of the fluorophore in the cytosol. Both were drawn using the segmented line tool in ImageJ. Five cells were measured for each embryo and 3–4 embryos for each genotype. The mean intensity of the membrane ROI was divided by mean intensity of the cytosolic ROI, and the ratio was plotted as "membrane to cytosolic ratio". The mean ± SD for the membrane to cytosolic ratios were computed and plotted with time using GraphPad Prism 8.0.

### Quantification of normalized intensity from live imaging and immunostaining

A grazing section of the z-stack that showed the apical section of the cell was chosen from Anillin-RBD-GFP and PAK-RBD-GFP movies and fixed immunostained embryos. A ROI was drawn to form a closed area, along the boundaries of the apical caps using the "Polygon Selection" tool in ImageJ. Another similar ROI was drawn on a plane above the apical cap of the cell to measure the mean intensity of the background noise of the fluorophore. 5 cells were measured for each embryo and 3–4 embryos for each genotype. The mean intensity of the membrane ROI was divided by mean intensity of the ROI for background noise, and the ratio termed as "normalized intensity" was plotted. The mean ± SD for the normalized apical intensities were computed and plotted with time using GraphPad Prism 8.0.

### Analysis of Pearson's correlation coefficient

Pearson's correlation coefficient for the colocalization of Arp3 and DMIM-mCherry was analyzed using JaCoP plugin and Otsu's threshold selection method in Image J.

### Statistical analysis

For all the quantifications, data were represented as the mean ± SD The $n$ values for each experiment are mentioned in their respective figure legends. The plots are represented as super plots with different shades of the same color to distinguish different embryos of the same genotype. The statistical significance was determined using the Kruskal–Wallis and Mann–Whitney test.

### Online supplemental material

Fig. S1 shows membrane and F-actin staining and analysis of complementation of the *dmim* locus with a genomic deficiency in increasing actin protrusions. DMIM-mCherry localizes to plasma membrane and leads to an overexpression of DMIM. Fig. S2 shows that apical sections of *dmim* mutant embryos show loss of Arp3, cortactin, and SCAR. Fig. S3 shows analysis of lateral Rac-GTP sensor, apical Rho-GTP sensor, and non-muscle myosin II cap periphery accumulation in DMIM-depleted embryos. Fig. S4 shows that active Rac1 recruits Arp3, inhibits Dia, and promotes endocytosis. Fig. S5 shows Dia is necessary for the formation of the actin protrusions. Fig. S6 shows overexpression of the DMIM I-BAR domain rescues loss of Arp3 in DMIM-depleted embryos. Video 1 shows tGPH in control embryos in syncytial cycle 12. Video 2 shows tGPH in *mim*[i] embryos in syncytial cycle 12. Video 3 shows tGPH and DMIM-mCherry in syncytial cycle 12. Video 4 shows DMIM-mCherry in *dmim*; DMIM-mCherry embryos in syncytial cycle 12. Video 5 shows DMIM-CD-WH2-GFP in syncytial nuclear division cycle 12. Video 6 shows DMIM-IBAR-CD-GFP in syncytial nuclear division cycle 12. Video 7 shows DMIM-IBAR-GFP in syncytial nuclear division cycle 12. Video 8 shows DE-Cad-GFP in control and *mim*[i] embryos in syncytial nuclear division cycle 12. Video 9 shows DMIM-mCherry in *dia*[i]; DMIM-mCherry in syncytial nuclear division cycle 12. Video 10 shows DMIM-mCherry in *Rac1*[DN]; DMIM-mCherry in syncytial nuclear division cycle 12.

### Data availability

Data are available in the article itself and its supplementary materials.

## Acknowledgments

We thank the RR lab members for continuous discussions on this project. We thank the FlyBase and BDSC for help with stocks. We thank the IISER Pune microscopy facility and *Drosophila* facilities for help with imaging and stocks and crosses, respectively.

Author contributions: Debasmita Mitra: conceptualization, data curation, formal analysis, investigation, methodology, project administration, software, validation, visualization, and writing—original draft, review, and editing. Georgina K. Goddard: methodology. Sanjana S: investigation, validation, and writing—review and editing. Aparna K: investigation, validation, and writing—review and editing. Tom H. Millard: methodology, project administration, resources, supervision, and writing—review and editing. Richa Rikhy: conceptualization, data curation, formal analysis, funding acquisition, investigation, methodology, project administration, resources, supervision, validation, visualization, and writing—original draft, review, and editing.

Disclosures: The authors declare no competing interests exist.

Submitted: 26 February 2025

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

# Supplemental material

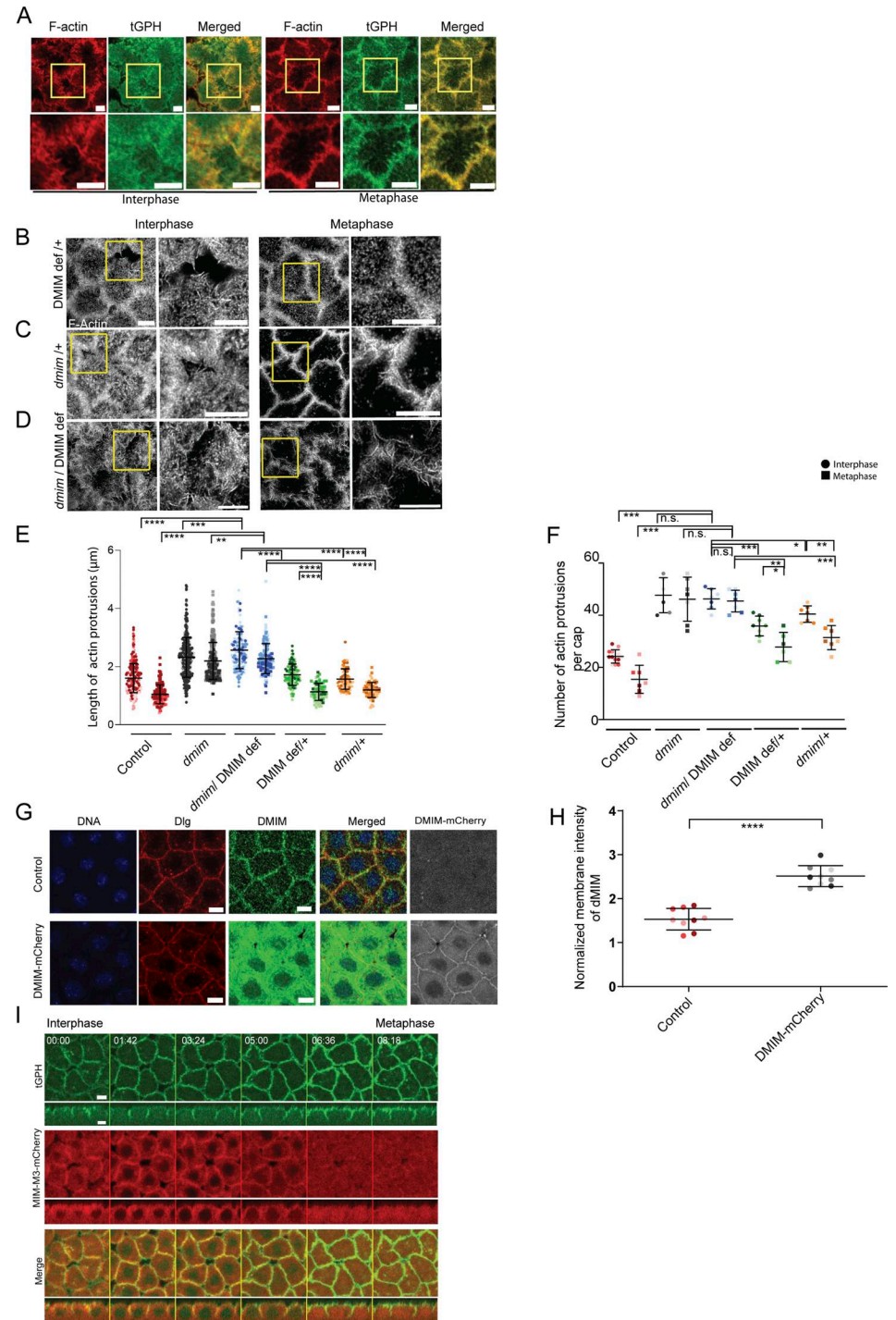

Figure S1. **Membrane and F-actin staining and analysis of complementation of the *dmim* locus with a genomic deficiency in increasing actin protrusions.** DMIM-mCherry localizes to plasma membrane and leads to an overexpression of DMIM. **(A)** tGPH (green)-expressing embryos are stained with phalloidin (red), and representative images are shown in interphase and metaphase. The region in the yellow box is further zoomed in to show the actin protrusions. **(B–D)** STED images showing the F-actin–containing protrusions at interphase and metaphase of nuclear cycle 12 in mim deficiency/+ (B), *dmim*/+ (C), and *dmim*; deficiency (*w*[1118]; Df(2R)BSC261/CyO; BL-23161) (D) embryos. The region in the yellow box is further zoomed (2.5×) in to show the actin protrusions. **(E and F)** Graphs showing the apical actin protrusion length (E) and numbers per apical cap (F) at interphase and metaphase; *n* = 2–3 cells from three embryos for each genotype, 50–200 protrusions from each embryo, and 3 embryos for each genotype. **(G and H)** Representative images of embryos expressing DMIM-mCherry (gray) immunostained with Hoechst (blue), anti-DMIM (green), and Dlg (red) in control, *mim*[i], and *dmim* embryos. **(H)** Graph showing the normalized membrane intensity of anti-DMIM in control and DMIM-mCherry embryos; *n* = 3 cells from 3 embryos for each genotype. **(I)** Representative grazing section images of embryos in syncytial nuclear cycle 12 from interphase to metaphase shows the localization of tGPH (green) and DMIM-mCherry embryos (red). The plots are represented as super plots with different shades of the same color to distinguish different embryos of the same genotype. Data are represented as mean ± SD; *P < 0.05, **P < 0.01, ***P < 0.001, and ****P < 0.0001, n.s. not significant; Mann–Whitney test. Time is in min:sec. Scale bar: 5 µm.

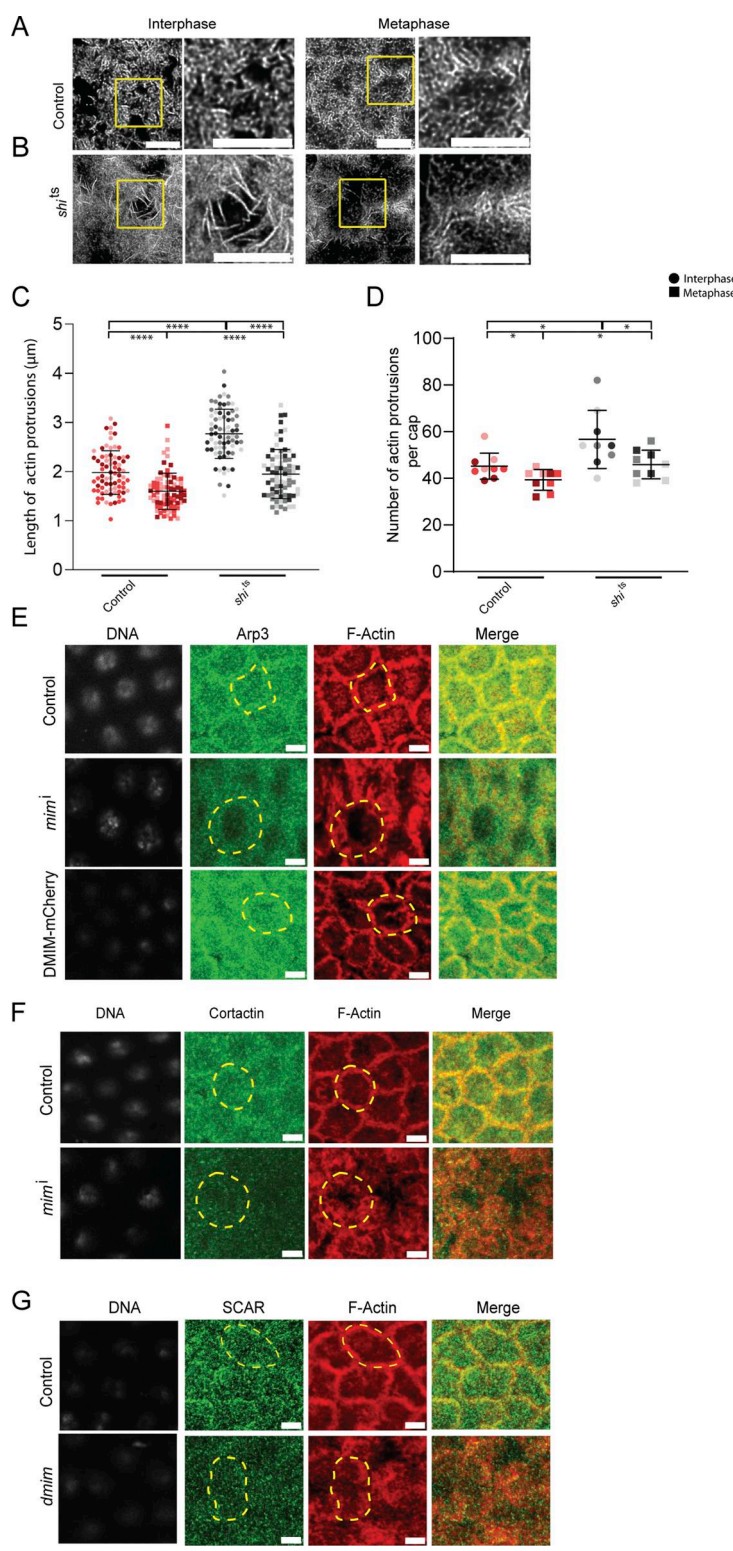

Figure S2. **Apical sections of *dmim* mutant embryos show loss of Arp3, cortactin, and SCAR. (A and B)** STED images showing the F-actin–containing protrusions at interphase and metaphase of nuclear cycle 12 in control (A) and *shi*^ts (B). The region in the yellow box is further zoomed (2.5×) in to show the actin protrusions. **(C and D)** Graphs showing the actin protrusion length (C) and numbers per apical cap (D) at interphase and metaphase; *n* = 2–3 cells from 3 embryos for each genotype, 50 protrusions from each embryo, and 3 embryos for each genotype. **(E)** Representative grazing section images showing apical/axial region of the cells with nuclei labelled using Hoechst (gray), Arp3 (green), and cortical F-actin labelled with Alexa Fluor 568 (red) in control, *mim*^i, and DMIM-mCherry embryos. **(F)** Representative grazing section images showing nuclei labelled with Hoechst (gray), cortactin (green), and cortical F-actin labelled with Alexa Fluor 568 (red) in control and *mim*^i embryos. **(G)** Representative grazing section images showing nuclei labelled with Hoechst (gray), SCAR/WAVE (green), and cortical F-actin labelled with Alexa Fluor 568 (red) in control and *dmim* embryos. Data are represented as mean ± SD, *P < 0.05, ****P < 0.0001; Mann–Whitney test. Scale bar: 5 µm.

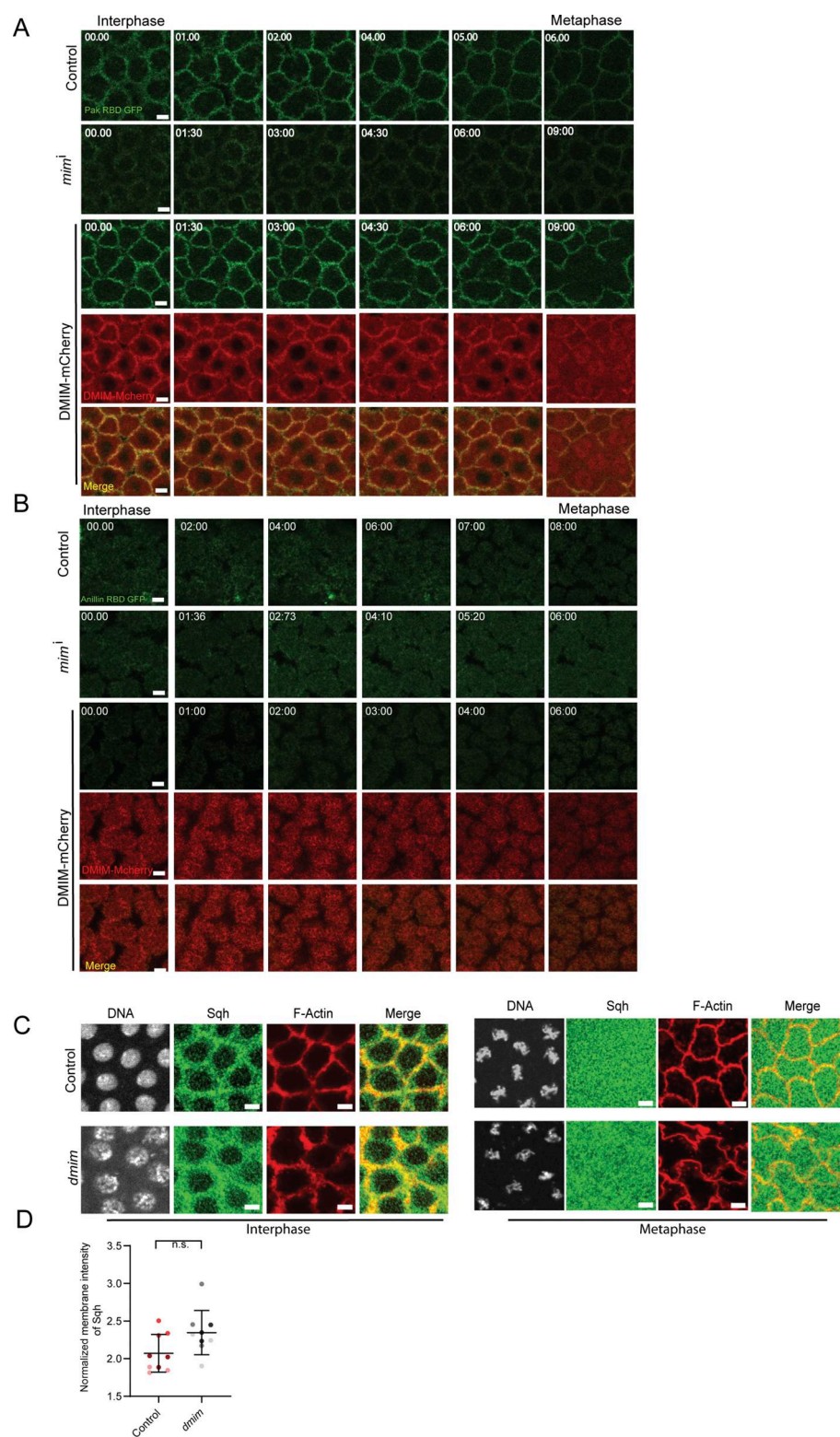

Figure S3. **Analysis of lateral Rac-GTP sensor, apical Rho-GTP sensor, and non-muscle myosin II cap periphery accumulation in DMIM-depleted embryos. (A)** Representative images of the grazing sections from lateral region of cells in the syncytial embryos in cycle 12, from interphase till metaphase, expressing Rac-GTP sensor PAK3-RBD-GFP in control, *mim*[i], and DMIM-mCherry. **(B)** Representative images of the grazing sections from apical region of cells of the syncytial embryos in cycle 12, from interphase till metaphase, expressing Rho-GTP sensor Anillin-RBD-GFP in control, *mim*[i], and DMIM-mCherry. **(C)** Representative grazing section images showing nuclei labelled with Hoechst (gray), non-muscle myosin II labelled with Squash (green), and cortical F-actin labelled with Alexa Fluor Phalloidin 568 (red) in control and *dmim* embryos at interphase and metaphase. **(D)** Graph showing the quantification of normalized membrane intensity of Squash in control and *dmim* embryos. $n$ = 3 cells from 3 embryos for each genotype. Data are represented as mean ± SD; n.s. not significant; Mann–Whitney test. Scale bar: 5 μm.

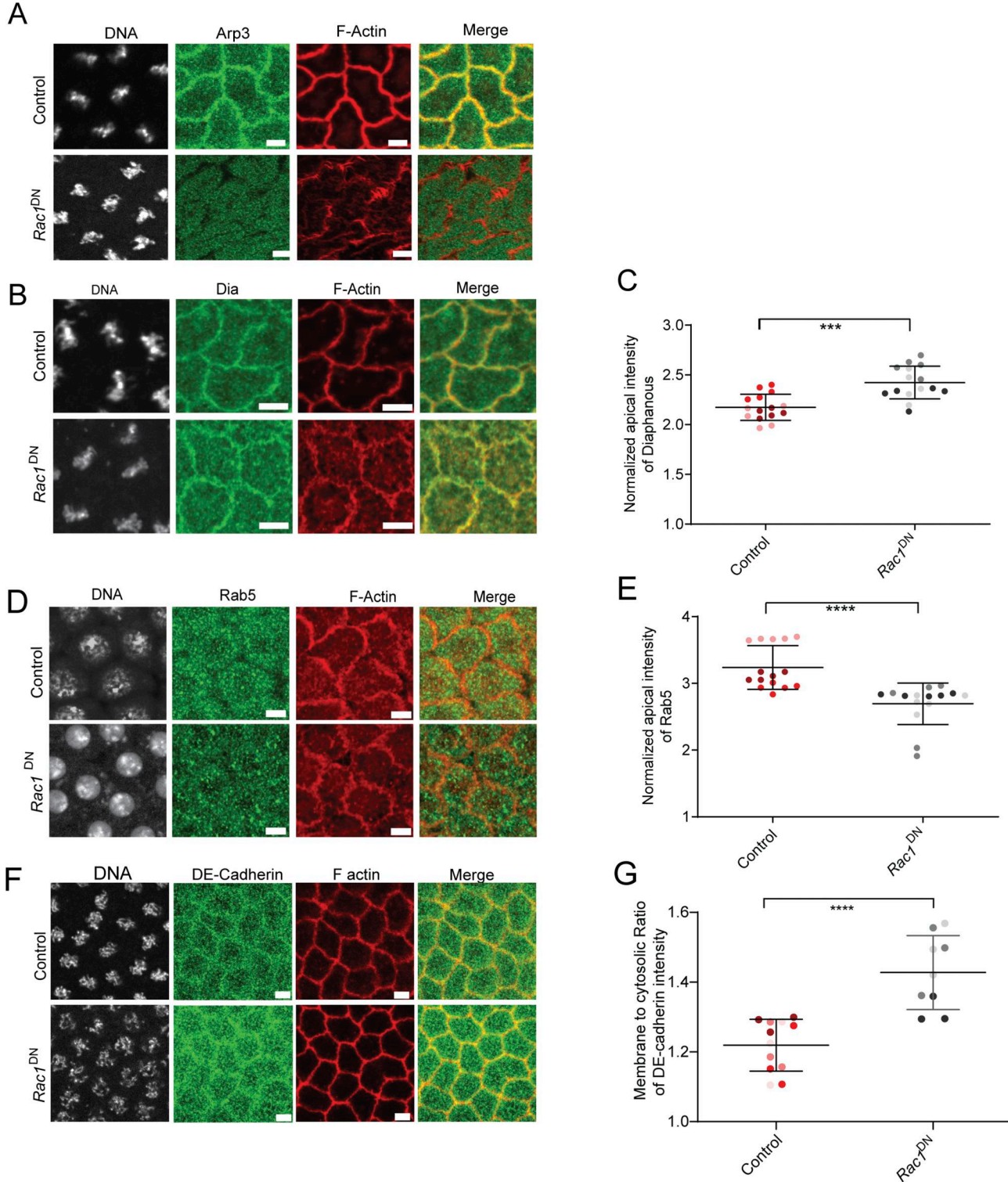

Figure S4. **Active Rac1 recruits Arp3, inhibits Dia, and promotes endocytosis. (A)** Representative grazing section images showing nuclei labelled with Hoechst (gray), Arp3 (green), cortical F-actin labelled with Alexa Fluor Phalloidin 568 (red) in control and Rac1DN embryos. **(B)** Representative grazing section images showing Dia (green), cortical F-actin labelled with Alexa Fluor Phalloidin 568 (red), and nuclei labelled with Hoechst (gray) in control and Rac1DN embryos. **(C)** Graph showing the quantification of the normalized apical intensity of Dia in control and Rac1DN embryos. n = 5 cells from 3 embryos for each genotype. **(D)** Representative grazing section images showing nuclei labelled with Hoechst (gray), early endosomes labelled with Rab5 (green), and cortical F-actin labelled with Alexa Fluor Phalloidin 568 (red) in control and Rac1DN embryos. **(E)** Graph showing the quantification of the normalized apical intensity of Rab5 in control and Rac1DN embryos. n = 5 cells from 3 embryos for each genotype. **(F)** Representative grazing section images showing nuclei labelled with Hoechst (gray), DE-Cad (green), and cortical F-actin labelled with Alexa Fluor Phalloidin 568 (red) in control and Rac1DN embryos. **(G)** Graph showing the quantification of the normalized apical intensity of DE-Cad in control and Rac1DN embryos. n = 3 cells from 3 to 4 embryos for each genotype. Data are represented as mean ± SD; ***P < 0.001, and ****P < 0.0001; Mann–Whitney test. Scale bar: 5 µm.

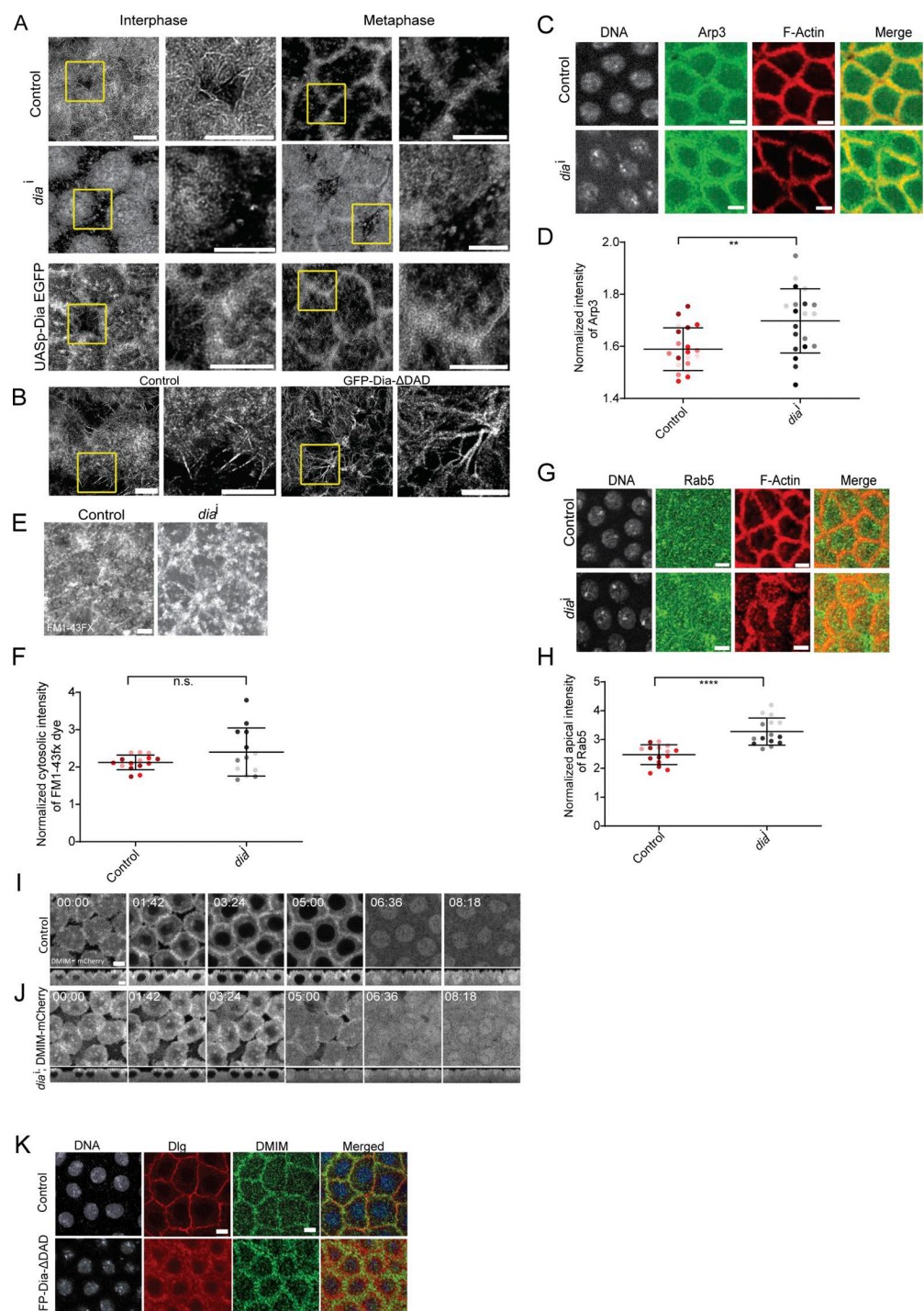

Figure S5.  **Dia is necessary for the formation of the actin protrusions. (A)** STED images showing the F-actin–containing protrusions at interphase and metaphase of nuclear cycle 12 for control, *dia*[i], and Dia-GFP. The region in the yellow box is further zoomed (2.5×) in to show the actin protrusions. **(B)** STED images showing the F-actin–containing protrusions at interphase and metaphase of nuclear cycle 12 for control and GFP-Dia-ΔDAD. The region in the yellow box is further zoomed (2.5×) in to show the actin protrusions. **(C)** Representative grazing section images showing nuclei labelled with Hoechst (gray), Arp3 (green), and cortical F-actin labelled with Alexa Fluor Phalloidin 568 (red) in control and *dia*[i] embryos. **(D)** Graph showing the quantification of the normalized apical intensity of Arp3 in control and *dia*[i] embryos. *n* = 5 cells from 3 to 4 embryos for each genotype. **(E)** Representative grazing section images labelled with FM1-43FX dye at interphase and metaphase in control and *dia*[i] embryos. **(F)** Graphs showing the quantification of the normalized cytosolic intensity of the FM1-43FX dye at interphase and metaphase in control and *dia*[i] embryos. *n* = 5 cells from 3 embryos for each genotype. **(G)** Representative grazing section images showing nuclei labelled with Hoechst (gray), Rab5 (green), and cortical F-actin labelled with Alexa Fluor Phalloidin 568 (red) in control and *dia*[i] embryos. **(H)** Graph showing the quantification of the normalized apical intensity of Rab5 in control and *dia*[i] embryos. *n* = 5 cells from 3 embryos for each genotype. **(I and J)** Representative images showing grazing and sagittal sections of DMIM-mCherry–expressing embryos from interphase to metaphase in control (H) and *dia*[i] (I). **(K)** Representative grazing section images showing nuclei labelled using Hoechst (gray), anti-DMIM (green), and Dlg (red) in control and GFP-Dia-ΔDAD embryos. Data are represented as mean ± SD; **P < 0.01, and ****P < 0.0001, n.s. not significant; Mann–Whitney test. Scale bar: 5 µm.

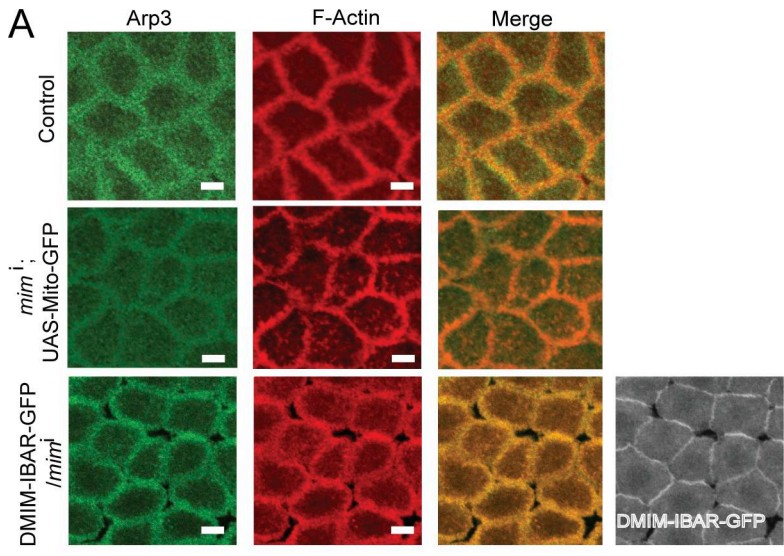

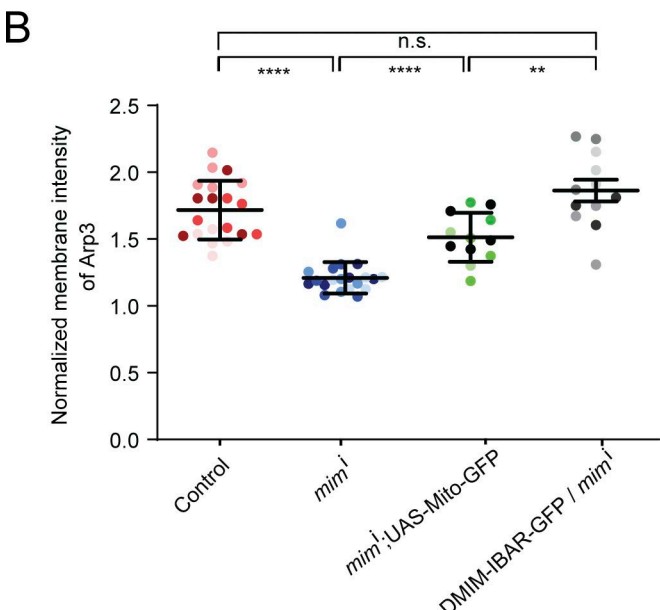

**Figure S6.** **Overexpression DMIM I-BAR domain rescues loss of Arp3 in DMIM-depleted embryos. (A)** Representative grazing section images showing Arp3 (green), cortical F-actin labelled with Alexa Fluor 568 (red), and DMIM-IBAR (gray) in control, *mim*[i]; UAS-mito-GFP, and *mim*[i]; DMIM-IBAR-GFP. **(B)** Graph showing the quantification of the normalized intensity of Arp3 in control, *mim*[i], *mim*[i]; UAS-mito-GFP, and *mim*[i]; DMIM-IBAR-GFP embryos. *n* = 3–5 cells from 4 embryos for each genotype. Data are represented as mean ± SD; *P < 0.05, **P < 0.01, ***P < 0.001, and ****P < 0.0001, n.s. not significant; Mann–Whitney test. Scale bar: 5 µm.

Video 1. **The movie shows live imaging of plasma membrane dynamics with tGPH (gray) in control embryos in syncytial cycle 12.** The apical caps increase in area, whereas the membrane protrusions decrease from interphase to metaphase. Scale bar: 5 µm.

Video 2. **The movie shows live imaging of plasma membrane dynamics with tGPH (gray) in *mim*[i] embryos in syncytial cycle 12.** *mim*[i] embryos show apical caps with bigger area and increased membrane protrusions. Scale bar: 5 µm.

Video 3. **The movie shows live imaging of tGPH (green) and DMIM-mCherry (red) in syncytial nuclear division cycle 12.** DMIM-mCherry shows co-localization with tGPH on the plasma membrane at the beginning of the nuclear cycle in interphase and prophase and becomes cytosolic at metaphase. Scale bar: 5 µm.

Video 4. **The movie shows live imaging of DMIM-mCherry (gray) in *dmim*; DMIM-mCherry embryos in syncytial nuclear division cycle 12.** DMIM-mCherry shows cortical localization from interphase to prophase and becomes cytosolic at metaphase. Scale bar: 5 µm.

Video 5.   **The movie shows live imaging of DMIM-CD-WH2-GFP (gray) in syncytial nuclear division cycle 12.** DMIM-CD-WH2-GFP is cytosolic from interphase to metaphase. Scale bar: 5 µm.

Video 6.   **The movie shows live imaging of DMIM-IBAR-CD-GFP (gray) in syncytial nuclear division cycle 12.** DMIM-IBAR-CD-GFP shows cortical localization from interphase to prophase and remains partially cortical even in metaphase. Scale bar: 5 µm.

Video 7.   **The movie shows live imaging of DMIM-IBAR-GFP (gray) in syncytial nuclear division cycle 12.** DMIM-IBAR-GFP shows cortical localization from interphase to metaphase. Scale bar: 5 µm.

Video 8.   **The movie shows live imaging of DE-Cad-GFP (gray) in control and *mim*[i] embryos in syncytial nuclear division cycle 12.** DE-Cad-GFP localizes to the cortical region at the beginning of the nuclear cycle that is at interphase and gets polarized on the cell boundary as sharp edges by metaphase. In *mim*[i] embryos, the distribution of DE-Cad-GFP is mislocalized and remains accumulated as patches in the cortical region and in the excess protrusions even at metaphase. Scale bar: 5 µm.

Video 9.   **The movie shows live imaging of DMIM-mCherry (gray) in *dia*[i]; DMIM-mCherry in syncytial nuclear division cycle 12.** In DMIM-mCherry embryos, DMIM-mCherry shows cortical localization in interphase and prophase and becomes cytosolic at metaphase. In *dia*[i]; DMIM-mCherry embryos, DMIM-mCherry localizes cortically, and there are increased punctae in the cytosol. Scale bar: 5 µm.

Video 10.   **The movie shows live imaging of DMIM-mCherry (gray) in *Rac1*[DN]; DMIM-mCherry in syncytial nuclear division cycle 12.** In DMIM-mCherry embryos, DMIM-mCherry shows cortical localization in interphase and prophase and becomes cytosolic in metaphase. In *Rac1*[DN]; DMIM-mCherry embryos, DMIM-mCherry shows patchy cortical distribution with enrichment in cap edges in a few places. Scale bar: 5 µm.

