## [Peer Review File · The Journal of Cell Biology]

MIM triggers formin to Arp2/3-based actin assembly in membrane remodeling in *Drosophila* embryos

Debasmita Mitra, Georgina Goddard, Sanjana S, Aparna K, Tom Millard, and Richa Rikhy

Corresponding Author(s): Richa Rikhy, Indian Institute of Science Education and Research Pune

Review Timeline:

Submission Date:	2025-02-26
Editorial Decision:	2025-04-10
Revision Received:	2025-11-20
Editorial Decision:	2025-12-17
Revision Received:	2025-12-25

Monitoring Editor: Pekka Lappalainen

Scientific Editor: Gabriele Stephan

Transaction Report:

DOI: <https://doi.org/10.1083/jcb.202502184>

April 10, 2025

Re: JCB manuscript #202502184

Richa Rikhy
Indian Institute of Science Education and Research Pune

Dear Dr. Rikhy,

Thank you for submitting your manuscript entitled "MIM triggers formin to Arp2/3-based actin assembly in membrane remodeling in *Drosophila* embryos". Your manuscript has been assessed by expert reviewers, whose comments are appended below. Although the reviewers express potential interest in this work, significant concerns unfortunately preclude publication of the current version of the manuscript in JCB.

You will see that the reviewers are overall supportive, thinking that this is a potentially interesting paper. However, they voice significant concerns regarding the layout of certain experiments, such as those regarding the over-expression of Diaphanous and MIM, the analysis of Rho1- and Rac-1 activity, and the role of the MTSS1/MIM WH2 domain. They further criticize the lack of mechanistic insights presented in this manuscript and reviewer #2 suggests substantial experiments to improve the paper. Reviewer #3 criticizes the characterization of lateral protrusions as microvilli, as well as the analysis of MTSS1/MIM dynamics. Additional clarifications on the dependence of Arp3, Dia and regulators on MTSS1/MIM as well as on the role of Rac signaling should further be addressed. Finally, the data should be put better into the context of the previous data on MIM/MTSS1. For example, there are earlier publications reporting interplay between MTSS1 and Arp2/3 & WAVE2 complexes, but these were not discussed in the present version of the manuscript.

Please let us know if you are able to address all concerns and wish to submit a revised manuscript to JCB. Note that a substantial amount of additional experimental data likely would be needed to satisfactorily address the concerns of the reviewers. The typical time frame for revisions is three to four months. If you anticipate any difficulties in meeting this aforementioned revision time limit, please contact us and we can work with you to find an appropriate time frame for resubmission. Please note that papers are generally considered through only one revision cycle, so any revised manuscript will likely be either accepted or rejected.

If you choose to revise and resubmit your manuscript, please also attend to the following editorial points. Please direct any editorial questions to the journal office.

GENERAL GUIDELINES:

Text limits: Character count is < 40,000, not including spaces. Count includes title page, abstract, introduction, results, discussion, and acknowledgments. Count does not include materials and methods, figure legends, references, tables, or supplemental legends.

Figures: Your manuscript may have up to 10 main text figures. To avoid delays in production, figures must be prepared according to the policies outlined in our Instructions to Authors, under Data Presentation, <https://jcb.rupress.org/site/misc/ifora.xhtml>. All figures in accepted manuscripts will be screened prior to publication.

Supplemental information: There are strict limits on the allowable amount of supplemental data. Your manuscript may have up to 5 supplemental figures. Up to 10 supplemental videos or flash animations are allowed. A summary of all supplemental material should appear at the end of the Materials and methods section.

Please note that JCB now requires authors to submit Source Data used to generate figures containing gels and Western blots with all revised manuscripts. This Source Data consists of fully uncropped and unprocessed images for each gel/blot displayed in the main and supplemental figures. For assays performed using capillary electrophoresis and/or immunoassay-based detection, authors should instead provide the electropherogram graph(s) for each experiment, plotting fluorescence/chemiluminescence intensity vs. molecular weight/size. Please be sure to provide one Source Data file for each figure gels, blots, and/or capillary electrophoresis assays along with your revised manuscript files. File names for Source Data figures should be alphanumeric without any spaces or special characters (i.e., SourceDataF#, where F# refers to the associated main figure number or SourceDataFS# for those associated with Supplementary figures). For traditional gels and blots, the lanes of the gels/blots should be labeled as they are in the associated figure, the place where cropping was applied should be marked (with a box), and molecular weight/size standards should be labeled wherever possible. For capillary electrophoresis assays, each trace in the graph should be color-coded and labeled to indicate which protein, gene, or sample is being measured

(please try to avoid red/green combinations to accommodate our color-blind readers).

If you choose to resubmit, please include a cover letter addressing the reviewers' comments point by point. Please also highlight all changes in the text of the manuscript.

Regardless of how you choose to proceed, we hope that the comments below will prove constructive as your work progresses. We would be happy to discuss them further once you've had a chance to consider the points raised. You can contact the journal office with any questions at cellbio@rockefeller.edu.

Thank you for thinking of JCB as an appropriate place to publish your work.

Sincerely,

Pekka Lappalainen
Monitoring Editor
Journal of Cell Biology

Gabriele Stephan, PhD
Scientific Editor
Journal of Cell Biology

Reviewer #1:

In this manuscript, Mitra and colleagues investigate the role of MIM, a BAR-domain protein, in the syncytial divisions of the early *Drosophila* embryo. Nuclear divisions in the fly embryo are associated with the depletion of apical microvilli and the formation of membrane furrows in between the nuclei. These activities require actin remodelling and endocytic activity. MIM loss of function resulted in longer and more abundant microvilli, larger apical cell surfaces and shallower furrows. In contrast, overexpression of the BAR and scaffold domains of MIM (MIM-M3) reduced the length and number of microvilli, and rescued the deleterious effects of the mutant of microvillus length and number. MIM localized to the membrane, but became cytosolic in metaphase. Structure function analysis suggested that the BAR domain controls membrane localization in interphase, and that the central and WH2 domains contribute to the change to cytosolic localization. Consistent with a role in promoting endocytosis, MIM loss of function reduced the uptake of a fluorescent dye, as well as the numbers of Rab5 and Rab11 puncta in the cells. MIM also controls F-actin regulators, and reducing MIM levels reduced Arp2/3, Cortactin and SCAR localization to the cell cortex. In contrast, the levels of the formin Diaphanous (which can compete with Arp2/3 for actin) increased when MIM levels decreased. Reducing the activity of the small GTPase Rac1 (potentially upstream of Arp2/3) largely phenocopied MIM loss of function, and so did increasing the levels of Diaphanous. In contrast, reducing Diaphanous levels caused phenotypes similar to MIM overexpression. Consistent with an interaction between Rac1 and MIM, co-expression of MIM and a dominant negative form of Rac1 partially rescued the phenotypes caused by MIM overexpression alone.

I enjoyed this study. It is well written and easy to follow. The interplay between Rac1, Arp2/3 and Dia is not too surprising (it has been demonstrated in other systems before), but the role of MIM coordinating membrane/actin remodelling and endocytic activity is interesting. However, I think some of the reagents used were suboptimal (e.g. overexpressing Diaphanous when a constitutively active form exists; or looking at total-Rac1 levels when there are probes for Rac1 activity), and some important controls are missing. I propose to address the following points:

MAJOR

1. The authors propose that MIM is an inhibitor of Diaphanous (line 354), and suggest that the effect could be through Rho1 (Discussion). Is that the case? Does apical Rho1 activity (not levels!!) increase/decrease with MIM loss/gain of function, respectively? This would strengthen the connection between MIM and GTPase-based cytoskeletal remodelling.
2. Figure 5G: what is the probe used to visualize Rac1? I could not find it listed in the methods. If this is showing total Rac1, the authors should use a probe of Rac1 activity (e.g. PlexinB RBD, <https://pmc.ncbi.nlm.nih.gov/>).
3. Figure S4C-E and line 392: The effect of overexpressing Diaphanous is not as strong as reducing DMIM. How about

overexpressing a constitutively active form of Diaphanous (e.g. Dia-delta(Dad), <https://pmc.ncbi.nlm.nih.gov/articles/PMC2793291/>)? Does that phenocopy the MIM loss of function more closely?

4. Figure 8: the rescue experiment is really neat! However, I could not find anywhere information about whether the individual Rac1DN or MIM-M3-mCherry treatments carried a second UAS construct. The concern here is that when you combine the two UAS constructs, the phenotypes may change because the Gal4 protein is split among the two UAS constructs and their expression levels decrease. Thus, Rac1DN or MIM-M3-mCherry should be co-expressed with a second UAS (e.g. luciferase, cerulean, etc.) to show that with the reduced Gal4 dosage they can still cause the expected phenotypes.

5. Figure 3G and line 217: the authors argue that the MIM-M3-delta(WH2) construct does not become cytosolic, but the membrane signal appears to decrease, and the quantification in Fig. 3I does not show an obvious difference with MIM-M3 at 8.18 minutes. To this reviewer, it is not clear that the WH2 domain is necessary for the membrane-to-cytosolic change in MIM localization during metaphase. The authors should examine the localization of a MIM-M3-delta(central scaffold) that still preserves the I-BAR and WH2 domains. Is the membrane retention of that construct lower than that of MIM-M3-delta(Central scaffold+WH2)?

6. The use of Student's t-test to compare means is not appropriate unless the authors validate that the distributions of the samples that they compare are normal. If normality is not tested, then a non-parametric test (e.g. Mann-Whitney) should be used.

MINOR

7. Figure 6I-J: the authors argue that both Arp3 and DMIM localize to the base on microvilli in a punctate distribution. However, from the images, the size of the Arp3 puncta seem smaller, and the number of puncta greater. Quantitative analysis of the colocalization of MIM and Arp3 would further support the idea that DMIM is controlling Arp2/3 recruitment to microvilli to promote their remodelling.

8. Figure 1: this is picky, but the authors are not overexpressing DMIM, but part of it. As such, I suggest to revise sentences such as "DMIM was overexpressed" (line 155) with "MIM-M3 was overexpressed".

TYPOS

9. Line 295: "changed DMIM depleted embryos" should be "changed in DMIM-depleted embryos".

Reviewer #2:

Comments for Mitra et al

This paper examines the role of MIM and its associated protein in Drosophila syncytium development. While MIM's presence at the cell junction is documented (PMID 36871754), the new insight is its role in syncytium development and endocytosis affecting microvilli formation. However, MIM is I-BAR protein for filopodia and protrusions, and how MIM is involved in endocytosis needs molecular models. The work contains substantial work; however, the molecular mechanisms behind it were not reported.

1. The use of MIM-M3, a truncated form, to fuse with mCherry is unclear. The DMIM localization and MIM-M3 localization should be compared. Some key data should be re-analyzed by DMIM tagged with mCherry instead of MIM-M3-mCherry.
2. Endocytosis was shown by the uptake of FM dye in most of the figures. FM dye is lipophilic dye it would not suggest the endocytosis of the membrane. Some complementary approaches would be desired. Dynamin inhibitor treatment would be the candidate approach.
3. The mutant in Figure 3 should be used to rescue the mutant phenotype for a molecular understanding of the action of MIM.
4. If Rac binds to MIM, then the binding region would be determined and then implemented by the rescue experiments.
5. Lines 134-135 state microvilli reduction from interphase to metaphase but lack statistical comparison support.
6. Lines 198-199 discuss decreased MIM-M3-mCherry intensity from interphase to metaphase without showing significance in related graphs, including Fig 3I.
7. Lines 356-358 mention Arp3 distribution without measuring co-localization levels. Provide co-localization values to confirm insignificance.
8. Fig 7G-H states Rac1DN leads to loss of DMIM antibody fluorescence, but images show remaining fluorescence-quantify this difference.
9. Fig 8F-G describes partial recovery of microvilli defects with Rac1DN overexpression but lacks quantification.
10. Specify the number of microvilli considered for length measurement in Fig 8D.
11. Improve Fig 9 illustration with more labeling and descriptions for clarity.

12. Differentiate cells and embryos with distinct colors in graphs, ensuring dots correspond to their respective counts. The number of dots does not appear to correspond to a number of cells or embryos.

Reviewer #3:

BAR proteins contribute to linking the plasma membrane with the actin cortex and the dynamics of the plasma membrane. The family-defining and membrane-binding BAR domain provides a curved surface, which bends membranes or senses curvature. In addition, they contain various domains, such as WH2 or SH3, which interact with the actin cytoskeleton.

Although having been studied for a number of years, the functions of BAR proteins including MIM have remained unclear, especially how the well-defined molecular activities contribute to specific roles in complex cell physiology, development and morphogenesis.

Here, Mitra et al. report a study of the I-BAR protein MIM membrane surface structure in syncytial *Drosophila* embryos.

The claims of the study are that MIM mediates a balance between Arp2/3-dependent branched actin and Dia dependent linear actin, which is observed in more and longer microvilli following MIM depletion, that MIM mediates cortical localization of Arp2/3 and some of its regulators, that Rac signaling mediates membrane localization of MIM.

The authors focus on nuclear cycles in syncytial embryos, which is characterized by spatially separated domains, (1) caps, rich in branched F-actin with Rac signaling and Arp2/3 activity and (2) the region outside of the disc-like caps, rich Rho signaling, myoII and dia, which will extend in mitosis to form the pseudo cleavage furrow. The authors start with the observation of micrometer long filapodia like protrusions, which are more abundant and longer in MIM depleted embryos than in wild type and less so in embryos overexpressing MIM. These protrusions are scored in surface views of the embryos and are thus laterally oriented extending from the caps towards the region between the caps. The authors also describe a reduced length of the metaphase furrow in MIM depleted embryos, which is comparable to dia mutants. It is confusing to me why the lateral protrusions are designated as microvilli, which would expect at the apical surface of the caps in a perpendicular orientation. These micrometer long protrusions do not look like typical microvilli to me, because of their length and because of their orientation. I would like to see either a demonstration that the microvilli in the caps are also affected in MIM embryos or alternatively a less-biased naming of the structures. In any case the micrometer-long protrusions should be better characterized. Are these F-actin structures surrounded by membrane? Are they labeled by bundling proteins? In fig.2 the authors present data with a membrane label. Here not such micro-meter long protrusions are observed. The images look very different, also because the images are shown in a smaller magnification. For clarity and easy readability, it would be advantageous to have the F-actin and membrane label side by side in the same magnification and ideally with double labelling. The same scoring should be used. The dynamics of MIM was assayed with a tagged version in a wild type and MIM mutant background. In my eyes the presented data are not fully consistent. MIM-Cherry in a mutant background with no potential competition between tagged and untagged molecules, Fig. S1F, shows some enrichment in particles apical to the nuclei. I do not see an enrichment in the region that becomes the metaphase furrow. In contrast, in case of competition, the label is clearly enriched in the furrow region. Clarification is required. It is also not clear to me, why the more convincing experiment is shown in the supplement, whereas the potentially more complicated experiment is shown in the main figures.

In Figure 5+6 the authors show a dependence of Arp3, Dia and regulators on MIM. Although the DAPI channel shows the interphase status of the cells, F-actin is present in the furrows but not in caps. In interphase F-actin, as well as Arp2/3 and its regulators are supposed to be largely present in the apical actin caps while dia and Myo are present in the furrow region, as I understand. Clarification is required probably by providing axial views.

Dia

In Figures 7+8, the authors investigate the relationship of Rac signaling and MIM. They employ expression of a dominant negative Rac mutant to suppress Rac signalling. Staining for F-actin shows that this induces a comparable phenotype as loss of MIM concerning the protrusions and reduction of endocytosis. The authors conclude that MIM depends on Rac signaling because the overall staining and enrichment is reduced in the furrow region. Formally the conclusion is valid. I am confused however, because Rac signaling is restricted to the cap region but an effect is observed outside of the cap region. This indicates me, that the effect is most likely indirect. The authors may complement their data with an alternative interference of Rac signaling such as with mutants of its upstream GEFs, Sponge-ELMO.

In summary, this interesting study provides novel insights into how the I-BAR protein MIM contributes to the antagonism of linear and branched F-actin at the cortex. I recommend publication of the study. Before moving to publication clarification of several issues is needed partially involving substantial new experiments and data.

Specific comments

- MIM mutants are viable and fertile. No overall morphological defects in syncytial blastoderm are apparently observed and nuclear divisions proceed fine in mutants. In my view it is an overstatement to designate MIM a "key integrator" (l. 33)

- L. 44. Although previous studies have demonstrated an antagonism between Arp2/3-dependent branched actin and Dia-dependent linear F-actin, I am not aware of studies showing that the antagonism would be due to a competition for the free pool of G-actin molecules.
- L. 154. Tagged MIM was expressed from a UAS transgene. Is it clear that this results in "overexpression"? The authors should check expression levels in comparison to endogenous MIM by western blot or alternatively avoid the term "overexpression".
- Fig. 4. Assay for endocytosis with FM dye. The images in panel A and the quantification in B are not consistent. Panel A shows a more or less complete loss of signal (black images). Quantification shows, however, shows a reduction to not even half.
- Dia and Rac1 were localized by antibody staining. I did not spot any source for the Rac1 antibody. As the antibody does not distinguish active and inactive forms of the proteins, the stainings are not very informative. The authors may employ probes for the active forms, i. e. a Rac bio sensor.

We thank the editor and the reviewers for their encouraging comments on our manuscript and
for their suggestions on experiments to increase the mechanistic analysis. We have taken time
to revise the manuscript as the revisions involved obtaining *Drosophila* lines, performing
multigenerational crosses, followed by imaging and analysis. We have performed new
experiments on the regulation of Rac-GTP and Rho-GTP levels by MIM, as well as the role of
the MIM I-BAR domain, role of Dynamin and RacGEF/Sponge in regulating membrane
remodeling. The revisions enable us to compile a summary model on the function of MIM in
regulating active Rac, and active Rac in controlling MIM. Rac-GTP-MIM, together, in turn,
activates the Wave regulatory complex and the Arp2/3 complex. The MIM, Rac-GTP, WRC,
Arp2/3 complex pathway inhibits the distribution of RhoGEF2 and activation of Rho, and Dia to
coordinate plasma membrane remodelling in the syncytial division cycles in *Drosophila*
embryogenesis. We summarise the new experiments added as a part of the revisions and
provide responses to each of the reviewers' comments in a point-wise manner below.

Re: JCB manuscript #202502184

Richa Rikhy
Indian Institute of Science Education and Research Pune

Dear Dr. Rikhy,

Thank you for submitting your manuscript entitled "MIM triggers formin to Arp2/3-based actin
assembly in membrane remodeling in *Drosophila* embryos". Your manuscript has been
assessed by expert reviewers, whose comments are appended below. Although the reviewers
express potential interest in this work, significant concerns unfortunately preclude publication of
the current version of the manuscript in JCB.

You will see that the reviewers are overall supportive, thinking that this is a potentially
interesting paper. However, they voice significant concerns regarding the layout of certain
experiments, such as those regarding the over-expression of Diaphanous and MIM, the
analysis of Rho1- and Rac-1 activity, and the role of the MTSS1/MIM WH2 domain.

**Response:** In the revised version of the manuscript, we have added the following new
experiments to address these specific points of the reviewers:

- 1. We have added analysis of Rho1 and Rac1 activity using their sensors. We find that
levels of active Rac-GTP depend upon MIM at the membrane, while the presence of
MIM reduces levels of Rho-GTP.
- 2. We have added new experiments elucidating the role of the MIM IBAR domain in
supporting actin protrusion remodelling.
- 3. We have added new experiments with activated Dia (Dia Δ DAD), which show increased
apical actin protrusions.
- 4. We have added new experiments to demonstrate MIM's regulation of RhoGEF2
distribution.

- 5. We have added new experiments for the role of Dynamin in regulating apical actin
protrusion remodelling.
6. We have added new experiments to show that the Rac GEF, ELMO/Sponge, regulates
apical actin protrusion remodelling.
7. We have added better images showing the distribution of Arp3, Dia, and the membrane
relative to F-actin.
8. We have added more representative images to show the recruitment of MIM domains
during the syncytial cycle. We have redone the quantification to compare interphase and
metaphase.
9. We have changed the statistics to Mann-Whitney analysis and the plots to super plots.
10. We have added experiments to control for Gal4 dosage dilution in embryos expressing 2
transgenes compared to 1 transgene.
11. We have added images of membrane and actin in apical sections and changed the label
for microvilli to apical actin protrusions, as they contain F-actin and are found on the
apical cap. We refer to them as apical actin protrusions or villous protrusions.
12. We have added literature to discuss our experiments and observations in the revised
manuscript.

They further criticize the lack of mechanistic insights presented in this manuscript and reviewer
#2 suggests substantial experiments to improve the paper.

Response: We have added new experiments to address the impact of MIM on Rac-GTP and
Rho-GTP levels at the plasma membrane, analysis of the role of I-BAR domain in regulating
actin-protrusion remodelling, analysis of the Rho pathway molecules, RhoGEF2 and Sqh and
analysis of the RacGEF, ELMO, to the revised manuscript.

Reviewer #3 criticizes the characterization of lateral protrusions as microvilli, as well as the
analysis of MTSS1/MIM dynamics. Additional clarifications on the dependence of Arp3, Dia and
regulators on MTSS1/MIM as well as on the role of Rac signaling should further be addressed.

Response: We previously called the actin rich protrusions as microvilli because, like the
intestinal brush border, they contain bundled actin, are membrane-bound, increase surface area
and are rapidly remodelled in the syncytial division cycle (Mitra et al., 2022; Sherlekar et al.,
2020). We have described the protrusions better in the revised manuscript. They are micron-
scale structures and present in the apical cap. They are membrane-bound and contain F-actin.
In grazing sections used for quantification, they are predominantly seen at the periphery,
emerging from the edge of the caps. We have changed the term to apical actin protrusions in
the revised manuscript. In *Drosophila*, cellularization, the protrusions in apical caps are referred
to as villous protrusions (Fabrowski et al., 2013).

We have added new experiments on sensors of Rho-GTP and Rac-GTP, added experiments
with Dia Δ DAD and Sponge/ELMO for further probing Rac signaling in the revised manuscript.

Finally, the data should be put better into the context of the previous data on MIM/MTSS1. For
example, there are earlier publications reporting interplay between MTSS1 and Arp2/3 &
WAVE2 complexes, but these were not discussed in the present version of the manuscript.

**Response:** We have added literature on previous analysis of MTSS1/MIM domains and the role
of MTSS1/MIM in regulating Arp2/3 and WAVE2 complexes, as seen previously in the revised
version of the manuscript.

Please let us know if you are able to address all concerns and wish to submit a revised
manuscript to JCB. Note that a substantial amount of additional experimental data likely would
be needed to satisfactorily address the concerns of the reviewers. The typical time frame for
revisions is three to four months. If you anticipate any difficulties in meeting this aforementioned
revision time limit, please contact us and we can work with you to find an appropriate time
frame for resubmission. Please note that papers are generally considered through only one
revision cycle, so any revised manuscript will likely be either accepted or rejected.

If you choose to revise and resubmit your manuscript, please also attend to the following
editorial points. Please direct any editorial questions to the journal office.

GENERAL GUIDELINES:

Text limits: Character count is < 40,000, not including spaces. Count includes title page,
abstract, introduction, results, discussion, and acknowledgments. Count does not include
materials and methods, figure legends, references, tables, or supplemental legends.

Figures: Your manuscript may have up to 10 main text figures. To avoid delays in production,
figures must be prepared according to the policies outlined in our Instructions to Authors, under
Data Presentation, <https://jcb.rupress.org/site/misc/ifora.xhtml>. All figures in accepted
manuscripts will be screened prior to publication.

*****IMPORTANT:** It is JCB policy that if requested, original data images must be made available.
Failure to provide original images upon request will result in unavoidable delays in publication.
Please ensure that you have access to all original microscopy and blot data images before
submitting your revision.***

Supplemental information: There are strict limits on the allowable amount of supplemental data.
Your manuscript may have up to 5 supplemental figures. Up to 10 supplemental videos or flash
animations are allowed. A summary of all supplemental material should appear at the end of the
Materials and methods section.

Please note that JCB now requires authors to submit Source Data used to generate figures
containing gels and Western blots with all revised manuscripts. This Source Data consists of
fully uncropped and unprocessed images for each gel/blot displayed in the main and
supplemental figures. For assays performed using capillary electrophoresis and/or
immunoassay-based detection, authors should instead provide the electropherogram graph(s)

for each experiment, plotting fluorescence/chemiluminescence intensity vs. molecular
weight/size. Please be sure to provide one Source Data file for each figure gels, blots, and/or
capillary electrophoresis assays along with your revised manuscript files. File names for Source
Data figures should be alphanumeric without any spaces or special characters (i.e.,
SourceDataF#, where F# refers to the associated main figure number or SourceDataFS# for
those associated with Supplementary figures). For traditional gels and blots, the lanes of the
gels/blots should be labeled as they are in the associated figure, the place where cropping was
applied should be marked (with a box), and molecular weight/size standards should be labeled
wherever possible. For capillary electrophoresis assays, each trace in the graph should be
color-coded and labeled to indicate which protein, gene, or sample is being measured (please
try to avoid red/green combinations to accommodate our color-blind readers).

Source Data files will be made available to reviewers during evaluation of revised manuscripts
and, if your paper is eventually published in JCB, the files will be directly linked to specific
figures in the published article.

Source Data Figures should be provided as individual PDF files (one file per figure). Authors
should endeavor to retain a minimum resolution of 300 dpi or pixels per inch. Please review our
instructions for export from Photoshop, Illustrator, and PowerPoint here:

<https://rupress.org/jcb/pages/submission-guidelines#revised>

If you choose to resubmit, please include a cover letter addressing the reviewers' comments
point by point. Please also highlight all changes in the text of the manuscript.

Regardless of how you choose to proceed, we hope that the comments below will prove
constructive as your work progresses. We would be happy to discuss them further once you've
had a chance to consider the points raised. You can contact the journal office with any
questions at cellbio@rockefeller.edu.

Thank you for thinking of JCB as an appropriate place to publish your work.

Sincerely,

Pekka Lappalainen
Monitoring Editor
Journal of Cell Biology

Gabriele Stephan, PhD
Scientific Editor
Journal of Cell Biology

-----

Summary of the revisions performed in response to the comments by Reviewer 1, 2 and 3:

- 1. We have added analysis of Rho1 and Rac1 activity using their sensors. We find that
- levels of active Rac-GTP depend upon MIM at the membrane, while the presence of
- MIM reduces levels of Rho-GTP.
- 2. We have added new experiments elucidating the role of the MIM IBAR domain in
- supporting actin protrusion remodelling.
- 3. We have added new experiments with activated Dia (Dia Δ DAD), which show increased
- apical actin protrusions.
- 4. We have added new experiments to demonstrate MIM's regulation of RhoGEF2
- distribution.
- 5. We have added new experiments for the role of Dynamin in regulating apical actin
- protrusion remodelling.
- 6. We have added new experiments to show that the Rac GEF, ELMO/Sponge, regulates
- apical actin protrusion remodelling.
- 7. We have added better images showing the distribution of Arp3, Dia, and the membrane
- relative to F-actin.
- 8. We have added more representative images to show the recruitment of MIM domains
- during the syncytial cycle. We have redone the quantification to compare interphase and
- metaphase.
- 9. We have changed the statistics to Mann-Whitney analysis and the plots to super plots.
- 10. We have added experiments to control for Gal4 dosage dilution in embryos expressing 2
- transgenes compared to 1 transgene.
- 11. We have added images of membrane and actin in apical sections and changed the label
- for microvilli to apical actin protrusions, as they contain F-actin and are found on the
- apical cap. We refer to them as apical actin protrusions or villous protrusions.
- 12. We have added literature to discuss our experiments and observations in the revised
- manuscript.

Reviewer #1:

Summary of the revisions performed in response to the comments by Reviewer 1:

We performed the following revisions in response to the comments by Reviewer 1:

- 1. We have added a new experiment to test the levels of Rac-GTP with the PAK3-RBD-
- GFP sensor.
- 2. We have performed analysis of RhoGEF2 distribution by immunostaining in MIM
- depleted embryos. We have added a new experiment to test the levels of Rho-GTP with
- the Anillin-RBD-GFP sensor.
- 3. We have added an experiment in which an activated form of Dia, Dia Δ DAD is
- overexpressed to assess its role in increasing apical actin protrusions.
- 4. We have added experiments to control for Gal4 dosage dilution in embryos expressing 2
- transgenes compared to 1 transgene.

- 5. We have added more representative images of MIM domains in their recruitment in the
syncytial cycle. We have redone the quantification to compare the distribution at the
membrane in interphase and metaphase.
6. We have changed the statistics to Mann-Whitney analysis and the plots to super plots.
7. We have changed the name MIM-M3-mCherry to DMIM-mCherry to avoid confusion,
since the full-length construct is being used for rescue experiments.
8. We have corrected the typos in the manuscript that the reviewer pointed out.

In this manuscript, Mitra and colleagues investigate the role of MIM, a BAR-domain protein, in
the syncytial divisions of the early *Drosophila* embryo. Nuclear divisions in the fly embryo are
associated with the depletion of apical microvilli and the formation of membrane furrows in
between the nuclei. These activities require actin remodelling and endocytic activity. MIM loss
of function resulted in longer and more abundant microvilli, larger apical cell surfaces and
shallower furrows. In contrast, overexpression of the BAR and scaffold domains of MIM (MIM-
M3) reduced the length and number of microvilli, and rescued the deleterious effects of the
mutant of microvillus length and number. MIM localized to the membrane, but became cytosolic
in metaphase. Structure function analysis suggested that the BAR domain controls membrane
localization in interphase, and that the central and WH2 domains contribute to the change to
cytosolic localization. Consistent with a role in promoting endocytosis, MIM loss of function
reduced the uptake of a fluorescent dye, as well as the numbers of Rab5 and Rab11 puncta in
the cells. MIM also controls F-actin regulators, and reducing MIM levels reduced Arp2/3,
Cortactin and SCAR localization to the cell cortex. In contrast, the levels of the formin
Diaphanous (which can compete with Arp2/3 for actin) increased when MIM levels decreased.
Reducing the activity of the small GTPase Rac1 (potentially upstream of Arp2/3) largely
phenocopied MIM loss of function, and so did increasing the levels of Diaphanous. In contrast,
reducing Diaphanous levels caused phenotypes similar to MIM overexpression. Consistent with
an interaction between Rac1 and MIM, co-expression of MIM and a dominant negative form of
Rac1 partially rescued the phenotypes caused by MIM overexpression alone.

I enjoyed this study. It is well written and easy to follow. The interplay between Rac1, Arp2/3
and Dia is not too surprising (it has been demonstrated in other systems before), but the role of
MIM coordinating membrane/actin remodelling and endocytic activity is interesting. However, I
think some of the reagents used were suboptimal (e.g. overexpressing Diaphanous when a
constitutively active form exists; or looking at total-?-Rac1 levels when there are probes for
Rac1 activity), and some important controls are missing. I propose to address the following
points:

**Response:** We thank the reviewer for their encouraging comments on our findings of the role of
I-BAR domain protein MIM in regulating plasma membrane remodelling in syncytial *Drosophila*
embryos.

As recommended by the reviewer, we have added an analysis for an activated version of Dia,
Dia Δ DAD in apical actin protrusion formation in early *Drosophila* embryos. Expression of
Dia Δ DAD did not result in embryos with maternal Gal4. We then combined this with maternal

Gal4 and Gal80 to suppress the Gal4 levels and obtained embryos that contained a strong
increase in apical actin protrusions at the apical surface. MIM is present in these regions at the
cortex, and therefore, these increased protrusions are likely to be independent of MIM loss.

The transgene encoding the Rac-GTP-binding domain of PAK3-RBD-GFP is used to assess
changes in active Rac levels in MIM RNAi. Unlike total Rac antibody staining, which shows no
change, we find that PAK3-RBD-GFP decreases significantly upon MIM depletion and
increases significantly upon MIM overexpression. Further there are distinct patches of
colocalization of PAK3-RBD-GFP with DMIM-mCherry (previously called MIM-M3-mCherry).
We also see that *Rac1^{DN}* shows a loss of MIM. A combination of DMIM-mCherry;*Rac1^{DN}* is
more like *Rac1^{DN}* and shows abundant and long apical actin protrusions thereby showing that
they regulate each other.

Moreover, we could show that the MIM-I-BAR domain which is known in literature to bind Rac-
GTP is able to rescue the apical actin protrusion defect seen on MIM depletion.

MAJOR

1. The authors propose that MIM is an inhibitor of Diaphanous (line 354), and suggest that the
effect could be through Rho1 (Discussion). Is that the case? Does apical Rho1 activity (not
levels!!) increase/decrease with MIM loss/gain of function, respectively? This would strengthen
the connection between MIM and GTPase-based cytoskeletal remodelling.

Response: We have assessed the Rho pathway activity in MIM depleted embryos in the
following ways:

- 1. We find that RhoGEF2, the GTP exchange factor for Rho has increased localization in
sub apical regions on MIM depletion This is a part of new Figure 7.
- 2. Increase in RhoGEF2 in MIM depleted embryos is coincident with an increase in Dia.
Sqh is also increased in many embryos, but since there is a variability in levels of Sqh,
this increase is not statistically significant. We have also previously shown that an
increase in RhoGEF2 leads to an increase in actomyosin contractility and an increase in
Sqh in syncytial cycles thereby affecting furrow extension(Dey and Rikhy, 2020).
- 3. The Rho-GTP binding domain of Anillin, Anillin-RBD-GFP is increased on MIM depletion
and decreased on MIM overexpression. This is a part of new Figure 7.

2. Figure 5G: what is the probe used to visualize Rac1? I could not find it listed in the methods.
If this is showing total Rac1, the authors should use a probe of Rac1 activity (e.g. PlexinB RBD,
<https://pmc.ncbi.nlm.nih.gov/>).

Response: As suggested by the reviewer we have performed live imaging of Rac-GTP binding
domain of PAK3 in PAK3-RBD-GFP expressing embryos depleted of MIM and overexpressing
MIM. We find that PAK3-RBD-GFP fluorescence decreases in apical caps on MIM depletion
and increases in apical caps on MIM overexpression. This is a part of new Figure 7.

3. Figure S4C-E and line 392: The effect of overexpressing Diaphanous is not as strong as
reducing DMIM. How about overexpressing a constitutively active form of Diaphanous (e.g. Dia-
delta(Dad), <https://pmc.ncbi.nlm.nih.gov/articles/PMC2793291/>)? Does that phenocopy the MIM
loss of function more closely?

Response: Removal of Dia leads to decrease in apical actin protrusions as shown in Figure
S5A in the revised manuscript. Overexpression of Dia with Dia-GFP gave an increase in apical
actin protrusions but this defect was not as strong as MIM depletion. Hence, as suggested, we
used the transgene expressing active Dia Δ DAD. We request the reviewer to note that
expression of this transgene leads to loss of fecundity and does not give any embryos with
maternal Gal4s. We were able to obtain embryos with a combination of maternal Gal4 and
Gal80 at 22 °C. We find that the embryos have a profound increase in actin protrusions. These
actin protrusions are formed despite the presence of MIM and Arp3. These data together show
that Dia supports the formation of apical actin protrusions.

4. Figure 8: the rescue experiment is really neat! However, I could not find anywhere
information about whether the individual Rac1DN or MIM-M3-mCherry treatments carried a
second UAS construct. The concern here is that when you combine the two UAS constructs,
the phenotypes may change because the Gal4 protein is split among the two UAS constructs
and their expression levels decrease. Thus, Rac1DN or MIM-M3-mCherry should be co-
expressed with a second UAS (e.g. luciferase, cerulean, etc.) to show that with the reduced
Gal4 dosage they can still cause the expected phenotypes.

Response: We thank the reviewer for this comment. We have added controls for taking care of
effects that may arise due to reduced dosage of the Gal4 in the presence of 2 transgenes as
compared to 1. We have added this data in Fig. 9G-K. We controlled for potential Gal4 dilution
effects on the actin-protrusion remodelling in these crosses by adding GFP and comparing
*Rac1*^{DN};DMIM-mCherry to *Rac1*^{DN};UAS-GFP and DMIM-mCherry;UAS-GFP. *Rac1*^{DN};UAS-GFP
showed abundant and longer actin protrusions as compared to controls and similar to *Rac1*^{DN}
alone (Fig. 8A-E). DMIM-mCherry;UAS-GFP showed shorter and reduced actin protrusions as
compared to controls and similar to DMIM-mCherry alone (Fig. 1F,H-I). In embryos expressing
DMIM-mCherry and *Rac1*^{DN} together, the length of actin protrusions was similar to
*Rac1*^{DN};UAS-GFP in interphase and showed a slight but significant decrease in metaphase (Fig.
9G,I,J). The average actin protrusion number per cap in *Rac1*^{DN};DMIM-mCherry expressing
embryos was however similar to *Rac1*^{DN}; UAS-GFP (Fig. 9I,K). The phenotype of reduced and
shorter protrusions as observed in embryos expressing DMIM-mCherry;UAS-GFP was not seen
in embryos expressing both DMIM-mCherry and *Rac1*^{DN} (Fig. 9H-I, J-K).

5. Figure 3G and line 217: the authors argue that the MIM-M3-delta(WH2) construct does not
become cytosolic, but the membrane signal appears to decrease, and the quantification in Fig.
3I does not show an obvious difference with MIM-M3 at 8.18 minutes. To this reviewer, it is not
clear that the WH2 domain is necessary for the membrane-to-cytosolic change in MIM
localization during metaphase. The authors should examine the localization of a MIM-M3-

delta(central scaffold) that still preserves the I-BAR and WH2 domains. Is the membrane
retention of that construct lower than that of MIM-M3-delta(Central scaffold+WH2)?

Response: We thank the reviewer for this comment on the functions of different domains of
MIM. In this manuscript, we have aimed to study the role of the I-BAR and WH2 domains in the
distribution of MIM to the membrane in distinct stages of the syncytial division cycle.

- 1. We have changed the representative images and quantification for domain deletion
constructs in Figure 3. We have included a new quantification of fluorescence
distribution on the membrane/furrow region as compared to background rather than the
previous quantification of membrane/furrow region to cytoplasmic ratio. DMIM-
deltaWH2-GFP (called DMIM-IBAR-CD-GFP in the revised manuscript) and DMIM-
deltaCD-deltaWH2-GFP (called DMIM-IBAR-GFP in the revised manuscript) show a
statistically significant increase in membrane localization as compared to the controls in
metaphase. This suggests that the WH2 domain is important for the regulation of MIM
loss at the metaphase stage.
- 2. We have further tested the role of the MIM I-BAR domain in supporting the role of apical
actin protrusion remodelling (Fig. 9 in the revised manuscript). We have tested a
combination of the DMIM-IBAR-GFP along with MIM RNAi to see if apical actin
protrusions are rescued in MIM depleted embryos. We find that the increase in
protrusions seen in MIM depleted embryos is suppressed by expressing the MIM-IBAR
domain. This also supports the recruitment of Arp3 (Fig. S6 in the revised manuscript).
Previous literature has shown that the MIM-IBAR domain binds F-actin and also Rac-
GTP (Bompard et al., 2005; Machesky and Johnston, 2007).
- 3. We were unable to test the role of CD and WH2 domains in affecting apical actin
protrusion remodelling or the recruitment of the WAVE regulatory complex in the
present study. We attempted to make combinations of the domain deletion transgenes
in the MIM mutant background but these combinations failed and did not give any
recombinants to allow further analysis. This is possibly due to their presence on the
same chromosome.
- 4. We appreciate the discussion on the requirement of the DMIM-deltaCD construct for
studying its role in MIM distribution and apical actin protrusion remodelling. We do not
have this construct and are in the process of making this. We hope to follow up the role
of different domains in recruitment of Rac-GTP, Wave regulatory complex and Actin
binding proteins in plasma membrane remodelling in syncytial *Drosophila* embryos in a
future study.

6. The use of Student's t-test to compare means is not appropriate unless the authors validate
that the distributions of the samples that they compare are normal. If normality is not tested,
then a non-parametric test (e.g. Mann-Whitney) should be used.

Response: We thank the reviewer for this comment. We have changed the statistics to
incorporate Mann Whitney tests as suggested. Also, as suggested by Reviewer 2, we have
changed the graphs into super plots with embryos and cells represented in distinct colors.

MINOR

7. Figure 6I-J: the authors argue that both Arp3 and DMIM localize to the base on microvilli in a
punctate distribution. However, from the images, the size of the Arp3 puncta seem smaller, and
the number of puncta greater. Quantitative analysis of the colocalization of MIM and Arp3 would
further support the idea that DMIM is controlling Arp2/3 recruitment to microvilli to promote their
remodelling.

Response: We have replaced the images for Arp3 and DMIM-mCherry. We find that DMIM-
mCherry is present on the membrane and Arp3 is present in on the membrane and in cortical
regions. The Pearson's coefficient analysis gives a value of 0.527 ± 0.146 .

8. Figure 1: this is picky, but the authors are not overexpressing DMIM, but part of it. As such, I
suggest to revise sentences such as "DMIM was overexpressed" (line 155) with "MIM-M3 was
overexpressed".

Response: We apologize for the confusion and thank both Reviewer 1 and 2 for pointing this
out. We are overexpressing the full-length DMIM splice variant that was amplified from the
*Drosophila* cDNA. This contains the I-BAR, central domain and WH2 domains. This was
wrongly represented in the schematic. We have corrected the schematic as shown in Fig.1A to
show that the MIM-M3-mCherry (DMIM-mCherry in the revised manuscript) construct is the full
length and contains the predicted I-BAR, central domain and WH2 domain. Since MIM-M3 was
a lab clone name, we have changed the name to DMIM-mCherry to avoid confusion.

We have shown immunofluorescence staining with the DMIM antibody in DMIM-
mCherry expressing embryos. We have now included a panel to show that the levels of MIM
staining are increased as compared to controls in DMIM-mCherry expressing embryos thereby
confirming that the line is indeed an overexpression of MIM (Fig. S1G,H).

TYPOS

9. Line 295: "changed DMIM depleted embryos" should be "changed in DMIM-depleted
embryos".

Response: We thank the reviewer for this comment. We have changed the line as suggested in
the revised manuscript.

**Reviewer #2:**

Summary of the revisions performed in response to the comments by Reviewer 2:

We performed the following revisions in response to the comments by Reviewer 2:

- 1. We have added analysis of Rho1 and Rac1 activity using fluorescent sensors. We find
that levels of active Rac-GTP depend upon MIM at the membrane, while presence of
MIM reduces levels of Rho-GTP.
- 2. We have assessed the role of the I-BAR domain of MIM in apical actin protrusion
remodelling defect of MIM depleted embryos.

- 3. We have added more representative images of MIM domains in their recruitment in the
syncytial cycle. We have redone the quantification in interphase and metaphase.
4. We have added data for Dynamin mutants *shl^{ts2}* in apical actin protrusion remodelling.
5. We have added statistics between all the genotypes.
6. We have changed the statistics to Mann-Whitney analysis and the plots to super plots.

Comments for Mitra et al

This paper examines the role of MIM and its associated protein in Drosophila syncytium
development. While MIM's presence at the cell junction is documented (PMID 36871754), the
new insight is its role in syncytium development and endocytosis affecting microvilli formation.
However, MIM is I-BAR protein for filopodia and protrusions, and how MIM is involved in
endocytosis needs molecular models. The work contains substantial work; however, the
molecular mechanisms behind it were not reported.

Response: We thank the reviewer for their comments. We have added new experiments as a
part of the revised manuscript. Among the new experiments, the following new experiments add
to mechanistic insights on how DMIM regulates plasma membrane remodelling.

- 1. We have added analysis of Rho1 and Rac1 activity using fluorescent sensors. We find
that levels of active Rac-GTP depend upon MIM at the membrane, while presence of
MIM reduces levels of Rho-GTP.
2. We find that the I-BAR domain of MIM can rescue the apical actin protrusion remodelling
defect of MIM depleted embryos.
3. We have better quantified the distribution of domain-deleted constructs in the revised
manuscript.
4. We have added data for Dynamin mutants *shl^{ts2}* in apical actin protrusion remodelling.

1. The use of MIM-M3, a truncated form, to fuse with mCherry is unclear. The DMIM localization
and MIM-M3 localization should be compared. Some key data should be re-analyzed by DMIM
tagged with mCherry instead of MIM-M3-mCherry.

Response: We apologize for the confusion and thank both Reviewer 1 and 2 for pointing this
out. This was wrongly represented in the schematic. Since the MIM-M3 was a lab name for the
clone, we have changed the name of the full length MIM construct to DMIM-mCherry. It
contains the I-BAR, central and WH2 domains. We have changed the schematic and also
changed the name as DMIM-mCherry to avoid confusion.

2. Endocytosis was shown by the uptake of FM dye in most of the figures. FM dye is lipophilic
dye it would not suggest the endocytosis of the membrane. Some complementary approaches
would be desired. Dynamin inhibitor treatment would be the candidate approach.

Response: FM143-FX dye is an amphipathic dye. It was incubated with living embryos for 15
478 min, then washed off. The dye shows increased fluorescence when it is trapped in vesicles after
479 the wash off. We saw an increase in fluorescence in vesicles in the interphase and prophase

stage of the syncytial cycle as compared to metaphase in control embryos. We saw a distinct
decrease in fluorescence in the MIM depleted embryos. This is consistent with previous studies
of increased association of Dynamin to the membrane in interphase and prophase and
decrease in levels at the furrow membrane in metaphase (Rikhy et al., 2015).

As suggested by the reviewer, we also used Dynamin mutant embryos, *shibire^{ts2}* for assessing
the change in actin protrusion remodelling and found that increased actin protrusions were
present in the metaphase stage of the embryos as compared to controls. This is a part of Figure
S2 A-B in the revised manuscript.

3. The mutant in Figure 3 should be used to rescue the mutant phenotype for a molecular
understanding of the action of MIM.

Response: We have added data for the rescue of the mutant phenotype of apical actin
protrusion remodelling defect in DMIM by combining it with DMIM-mCherry in the main figures
(Figure 1 and 3).

The I-BAR domain binds actin and negatively curved lipids. The I-BAR domain is crucial
for the dimerisation of the MIM protein and subsequent endocytosis which is independent of the
function of the central and WH2 domain (Cao et al., 2012). An overexpression of only the I-BAR
domain is potent enough to induce plasma membrane deformation (Mattila et al., 2003, 2007;
Woodings et al., 2003). We expressed the DMIM-IBAR-GFP transgene in the background of
MIM RNAi and found that the actin protrusion remodelling phenotype was rescued confirming
its role in actin protrusion remodelling. This also rescued Arp3 levels at the cortex (Fig. S6A,B).

The experiment for making combinations of the *dmim* mutants with the domain deleted
transgenes did not work. The transgenes are present on the same chromosome as the *dmim*
mutants. We attempted an experiment for making recombinants and this was not successful.
We will make transgenes on different chromosomes in the future, allowing us to test the
function of each domain in the absence of MIM. Since this could take almost 6 months to
generate, we have not been able to include this as a part of the current study. In the present
study, we have been able to assess the role of MIM domains in its spatiotemporal recruitment in
the syncytial division cycle and the role of the I-BAR domain in regulating apical actin protrusion
remodelling.

4. If Rac binds to MIM, then the binding region would be determined and then implemented by
the rescue experiments.

Response: The I-BAR domain of MIM is known to bind Rac-GTP (Bompard et al., 2005). In the
revised manuscript we tested the change in Rac-GTP levels by using the PAK3-RBD-GFP
sensor in MIM RNAi and DMIM-mCherry expressing embryos. We found that PAK3-RBD-GFP
fluorescence was increased in apical caps in DMIM-mCherry expressing embryos and
decreased in the apical caps in MIM RNAi expressing embryos. We also found that the actin
caps contained distinct regions of colocalization of DMIM-mCherry and PAK3-RBD-GFP (new
Fig. 7 in the revised manuscript).

We further overexpressed the DMIM-IBAR-GFP domain in the background of the MIM
RNAi and found that it rescued the excess apical actin protrusion phenotype for MIM depleted
embryos. This showed that the DMIM-IBAR domain regulates Rac-GTP levels to limit F-actin
remodelling in caps in syncytial *Drosophila* embryos.

5. Lines 134-135 state microvilli reduction from interphase to metaphase but lack statistical
comparison support.

Response: We have added the statistics for testing the reduction and this is a part of the
revised Figure 1.

6. Lines 198-199 discuss decreased MIM-M3-mCherry intensity from interphase to metaphase
without showing significance in related graphs, including Fig 3I.

Response: We have changed this quantification to provide fluorescence levels in interphase
and metaphase and have added statistics for the same. We show that DMIM-mCherry levels
(previously called MIM-M3-mCherry) are significantly reduced from interphase to metaphase.

7. Lines 356-358 mention Arp3 distribution without measuring co-localization levels. Provide co-
localization values to confirm insignificance.

Response: We have revised this paragraph based on the distribution of Arp3 and MIM. MIM is
a membrane associated protein and is found at the furrow whereas Arp3 is found in the
membrane cortex but is present more in cortical regions inside the cap as compared to MIM.
The Pearson's coefficient analysis gives a value of 0.527 ± 0.146 . We have added this value to
the revised manuscript.

8. Fig 7G-H states Rac1DN leads to loss of DMIM antibody fluorescence, but images show
remaining fluorescence-quantify this difference.

Response: There is a significant loss of DMIM as shown by the quantification in the revised
version of the manuscript. This is a part of new Figure 8H-I.

9. Fig 8F-G describes partial recovery of microvilli defects with Rac1DN overexpression but
lacks quantification.

Response: We have added quantification and statistics for this figure as a revised Figure 9.

10. Specify the number of microvilli considered for length measurement in Fig 8D.

Response: This information is a part of the Figure legend for revised Figure 9.

11. Improve Fig 9 illustration with more labeling and descriptions for clarity.

Response: As recommended by the reviewer, we have added more labels in the summary
schematic in revised Figure 10.

12. Differentiate cells and embryos with distinct colors in graphs, ensuring dots correspond to
their respective counts. The number of dots does not appear to correspond to a number of cells
or embryos.

Response: As recommended by the reviewer, we have changed the plots to superplots to add
details about embryos and cells in each plot. We have now changed the statistical analysis to
Mann-Whitney test as suggested by Reviewer 1.

Reviewer #3:

Summary of the revisions performed in response to the comments by Reviewer 3:

We performed the following revisions in response to the comments by Reviewer 3:

1. We have changed the name microvilli to apical actin protrusions or villous protrusions.

2. We have added images of membrane stained embryos with F-actin.

3. We have changed the figure with domain analysis to have all the genotypes and also
changed the quantification to better document the differences in distribution in
interphase and metaphase.

4. We have added new experiments with Rac-GTP sensor PAK3-RBD-GFP and with the
589 RacGEF Sponge RNAi.

5. We have added an experiment of staining of DMIM-mCherry expressing embryos to
show this is an overexpression.

BAR proteins contribute to linking the plasma membrane with the actin cortex and the dynamics
of the plasma membrane. The family-defining and membrane-binding BAR domain provides a
curved surface, which bends membranes or senses curvature. In addition, they contain various
domains, such as WH2 or SH3, which interact with the actin cytoskeleton.

Although having been studied for a number of years, the functions of BAR proteins including
598 MIM have remained unclear, especially how the well-defined molecular activities contribute to
599 specific roles in complex cell physiology, development and morphogenesis.

Here, Mitra et al. report a study of the I-BAR protein MIM membrane surface structure in
syncytial Drosophila embryos.

The claims of the study are that MIM mediates a balance between Arp2/3-dependent branched
actin and Dia dependent linear actin, which is observed in more and longer microvilli following
MIM depletion, that MIM mediates cortical localization of Arp2/3 and some of its regulators, that
Rac signaling mediates membrane localization of MIM.

The authors focus on nuclear cycles in syncytial embryos, which is characterized by spatially
separated domains, (1) caps, rich in branched F-actin with Rac signaling and Arp2/3 activity
and (2) the region outside of the disc-like caps, rich Rho signaling, myoII and dia, which will

extend in mitosis to form the pseudo cleavage furrow. The authors start with the observation of
micro meter long filapodia like protrusions, which are more abundant and longer in MIM
depleted embryos than in wild type and less so in embryos overexpressing MIM. These
protrusions are scored in surface views of the embryos and are thus laterally oriented extending
from the caps towards the region between the caps. The authors also describe a reduced
length of the metaphase furrow in MIM depleted embryos, which is comparable to dia mutants.
It is confusing to me why the lateral protrusions are designated as microvilli, which would expect
at the apical surface of the caps in a perpendicular orientation. These micrometer long
protrusions do not look like typical microvilli to me, because of their length and because of their
orientation. I would like to see either a demonstration that the microvilli in the caps are also
affected in MIM embryos or alternatively a less-biased naming of the structures. In any case the
micrometer-long protrusions should be better characterized. Are these F-actin structures
surrounded by membrane? Are they labeled by bundling proteins?

**Response:** As mentioned by the reviewer, we have imaged and quantified the apical actin
protrusions in actin caps in grazing sections. We have changed the name to 'villous projections'
or 'apical actin projections'. The term villous projections has been previously used by
(Fabrowski et al., 2013) for apical projections in cellularization. The villous projections or apical
actin projections have been called "microprojections" and "microvilli" in (Turner and Mahowald,
1976). In confocal and STED sections these projections are present in the very first optical
section in the cap in apical regions. They are abundant at the cap periphery, and the
protrusions in the apical central region of the caps are not visible in these optical sections.

As suggested, the protrusions were imaged in embryos expressing a membrane and F-actin
label in the revised manuscript. We stained embryos expressing GFP tagged PH domain of
general receptor for phosphoinositides-1 (GRP1) that localizes to plasma membrane in the
presence of phosphatidylinositol-3,4,5-trisphosphate (PIP3) (tGPH) (Britton et al., 2002) with
fluorescently coupled phalloidin to label F-actin in the syncytial division cycle 12 (Figure S1 in
the revised manuscript)(Britton et al., 2002). There is a colocalization of tGPH and fluorescently
tagged phalloidin in these embryos. We have used fluorescently coupled phalloidin for
quantification of apical actin protrusions, as tGPH is not as sharp as fluorescently coupled
phalloidin. There is a distinct increase in these apical actin protrusions in the MIM depleted
embryos.

We have added an experiment showing that the apical actin protrusions were also labelled with
Dia-GFP in the revised Figure 5B. Dia is further increased in MIM depleted embryos (Figure 5I).
Also, loss of Dia in Dia RNAi expressing embryos leads to a complete loss of protrusions
(Figure S5A). These apical actin labelled projections are different from filamentous actin found
in the cap, as documented by (Jiang and Harris, 2019).

Dia antibody staining does not label the apical protrusions with clarity because the bright
staining in the furrow region in between the caps obscures the weaker actin protrusion signal in
control embryos. It is seen in some of the villi. The Dia-GFP on the other hand labels the actin

protrusions very distinctly and has been shown to document the Dia distribution on apical actin
protrusions in revised Figure 5B.

We have made the name changes to apical actin protrusions and added these descriptions to
the introduction and results section of the revised manuscript.

In fig.2 the authors present data with a membrane label. Here not such micro-meter long
protrusions are observed. The images look very different, also because the images are shown
in a smaller magnification. For clarity and easy readability, it would be advantageous to have
the F-actin and membrane label side by side in the same magnification and ideally with double
labelling. The same scoring should be used.

Response: We have added a new experiment to show the colocalization of tGPH with
fluorescently tagged phalloidin in revised Figure S1. This shows a more widespread distribution
for tGPH as compared to phalloidin. The apical actin protrusions are counted at the apical cap
periphery from the first optical section, and they are in between adjacent caps. The membrane
label tGPH is further used in Figure 2 to show cap and furrow dynamics. We have added figures
in the inset in these movies to show protrusions at the cap edge in the different genotypes. The
apical actin protrusions are visible at the cap edges as micron-scale structures.

The dynamics of MIM was assayed with a tagged version in a wild type and MIM mutant
background. In my eyes the presented data are not fully consistent. MIM-Cherry in a mutant
background with no potential competition between tagged and untagged molecules, Fig. S1F,
shows some enrichment in particles apical to the nuclei. I do not see an enrichment in the
region that becomes the metaphase furrow. In contrast, in case of competition, the label is
clearly enriched in the furrow region. Clarification is required. It is also not clear to me, why the
more convincing experiment is shown in the supplement, whereas the potentially more
complicated experiment is shown in the main figures.

Response: As suggested by the reviewer, we have added the combination *dmim*; DMIM-
mCherry in the main Figure 3. We have added the images to be able to compare the *dmim*;
DMIM-mCherry, and the DMIM-mCherry overexpression. We have re-quantified the distribution
of all the transgenes in interphase and metaphase. DMIM-mCherry is reduced on the
metaphase furrow as compared to interphase in both *dmim*; DMIM-mCherry, and the DMIM-
mCherry overexpression.

In Figure 5+6 the authors show a dependance of Arp3, Dia and regulators on MIM. Although the
DAPI channel shows the interphase status of the cells, F-actin is present in the furrows but not
in caps. In interphase F-actin, as well as Arp2/3 and its regulators are supposed to be largely
present in the apical actin caps while dia and Myo are present in the furrow region, as I
understand. Clarification is required probably by providing axial views.

Response: The loss of fluorescence for Arp3, Scar and Cortactin is throughout the cap and
furrow regions in MIM depleted embryos. The sagittal views are very grainy, and for comparison

we have added the apical-most section for Arp3, Scar and Cortactin in Fig. S2E-G in the
revised manuscript.

In Figures 7+8, the authors investigate the relationship of Rac signaling and MIM. They employ
expression of a dominant negative Rac mutant to suppress Rac signalling. Staining for F-actin
shows that this induces a comparable phenotype as loss of MIM concerning the protrusions and
reduction of endocytosis. The authors conclude that MIM depends on Rac signaling because
the overall staining and enrichment is reduced in the furrow region. Formally the conclusion is
valid. I am confused however, because Rac signaling is restricted to the cap region but an
effect is observed outside of the cap region. This indicates me, that the effect is most likely
indirect. The authors may complement their data with an alternative interference of Rac
signaling such as with mutants of its upstream GEFs, Sponge-ELMO.

Response: We thank the reviewer for these comments, we have addressed these comments in
the following 3 ways in the revised manuscript:

1. We have evaluated the distribution and levels of PAK3-RBD-GFP, the Rac-GTP sensor
in the syncytial division cycle in controls, MIM RNAi and DMIM-mCherry expressing
embryos. PAK3-RBD-GFP is clearly present in apical caps and at the furrow region in
controls (Zhang et al., 2018). We find that PAK3-RBD-GFP is enriched at the cap in
DMIM-mCherry expressing embryos in the cap region, and it decreases significantly at
the cap as compared to controls in MIM RNAi expressing embryos. PAK3-RBD-GFP
colocalizes in patches in the apical regions with DMIM-mCherry. This is a part of Figure
7 in the revised manuscript.

721

2. We have also carried out an analysis of apical actin protrusions in ELMO/sponge RNAi
expressing embryos and found that they have apical actin protrusion defects similar to
MIM. This is part of new Figure 8 in the revised manuscript.

724

3. We have also used Arp3 RNAi to test the defect in apical actin protrusions and find that
there is an increase similar to MIM depleted embryos. We add the Arp3 RNAi data here:

Response letter Figure 1: (A) Representative grazing section images showing nuclei labelled
with Hoechst (grey), Arp3 (green), cortical F-actin labelled with Alexa Fluor Phalloidin 568 (red)
in control and *arp3*^Δ embryos. (B) Representative grazing section images showing nuclei
labelled with Hoechst (grey), Dia (green) and cortical F-actin labelled with Alexa Fluor Phalloidin
568 (red) in control and *arp3*^Δ embryos. (C) Graph showing the quantification of the normalised
apical intensity of Dia in control and *arp3*^Δ embryos. n=3 cells from 3 embryos for each
genotype. (D-E) STED images showing the F-actin containing actin protrusions at interphase
and metaphase of nuclear cycle 12 for control (D) and *arp3*^Δ (E) embryos. The region in the
yellow box is further zoomed in to show the actin protrusions. (F-G) Graphs showing the actin
protrusions length (F) and numbers per apical cap (G) at interphase and metaphase, n=2-3
cells from 3 embryos for each genotype.

In summary, this interesting study provides novel insights into how the I-BAR protein MIM
contributes to the antagonism of linear and branched F-actin at the cortex. I recommend
publication of the study. Before moving to publication clarification of several issues is needed
partially involving substantial new experiments and data.

Response: We thank the reviewer for the encouraging comments on this study.

Specific comments

- MIM mutants are viable and fertile. No overall morphological defects in syncytial blastoderm
are apparently observed and nuclear divisions proceed fine in mutants. In my view it is an
overstatement to designate MIM a "key integrator" (l. 33)

Response: MIM mutant and RNAi embryos are partially lethal. They show 30% lethality when
grown at 25 °C. We have added this information to the revised manuscript. They do show an
increase in nuclear fallouts, a defect commonly observed in many actin cytoskeleton
remodelling mutant embryos. Based on the reviewers suggestion, we have revised this

statement as follows: "These findings identify MIM as an integrator of actin and endocytic
dynamics that enables rapid membrane remodeling in the syncytial division cycles."

- L. 44. Although previous studies have demonstrated an antagonism between Arp2/3-
dependent branched actin and Dia-dependent linear F-actin, I am not aware of studies showing
that the antagonism would be due to a competition for the free pool of G-actin molecules.

Response: Competition for free G-actin has been shown recently by using FRAP in *Drosophila*
embryos depleted of actin nucleating proteins (Xie et al., 2021). The recovery time in FRAP
experiments decreased in ArpC4 RNAi, Dia RNAi, Cortactin RNAi, Coronin RNAi and DPPOD1
RNAi expressing embryos as compared to controls. The recovery time increased upon
stabilization of F-actin using Jasplakinolide. We have added this to the introduction and
discussion in the revised manuscript.

- L. 154. Tagged MIM was expressed from a UAS transgene. Is it clear that this results in
"overexpression"? The authors should check expression levels in comparison to endogenous
MIM by western blot or alternatively avoid the term "overexpression".

Response: The MIM antibody generated by us with the help of Boster Bio did not work on
Western blots. It shows a good antibody staining in control embryos, and this staining is lost in
MIM RNAi and in the MIM mutant embryos. We have now included a panel to show that the
levels of MIM staining are increased as compared to controls in DMIM-mCherry expressing
embryos thereby confirming that the line is indeed an over expression of MIM (Fig. S1G-H)

- Fig. 4. Assay for endocytosis with FM dye. The images in panel A and the quantification in B
are not consistent. Panel A shows a more or less complete loss of signal (black images).
Quantification shows, however, shows a reduction to not even half.

Response: We thank the reviewer for pointing this out. We have shown a more representative
image for this experiment. We have chosen an image that is approximately the average after
obtaining the quantification.

- Dia and Rac1 were localized by antibody staining. I did not spot any source for the Rac1
antibody. As the antibody does not distinguish active and inactive forms of the proteins, the
stainings are not very informative. The authors may employ probes for the active forms, i. e. a
Rac bio sensor.

Response: We have included the source of the Rac antibody. We thank the reviewer for
suggesting the use of the Rac sensor. We have added new experiments for the change in the
Rac sensor PAK3-RBD-GFP in controls, MIM RNAi and DMIM-mCherry. We find that the
PAK3-RBD-GFP fluorescence in the apical cap reduces significantly in MIM RNAi and
increases significantly in DMIM-mCherry overexpression, thus confirming the role of MIM in
supporting levels of Rac-GTP at the plasma membrane. Also there is a distinct colocalization of
DMIM-mCherry and PAK3-RBD-GFP in patches in the apical cap.

References:

- Bompard, G., S.J. Sharp, G. Freiss, and L.M. Machesky. 2005. Involvement of Rac in actin
cytoskeleton rearrangements induced by MIM-B. *J. Cell Sci.* 118:5393–5403.
- Britton, J.S., W.K. Lockwood, L. Li, S.M. Cohen, and B.A. Edgar. 2002. Drosophila's insulin/PI3-
kinase pathway coordinates cellular metabolism with nutritional conditions. *Dev. Cell.*
2:239–249.
- Cao, M., T. Zhan, M. Ji, and X. Zhan. 2012. Dimerization is necessary for MIM-mediated
membrane deformation and endocytosis. *Biochem. J.* 446:469–475.
- Dey, B., and R. Rikhy. 2020. DE-cadherin and Myosin II balance regulates furrow length for
onset of polygon shape in syncytial Drosophila embryos. *J. Cell Sci.*
doi:10.1242/jcs.240168.
- Fabrowski, P., A.S. Necakov, S. Mumbauer, E. Loeser, A. Reversi, S. Streichan, J.A.G. Briggs,
and S. De Renzis. 2013. Tubular endocytosis drives remodelling of the apical surface
during epithelial morphogenesis in Drosophila. *Nat. Commun.* 4:2244.
- Jiang, T., and T.J.C. Harris. 2019. Par-1 controls the composition and growth of cortical actin
caps during Drosophila embryo cleavage. *J. Cell Biol.* 218:4195–4214.
- Machesky, L.M., and S.A. Johnston. 2007. MIM: a multifunctional scaffold protein. *J. Mol. Med.*
85:569–576.
- Mattila, P.K., A. Pykäläinen, J. Saarikangas, V.O. Paavilainen, H. Vihinen, E. Jokitalo, and P.
Lappalainen. 2007. Missing-in-metastasis and IRSp53 deform PI(4,5)P2-rich membranes
by an inverse BAR domain-like mechanism. *J. Cell Biol.* 176:953–964.
- Mattila, P.K., M. Salminen, T. Yamashiro, and P. Lappalainen. 2003. Mouse MIM, a tissue-
specific regulator of cytoskeletal dynamics, interacts with ATP-actin monomers through its
C-terminal WH2 domain. *J. Biol. Chem.* 278:8452–8459.
- Mitra, D., A. Swaminathan, G. Mundhe, and R. Rikhy. 2022. Imaging and quantification of apical
microvilli in the syncytial blastoderm of Drosophila embryos. *STAR Protoc.* 3:101736.
- Rikhy, R., M. Mavrakis, and J. Lippincott-Schwartz. 2015. Dynamin regulates metaphase furrow
formation and plasma membrane compartmentalization in the syncytial Drosophila embryo.
*Biol. Open.* 4:301–311.
- Sherlekar, A., G. Mundhe, P. Richa, B. Dey, S. Sharma, and R. Rikhy. 2020. F-BAR domain
protein Syndapin regulates actomyosin dynamics during apical cap remodeling in syncytial
Drosophila embryos. *J. Cell Sci.* 133. doi:10.1242/jcs.235846.
- Turner, F.R., and A.P. Mahowald. 1976. Scanning electron microscopy of Drosophila
embryogenesis. 1. The structure of the egg envelopes and the formation of the cellular
blastoderm. *Dev. Biol.* 50:95–108.
- Woodings, J.A., S.J. Sharp, and L.M. Machesky. 2003. MIM-B, a putative metastasis
suppressor protein, binds to actin and to protein tyrosine phosphatase δ . *Biochem. J.*

371:463–471.

Xie, Y., R. Budhathoki, and J.T. Blankenship. 2021. Combinatorial deployment of F-actin
regulators to build complex 3D actin structures in vivo. *Elife*. 10. doi:10.7554/eLife.63046.

Zhang, Y., J.C. Yu, T. Jiang, R. Fernandez-Gonzalez, and T.J.C. Harris. 2018. Collision of
Expanding Actin Caps with Actomyosin Borders for Cortical Bending and Mitotic Rounding
in a Syncytium. *Dev. Cell*. 45:551–564.e4.

December 17, 2025

RE: JCB Manuscript #202502184R

Richa Rikhy
Indian Institute of Science Education and Research Pune

Dear Dr. Rikhy:

Thank you for submitting your revised manuscript entitled "MIM triggers formin to Arp2/3-based actin assembly in membrane remodeling in *Drosophila* embryos". The reviewers all now support publication so we would be happy to publish your paper in JCB pending final revisions necessary to meet our formatting guidelines (see details below).

In your final revision, please be sure to address the final concerns of reviewers #1 and #2.

A. MANUSCRIPT ORGANIZATION AND FORMATTING:

Full guidelines are available on our Instructions for Authors page, <http://jcb.rupress.org/submission-guidelines#revised>. **Submission of a paper that does not conform to JCB guidelines will delay the acceptance of your manuscript.**

**1) Text limits: Character count for Articles is < 40,000, not including spaces. Count includes abstract, introduction, results, discussion, and acknowledgments. Count does not include title page, figure legends, materials and methods, references, tables, or supplemental legends.

We realize that your text is currently over the 40,000 limit but we can allow the extra space in this case. If you need to add more text to address the final reviewer points, please aim to be as concise as possible.

2) Figures limits: Articles may have up to 10 main text figures.

3) Figure formatting: Scale bars must be present on all microscopy images, including inset magnifications. Molecular weight or nucleic acid size markers must be included on all gel electrophoresis. Aspect ratios of images may not be altered.

4) Statistical analysis: Error bars on graphic representations of numerical data must be clearly described in the figure legend. The number of independent data points (n) represented in a graph must be indicated in the legend. Statistical methods should be explained in full in the materials and methods. For figures presenting pooled data the statistical measure should be defined in the figure legends. Please also be sure to indicate the statistical tests used in each of your experiments (either in the figure legend itself or in a separate methods section) as well as the parameters of the test (for example, if you ran a t-test, please indicate if it was one- or two-sided, etc.). Also, if you used parametric tests, please indicate if the data distribution was tested for normality (and if so, how). If not, you must state something to the effect that "Data distribution was assumed to be normal but this was not formally tested."

5) Abstract and title: The abstract should be no longer than 160 words and should communicate the significance of the paper for a general audience. The title should be less than 100 characters including spaces. Make the title concise but accessible to a general readership.

6) Materials and methods: Should be comprehensive and not simply reference a previous publication for details on how an experiment was performed. Please provide full descriptions in the text for readers who may not have access to referenced manuscripts.

7) All antibodies, cell lines, animals, and tools used in the manuscript should be described in full, including accession numbers for materials available in a public repository such as the Resource Identification Portal. Please be sure to provide the sequences for all of your primers/oligos and RNAi constructs in the materials and methods. You must also indicate in the methods the source, species, and catalog numbers (where appropriate) for all of your antibodies. Please also indicate the acquisition and quantification methods for immunoblotting/western blots.

**8) Microscope image acquisition: The following information must be provided about the acquisition and processing of images:

a. Make and model of microscope

b. Type, magnification, and numerical aperture of the objective lenses

c. Temperature

d. Imaging medium

e. Fluorochromes

f. Camera make and model

g. Acquisition software

h. Any software used for image processing subsequent to data acquisition. Please include details and types of operations involved (e.g., type of deconvolution, 3D reconstitutions, surface or volume rendering, gamma adjustments, etc.).

10) Supplemental materials: There are strict limits on the allowable amount of supplemental data. Articles may have up to 5 supplemental figures. We realize that you currently have 6 supplemental figures, which is acceptable in this case. Please also note that tables, like figures, should be provided as individual, editable files. A summary of all supplemental material should appear at the end of the Materials and methods section.

**12) Conflict of interest statement: JCB requires inclusion of a statement regarding competing financial interests. If no competing financial interests exist, please include the following statement: "The authors declare no competing financial interests." If competing interests are declared, please follow your statement of these competing interests with the following statement: "The authors declare no further competing financial interests."

**13) ORCID IDs: ORCID IDs are unique identifiers allowing researchers to create a record of their various scholarly contributions in a single place. Please note that ORCID IDs are now *required* for all authors. At resubmission of your final files, please be sure to provide your ORCID ID and those of all co-authors.

**14) A separate author contribution section following the Acknowledgments. All authors should be mentioned and designated by their full names. We encourage use of the CRediT nomenclature.

Please note that JCB now requires authors to submit Source Data used to generate figures containing gels and Western blots with all revised manuscripts. This Source Data consists of fully uncropped and unprocessed images for each gel/blot displayed in the main and supplemental figures. For assays performed using capillary electrophoresis and/or immunoassay-based detection, authors should instead provide the electropherogram graph(s) for each experiment, plotting fluorescence/chemiluminescence intensity vs. molecular weight/size. Please be sure to provide one Source Data file for each figure gels, blots, and/or capillary electrophoresis assays along with your revised manuscript files. File names for Source Data figures should be alphanumeric without any spaces or special characters (i.e., SourceDataF#, where F# refers to the associated main figure number or SourceDataFS# for those associated with Supplemental figures). For traditional gels and blots, the lanes of the gels/blots should be labeled as they are in the associated figure, the place where cropping was applied should be marked (with a box), and molecular weight/size standards should be labeled wherever possible. For capillary electrophoresis assays, each trace in the graph should be color-coded and labeled to indicate which protein, gene, or sample is being measured (please try to avoid red/green combinations to accommodate our color-blind readers).

Journal of Cell Biology now requires a data availability statement for all research article submissions. These statements will be published in the article directly above the Acknowledgments. The statement should address all data underlying the research presented in the manuscript. Please visit the JCB instructions for authors for guidelines and examples of statements at (<https://rupress.org/jcb/pages/editorial-policies#data-availability-statement>).

B. FINAL FILES:

Please upload the following materials to our online submission system. These items are required prior to acceptance. If you have any questions, contact JCB's Managing Editor, Lindsey Hollander

(lhollander@rockefeller.edu).

Thank you for your attention to these final processing requirements. Please revise and format the manuscript and upload materials within 7 days. If you need an extension for whatever reason, please let us know and we can work with you to determine a suitable revision period.

Thank you for this interesting contribution, we look forward to publishing your paper in Journal of Cell Biology.

Sincerely,

Pekka Lappalainen
Monitoring Editor
Journal of Cell Biology

Gabriele Stephan
Scientific Editor
Journal of Cell Biology

Reviewer #1:

The authors have addressed most of my concerns. I would suggest not just reporting the correlation coefficient for the Arp3 and DMIM-mCherry signals, but also the statistical significance of that correlation (see for instance [https://stats.libretexts.org/Bookshelves/Introductory_Statistics/Introductory_Statistics_1e_\(OpenStax\)/12%3A_Linear_Regression_and_Correlation/12.05%3A_Testing_the_Significance_of_the_Correlation_Coefor_how_to_calculate_that_correlation](https://stats.libretexts.org/Bookshelves/Introductory_Statistics/Introductory_Statistics_1e_(OpenStax)/12%3A_Linear_Regression_and_Correlation/12.05%3A_Testing_the_Significance_of_the_Correlation_Coefor_how_to_calculate_that_correlation)) ... most software packages and programming languages that can be used to calculate correlation coefficients between two signals will also provide a P value measuring the significance of the correlation.

Reviewer #2:

The manuscript was significantly improved. The observation was consistent with the paper by Kawabata Galbraith, as the authors cited, and the novelty is in cell cycle progression. Discussion would be better if that discussion were included. MIM-dependent filopodia can be converted to extracellular vesicles, which is speculated to shorten the filopodia, and the elongation of actin-rich filopodia in the absence of MIM is consistent with the reduced scission of filopodia, which releases Rac1 (PMID 33756122, 41360804), which should be included in the discussion. The abstract wrote that Missing-in-Metastasis (MIM) (also called MTSS1), promotes branched actin network formation and endocytosis,.. However, MIM is not the direct inducer of actin polymerization nor branching, and thus, more prudent writing will be suitable. Furthermore, MIM appears to be involved in endocytosis, but this is not due to a direct molecular action of MIM, because MIM localization at endocytic pits was not shown here. Therefore, the abstract merits significant rewriting. The discussion should also address these limitations of the study.

Reviewer #3:

All my criticisms and comments have been satisfactorily addressed. The manuscript can be published according to my view.

The pointwise response to the editor and the reviewers' comments is given below.

December 17, 2025

RE: JCB Manuscript #202502184R

Richa Rikhy
Indian Institute of Science Education and Research Pune

Dear Dr. Rikhy:

Thank you for submitting your revised manuscript entitled "MIM triggers formin to Arp2/3-based actin assembly in membrane remodeling in Drosophila embryos". The reviewers all now support publication so we would be happy to publish your paper in JCB pending final revisions necessary to meet our formatting guidelines (see details below).

In your final revision, please be sure to address the final concerns of reviewers #1 and #2.

A. MANUSCRIPT ORGANIZATION AND FORMATTING:

Full guidelines are available on our Instructions for Authors page, <http://jcb.rupress.org/submission-guidelines#revised>. **Submission of a paper that does not conform to JCB guidelines will delay the acceptance of your manuscript.**

**1) Text limits: Character count for Articles is < 40,000, not including spaces. Count includes abstract, introduction, results, discussion, and acknowledgments. Count does not include title page, figure legends, materials and methods, references, tables, or supplemental legends.

We realize that your text is currently over the 40,000 limit but we can allow the extra space in this case. If you need to add more text to address the final reviewer points, please aim to be as concise as possible.

2) Figures limits: Articles may have up to 10 main text figures.

3) Figure formatting: Scale bars must be present on all microscopy images, including inset magnifications. Molecular weight or nucleic acid size markers must be included on all gel electrophoresis. Aspect ratios of images may not be altered.

4) Statistical analysis: Error bars on graphic representations of numerical data must be clearly described in the figure legend. The number of independent data points (n) represented in a

graph must be indicated in the legend. Statistical methods should be explained in full in the materials and methods. For figures presenting pooled data the statistical measure should be defined in the figure legends. Please also be sure to indicate the statistical tests used in each of your experiments (either in the figure legend itself or in a separate methods section) as well as the parameters of the test (for example, if you ran a t-test, please indicate if it was one- or two-sided, etc.). Also, if you used parametric tests, please indicate if the data distribution was tested for normality (and if so, how). If not, you must state something to the effect that "Data distribution was assumed to be normal but this was not formally tested."

5) Abstract and title: The abstract should be no longer than 160 words and should communicate the significance of the paper for a general audience. The title should be less than 100 characters including spaces. Make the title concise but accessible to a general readership.

6) Materials and methods: Should be comprehensive and not simply reference a previous publication for details on how an experiment was performed. Please provide full descriptions in the text for readers who may not have access to referenced manuscripts.

7) All antibodies, cell lines, animals, and tools used in the manuscript should be described in full, including accession numbers for materials available in a public repository such as the Resource Identification Portal. Please be sure to provide the sequences for all of your primers/oligos and RNAi constructs in the materials and methods. You must also indicate in the methods the source, species, and catalog numbers (where appropriate) for all of your antibodies. Please also indicate the acquisition and quantification methods for immunoblotting/western blots.

**8) Microscope image acquisition: The following information must be provided about the acquisition and processing of images:

- a. Make and model of microscope
- b. Type, magnification, and numerical aperture of the objective lenses
- c. Temperature
- d. Imaging medium
- e. Fluorochromes
- f. Camera make and model
- g. Acquisition software
- h. Any software used for image processing subsequent to data acquisition. Please include details and types of operations involved (e.g., type of deconvolution, 3D reconstitutions, surface or volume rendering, gamma adjustments, etc.).

Response: We have checked for the requested information as a part of points 1-9. We had already incorporated the requested information in the previous version of the revised manuscript.

10) Supplemental materials: There are strict limits on the allowable amount of supplemental data. Articles may have up to 5 supplemental figures. We realize that you currently have 6 supplemental figures, which is acceptable in this case. Please also note that tables, like figures, should be provided as individual, editable files. A summary of all supplemental material should appear at the end of the Materials and methods section.

Response: A summary of the supplementary figures and videos is added at the end of the materials and methods section.

Response: The eTOC summary is added to the revised manuscript.

**12) Conflict of interest statement: JCB requires inclusion of a statement regarding competing financial interests. If no competing financial interests exist, please include the following statement: "The authors declare no competing financial interests." If competing interests are declared, please follow your statement of these competing interests with the following statement: "The authors declare no further competing financial interests."

**13) ORCID IDs: ORCID IDs are unique identifiers allowing researchers to create a record of their various scholarly contributions in a single place. Please note that ORCID IDs are now *required* for all authors. At resubmission of your final files, please be sure to provide your ORCID ID and those of all co-authors.

**14) A separate author contribution section following the Acknowledgments. All authors should be mentioned and designated by their full names. We encourage use of the CRediT nomenclature.

Response: A separate author contribution section based on CRediT nomenclature is added.

Please note that JCB now requires authors to submit Source Data used to generate figures containing gels and Western blots with all revised manuscripts. This Source Data consists of fully uncropped and unprocessed images for each gel/blot displayed in the main and supplemental figures. For assays performed using capillary electrophoresis and/or immunoassay-based detection, authors should instead provide the electropherogram graph(s) for each experiment, plotting fluorescence/chemiluminescence intensity vs. molecular weight/size. Please be sure to provide one Source Data file for each figure gels, blots, and/or capillary electrophoresis assays along with your revised manuscript files. File names for Source

Data figures should be alphanumeric without any spaces or special characters (i.e., SourceDataF#, where F# refers to the associated main figure number or SourceDataFS# for those associated with Supplementary figures). For traditional gels and blots, the lanes of the gels/blots should be labeled as they are in the associated figure, the place where cropping was applied should be marked (with a box), and molecular weight/size standards should be labeled wherever possible. For capillary electrophoresis assays, each trace in the graph should be color-coded and labeled to indicate which protein, gene, or sample is being measured (please try to avoid red/green combinations to accommodate our color-blind readers).

Source Data Figures should be provided as individual PDF files (one file per figure). Authors should endeavor to retain a minimum resolution of 300 dpi or pixels per inch. Please review our instructions for export from Photoshop, Illustrator, and PowerPoint here:

<https://rupress.org/jcb/pages/submission-guidelines#revised>

Journal of Cell Biology now requires a data availability statement for all research article submissions. These statements will be published in the article directly above the Acknowledgments. The statement should address all data underlying the research presented in the manuscript. Please visit the JCB instructions for authors for guidelines and examples of statements at (<https://rupress.org/jcb/pages/editorial-policies#data-availability-statement>).

B. FINAL FILES:

****The license to publish form must be signed before your manuscript can be sent to production. A link to the electronic license to publish form will be sent to the corresponding author only. Please take a moment to check your funder requirements before choosing the appropriate license.****

Thank you for your attention to these final processing requirements. Please revise and format the manuscript and upload materials within 7 days. If you need an extension for whatever reason, please let us know and we can work with you to determine a suitable revision period.

Thank you for this interesting contribution, we look forward to publishing your paper in Journal of Cell Biology.

Sincerely,

Pekka Lappalainen
Monitoring Editor
Journal of Cell Biology

Gabriele Stephan
Scientific Editor
Journal of Cell Biology

Reviewer #1:

The authors have addressed most of my concerns. I would suggest not just reporting the correlation coefficient for the Arp3 and DMIM-mCherry signals, but also the statistical significance of that correlation (see for instance [https://stats.libretexts.org/Bookshelves/Introductory_Statistics/Introductory_Statistics_1e_\(Open_Stax\)/12%3A_Linear_Regression_and_Correlation/12.05%3A_Testing_the_Significance_of_the_Correlation_Coefficient](https://stats.libretexts.org/Bookshelves/Introductory_Statistics/Introductory_Statistics_1e_(Open_Stax)/12%3A_Linear_Regression_and_Correlation/12.05%3A_Testing_the_Significance_of_the_Correlation_Coefficient) for how to calculate that correlation) ... most software packages and

programming languages that can be used to calculate correlation coefficients between two signals will also provide a P value measuring the significance of the correlation.

Response: We thank the reviewer for pointing this out, we have added the p value to the legends of Figure 6. The following information is added for the analysis of Pearson's coefficient in the revised manuscript in the results and figure legends: "Pearson's correlation coefficient for the co-localisation of Arp3 and DMIM-mCherry was found to be 0.527 ± 0.146 , $p < 0.0001$ ($n = df - 2$)."

Reviewer #2:

The manuscript was significantly improved. The observation was consistent with the paper by Kawabata Galbraith, as the authors cited, and the novelty is in cell cycle progression. Discussion would be better if that discussion were included.

Response: As suggested, we have added a comment on the similarity of MIM function in syncytial division cycles in *Drosophila* and dendrites in mammalian neurons in the discussion in the revised manuscript. We add the following sentence to the discussion: "In addition, our observations of DMIM loss giving rise to an increase in the formin Dia supported protrusions in *Drosophila* syncytial cycles is consistent with increase in DAAM supported dendritic protrusions."

MIM-dependent filopodia can be converted to extracellular vesicles, which is speculated to shorten the filopodia, and the elongation of actin-rich filopodia in the absence of MIM is consistent with the reduced scission of filopodia, which releases Rac1 (PMID 33756122, 41360804), which should be included in the discussion.

Response: We thank the reviewer for highlighting these studies on the role of MIM in generating extracellular vesicles and its potential role in shortening filopodia. We have added these studies to the discussion of the revised manuscript. However, we also request that the reviewer note that we do not observe vesicles labelled with DMIM in the intercap region during syncytial cycles in *Drosophila* embryos, and that this is unlikely to be the mechanism by which DMIM shortens protrusions at these stages.

The abstract wrote that Missing-in-Metastasis (MIM) (also called MTSS1), promotes branched actin network formation and endocytosis. However, MIM is not the direct inducer of actin polymerization nor branching, and thus, more prudent writing will be suitable. Furthermore, MIM appears to be involved in endocytosis, but this is not due to a direct molecular action of MIM, because MIM localization at endocytic pits was not shown here. Therefore, the abstract merits significant rewriting. The discussion should also address these limitations of the study.

Response: We find that MIM regulates the formation of the branched actin network and endocytosis. We have taken care to see that we do not state that MIM is directly involved in

endocytosis. In the abstract, we say that “We report that the I-BAR domain-containing protein, Missing-in-Metastasis (MIM) (also called MTSS1), promotes branched actin network formation and endocytosis to drive rapid, cyclical plasma membrane remodeling during syncytial divisions in *Drosophila* embryos.” We do not imply a direct role for DMIM in endocytosis in the abstract with this description. We find that decreased DMIM levels lead to reduced FM1-43FX endocytic uptake, whereas DMIM overexpression increases it. This is also coincident with changes in Rab5 levels. We add the following description in response to the reviewer’s comment for the role of DMIM in endocytosis in the results section: “We find that DMIM was present at the membrane and endocytosis increased on its overexpression; however future experiments will be needed to evaluate if DMIM is present at endocytic pits along with the branched actin network to directly mediate endocytic vesicle formation.” We also reiterate this in the Discussion section.

Reviewer #3:

All my criticisms and comments have been satisfactorily addressed. The manuscript can be published according to my view.

Response: We thank the reviewers for the feedback on our work.